UWThPh 2025-16

# Orthosymplectic Chern–Simons Matter Theories: Global Forms, Dualities, and Vacua

Fabio Marino[a], Sinan Moura Soysüren[a,b], and Marcus Sperling[a]

[a] *University of Vienna, Faculty of Physics, Mathematical Physics Group, Boltzmanngasse 5, 1090 Vienna, Austria,*

[b] *University of Vienna, Vienna Doctoral School in Physics, Boltzmanngasse 5, 1090 Vienna, Austria.*

Email: `fabio.marino@univie.ac.at`,
`sinan.moura.soysueren@univie.ac.at`,
`marcus.sperling@univie.ac.at`

## Abstract

A magnetic quiver framework is proposed for studying maximal branches of 3d orthosymplectic Chern–Simons matter theories with $\mathcal{N} \geqslant 3$ supersymmetry, arising from Type IIB brane setups with O3 planes. These branches are extracted via brane moves, yielding orthosymplectic $\mathcal{N} = 4$ magnetic quivers whose Coulomb branches match the moduli spaces of interest. Global gauge group data, inaccessible from brane configurations alone, are determined through supersymmetric indices, Hilbert series, and fugacity maps.

The analysis is exploratory in nature and highlights several subtle features. In particular, magnetic quivers are proposed as predictions for the maximal branches in a range of examples. Along the way, dualities and structural puzzles are uncovered, reminiscent of challenges in 3d mirror symmetry with orientifolds.

# 1  Introduction

The infrared (IR) properties of strongly coupled $2+1$-dimensional supersymmetric field theories can be studied in great detail, thanks to a rich web of dualities and the availability of indices and partition functions computations via localisation techniques.

In contrast to $3+1$ dimensions, the dynamics of $2+1$-dimensional theories is enriched by the possibility of a Chern–Simons interaction. On general grounds [1–3], the maximal amount of supersymmetry compatible with such an interaction is $\mathcal{N}=3$. However, it has long been known that Chern–Simons theories with extended supersymmetry are possible: $\mathcal{N}=4$ [4,5], $\mathcal{N}=5$ [6], $\mathcal{N}=6$ [7,8], and even $\mathcal{N}=8$ [9,10].

A powerful framework for realising such theories, and for understanding dualities among them, arises from Type IIB brane constructions. In particular, three-dimensional supersymmetric field theories emerge on the world-volume of D3 branes stretched between various types of $(p,q)$ 5-branes [11–14]. These setups typically yield $\mathcal{N}=3$ theories. If only two distinct types of 5-branes are present, the IR theory enjoys enhanced $\mathcal{N}=4$ supersymmetry. In special cases, such as circular brane configurations, $\mathcal{N}=5$ or 6 supersymmetry can be achieved [6–8].

A crucial feature of these brane systems is the ability to pass 5-branes through one another, resulting in equivalent configurations, provided D3-brane creation and annihilation [15] is properly accounted for. Since such moves lead to different low-energy theories on the D3 world-volume, they imply dualities among the corresponding field theories. The prototype of such a duality in Chern–Simons theories with unitary gauge groups was established in [16].

By including orientifold 3-planes (O3), one obtains brane systems that realise orthogonal and symplectic gauge groups. The associated brane creation/annihilation rules lead to dualities in Chern–Simons theories with such gauge groups. For symplectic gauge groups $\mathrm{Sp}(n) \cong \mathrm{USp}(2n)$, such dualities were proposed in [17]. Extensions to orthogonal gauge groups [18–20] require a careful treatment of the global form of the gauge group.

Of particular interest are Chern–Simons quiver theories with alternating $(\mathrm{S})\mathrm{O}(n)$ and $\mathrm{Sp}(n)$ gauge nodes, referred to as *orthosymplectic* Chern–Simons quivers. These theories naturally arise from brane configurations involving both O3-planes and $(p,q)$ 5-branes, and are the main focus of this work.

An important property of $2+1$-dimensional Chern–Simons theories, as with any quantum field theory, is the structure of their ground states. For highly supersymmetric $\mathcal{N} \geqslant 3$ Chern–Simons theories, there exists a continuous space of supersymmetric vacua, known as the moduli space. Depending on the R-symmetry — $\mathrm{SU}(2)_R$ for $\mathcal{N}=3$ and $\mathrm{SU}(2)_A \times \mathrm{SU}(2)_B$ for $\mathcal{N}=4$ SCFTs — the moduli space decomposes into maximal hyper-Kähler branches $\mathcal{B}_i$, each of which is a symplectic singularity [21]. In the brane realisation, these branches are associated with D3 segments that can move along a given type of $(p,q)$ 5-brane [15, 22–24]. Accordingly, the maximal branches $\mathcal{B}_i$ are naturally labelled by the type of 5-brane they correspond to.

Despite this transparent brane picture, a systematic field-theoretic description of these branches has been lacking [3,24–27]. One reason is that the branches are described in terms of dressed monopole operators. In the presence of a Chern–Simons interaction, monopole operators carry non-trivial gauge charges, and must be dressed by suitably charged matter fields to form gauge-invariant chiral operators. As a result, the maximal branches are genuinely quantum in nature, much like the now well-understood Coulomb branches of 3d $\mathcal{N}=4$ non–Chern–Simons gauge theories.

To overcome this difficulty, [28] proposed the use of magnetic quivers: auxiliary 3d $\mathcal{N}=4$ gauge theories whose Coulomb branches reproduce the maximal branches of CSM theories. As a proof of concept, it was shown that the maximal branches $\mathcal{B}_i$ of linear and circular brane systems can indeed be captured by magnetic quivers $\mathsf{MQ}_i$, derived

systematically through brane creation/annihilation moves (i.e. Giveon–Kutasov dualities). The corresponding world-volume theories include 3d $\mathcal{N} \geqslant 3$ linear and circular Chern–Simons quiver theories with unitary gauge groups and fundamental matter, as well as certain non-Lagrangian theories.

In this paper, a magnetic quiver framework is developed for Chern–Simons matter theories arising from linear and circular Type IIB brane configurations with O3 orientifold planes, which give rise to orthosymplectic gauge groups. The maximal quantum branches of such theories have received comparatively little attention. The proposed magnetic quivers are derived directly from the brane setup via D3-brane creation and annihilation moves and are themselves of orthosymplectic type.

While this extension may appear as a natural generalisation of the magnetic quiver proposal for unitary Chern–Simons theories [28], combined with known results on orthosymplectic magnetic quivers in higher-dimensional Type II setups [29–34], the present 3d Chern–Simons setting offers a key advantage: the proposed magnetic quivers can be tested explicitly. Unlike in the higher-dimensional cases, where independent computations of moduli spaces are typically unavailable, supersymmetric index computations, and Hilbert series limits are accessible here. The availability of these tools not only enables consistency checks, but also renders them necessary, as they provide crucial information about the global form of the gauge group and its transformation under brane duality moves, data not captured by the brane configurations themselves.

This analysis reveals several subtleties and open questions, including ambiguities in global structure and the emergence of unexpected dualities. These challenges echo puzzles in 3d mirror symmetry for orthosymplectic quivers [35–38], many of which also originate from brane systems with O3 planes [39]. The present work is therefore exploratory in nature and represents a first step toward a systematic understanding of orthosymplectic Chern–Simons theories through magnetic quivers.

The remainder of the paper is structured as follows. Section 2 introduces the theories under consideration. Section 3 discusses 3d $\mathcal{N} \geqslant 3$ Chern–Simons matter dualities, their brane realisations, and the necessary fugacity maps. Two fundamental duality moves are identified, forming the basis for validating the magnetic quiver construction. Section 4 presents the derivation of magnetic quivers for a specific class of brane systems, with detailed examples supported by perturbative checks in Appendix B. Section 5 examines a more subtle case beyond this class, highlighting structural puzzles and the potential need for unconventional operator counting. Conclusions and open questions are summarised in Section 6.

Additional background and computational details are collected in the appendices.

**Notation.** In this paper, different graphical tools are employed:

- The 3d supersymmetric QFTs are displayed through the quiver notation: round nodes ( $\bigcirc$ ) encode 3 $\mathcal{N} = 4$ vector multiplets, while round nodes with a blue CS-level on top represent $\mathcal{N} = 2$ vector multiplets; edges encode a pair of chiral multiplets in conjugate representations; square nodes ( $\square$ ) encode flavour symmetries. The group type is written below the nodes: for gauge groups the U/SO/Sp notation is employed, whereas for flavour symmetries the $A/B/C/D$ convention is used.

- Type IIB brane configurations are composed of D3 branes ($-$), NS5 branes ( $|$ ), $(p,q)$ 5-branes ( $\diagup$ , $\diagup$ , ... ), and D5 branes ($\otimes$). See Appendix A.1 for translating branes into quiver theories. Moreover, in most of the considered brane configurations, orientifold planes are present: O3$^+$ ($\cdots$), $\widetilde{\text{O3}}^+$ ($\cdots$), O3$^-$ (—) and $\widetilde{\text{O3}}^-$ ( ). See Section 2 for details on these conventions.

# 2 Orthosymplectic 3d $\mathcal{N} \geqslant 3$ Chern–Simons Matter Theories

To begin with, the relevant Type IIB brane configurations for 3d orthosymplectic Chern–Simons Matter theories are introduced and the $\mathrm{SL}(2, \mathbb{Z})$ duality group action on the branes and orientifolds is recalled. As orthogonal gauge group factors play a prominent role, one also needs to pay attention to global forms and how to access them via 3d $\mathcal{N} = 2$ superconformal index and 3d $\mathcal{N} = 4$ Coulomb branch Hilbert series computations.

## 2.1 Brane systems with orientifolds

Consider the 3d $\mathcal{N} \geqslant 3$ Chern–Simons matter theories that can be realised as world-volume theories of D3 branes in between NS5 and $(1, q)$ 5-branes [11–14]. The resulting theories are composed of unitary gauge group factors (with or without Chern–Simons level) together with charged matter fields transforming in the (anti-)fundamental representation of some gauge group factors, see Appendix A.1 for a summary. If O3 planes are added, the orientifold projection leads to orthogonal and symplectic gauge groups, depending on the O3-type, see Table 1.

The setup gives rise to a 3d $\mathcal{N} = 4$ SCFT if there are only two types of distinct 5-branes, here chosen to be NS5 and $(1, q)$ 5-branes. If there are more than two distinct 5-branes, then the IR theory is generically a 3d $\mathcal{N} = 3$ SCFT.

| O3 plane | Branes | Group | O3 plane | Branes | Group |
|:---:|:---:|:---:|:---:|:---:|:---:|
| $\mathrm{O3}^+$ | $\overset{N}{\cdots\cdots\cdots}$ | $\mathrm{Sp}(N)$ | $\mathrm{O3}^-$ | $\overset{N}{\rule{0pt}{0pt}}$ | $\mathrm{SO}(2N)$ |
| $\widetilde{\mathrm{O3}}^+$ | $\overset{N}{\text{-----}}$ | $\mathrm{Sp}'(N)$ | $\widetilde{\mathrm{O3}}^-$ | $\overset{N}{\rule{0pt}{0pt}}$ | $\mathrm{SO}(2N+1)$ |

**Table 1:** Conventions for O3 planes (following [23]). Each variant is denoted by a different stroke in the brane setup separating 2 stacks of $N$ half D3-branes: $\mathrm{O3}^+$ a dotted line, $\widetilde{\mathrm{O3}}^+$ a dashed line, $\mathrm{O3}^-$ an empty line, $\widetilde{\mathrm{O3}}^-$ a solid line. The right column displays the gauge group on the D3 world-volume after taking the orientifold projection into account. The difference between $\mathrm{Sp}(N)$ and $\mathrm{Sp}'(N)$ is not relevant here; thus, both $\mathrm{O3}^+$ and $\widetilde{\mathrm{O3}}^+$ are understood as giving rise to symplectic gauge nodes $\mathrm{Sp}(N)$ in this work.

Reading the CSM theory from the brane system requires some care. For instance, semi-infinite $\widetilde{\mathrm{O3}}^-$ planes lead to an extra half-hypermultiplet, even if there are no other semi-infinite D3s. This single half-hypermultiplet is written below as $B_0 \cong \mathfrak{so}(1)$ flavour node.

In addition, for $\mathrm{Sp}(N)$ gauge groups with CS-level $q$ and $N_f$ fundamental half-hyper-multiplets, the cancellation of the Witten-anomaly [40] requires $N_f + 2q = 0 \bmod 2$. Hence, if $N_f$ is even, the CS-level $q \in \mathbb{Z}$ has to be integer (including trivial level); while, if $N_f$ is odd, a half-integer CS-level $q \in \mathbb{Z} + \frac{1}{2}$ is required for an anomaly free theory. (See also [41].)

## 2.2 $\mathrm{SL}(2, \mathbb{Z})$ duality group action

In type IIB superstring theory, the $\mathrm{SL}(2, \mathbb{Z})$ duality group, generated by $\mathcal{S}$ and $\mathcal{T}$ satisfying $\mathcal{S}^2 = -\mathbb{1}$ and $(\mathcal{S}\mathcal{T})^3 = \mathbb{1}$, acts on 5-branes as well as orientifold 3-planes.

To realise this action, first introduce the 5-brane charges by defining: $\mathrm{NS5} = (\pm 1, 0)$ and $\mathrm{D5} = (0, \pm 1)$. Then the action of the $\mathrm{SL}(2, \mathbb{Z})$ generators on a $(p, q)$ 5-brane reads [11]

$$(p, q) \xrightarrow{\mathcal{S}} (q, -p) \qquad :\Leftrightarrow \quad (p, q) = \mathcal{S}(q, -p), \tag{2.1a}$$

$$(p, q) \xrightarrow{\mathcal{T}} (p, q - p) \quad :\Leftrightarrow \quad (p, q) = \mathcal{T}(p, q - p). \tag{2.1b}$$

Moreover, under $\mathrm{SL}(2, \mathbb{Z})$ the O3 planes transform as [42]

$$\mathrm{O3}^+ \xleftrightarrow{\mathcal{S}} \widetilde{\mathrm{O3}}^-, \quad \widetilde{\mathrm{O3}}^+ \xleftrightarrow{\mathcal{S}} \widetilde{\mathrm{O3}}^+, \quad \mathrm{O3}^- \xleftrightarrow{\mathcal{S}} \mathrm{O3}^-, \tag{2.2a}$$

$$\mathrm{O3}^+ \xleftrightarrow{\mathcal{T}} \widetilde{\mathrm{O3}}^+, \quad \mathrm{O3}^- \xleftrightarrow{\mathcal{T}} \mathrm{O3}^-, \quad \widetilde{\mathrm{O3}}^- \xleftrightarrow{\mathcal{T}} \widetilde{\mathrm{O3}}^-. \tag{2.2b}$$

These transformations are crucial in CSM dualities and in magnetic quiver derivations.

## 2.3 Global forms and computational tools

A common caveat of brane systems with orthogonal and symplectic gauge groups in the world-volume theory is the choice of global form of the gauge groups, see for instance [35, 36, 38, 43]. Here the approach is the following: the brane systems provides guidance on what gauge algebras are realised in the theory; i.e. the classical Lie-algebras $B_n$, $C_n$, and $D_n$. But whether the orthogonal groups are realised as $\mathrm{SO}(n)$, $\mathrm{Spin}(n)$, $\mathrm{O}(n)$, or $\mathrm{Pin}(n)$ cannot be inferred from the brane system alone (at least as far as the authors are aware). Hence, the overall *choice* of the global form is an *additional input* when computations like supersymmetric indices or Hilbert series are employed.

These different global forms are related by (gauging) discrete symmetries. Consider an $\mathrm{SO}(n)$ gauge group in $2 + 1$ space-time dimensions, then there are two discrete 0-form symmetries associated with it: the magnetic symmetry $\mathbb{Z}_2^{\mathcal{M}}$ and the chirality $\mathbb{Z}_2^{\mathcal{C}}$. Denote the associated fugacities by $\zeta$ and $\chi$, respectively.

**Supersymmetric index.** Concretely, given the 3d $\mathcal{N} = 2$ index $\mathcal{I}_{\mathrm{SO}(N)}(\chi, \zeta)$ of a gauge theory with a $\mathrm{SO}(N)$-type node that is refined with $\chi$ and $\zeta$ fugacities for its $\mathbb{Z}_2^{\mathcal{C}}$ and $\mathbb{Z}_2^{\mathcal{M}}$ symmetries, one computes the index for other choice of global forms as follows:

$$\mathcal{I}_{\mathrm{O}(N)^+}(\zeta) = \frac{1}{2}\Big[\mathcal{I}_{\mathrm{SO}(N)}(\chi = +1, +\zeta) + \mathcal{I}_{\mathrm{SO}(N)}(\chi = -1, +\zeta)\Big], \tag{2.3a}$$

$$\mathcal{I}_{\mathrm{O}(N)^-}(\zeta) = \frac{1}{2}\Big[\mathcal{I}_{\mathrm{SO}(N)}(\chi = +1, +\zeta) + \mathcal{I}_{\mathrm{SO}(N)}(\chi = -1, -\zeta)\Big], \tag{2.3b}$$

$$\mathcal{I}_{\mathrm{Spin}(N)}(\chi) = \frac{1}{2}\Big[\mathcal{I}_{\mathrm{SO}(N)}(\chi, \zeta = +1) + \mathcal{I}_{\mathrm{SO}(N)}(\chi, \zeta = -1)\Big], \tag{2.3c}$$

$$\mathcal{I}_{\mathrm{Pin}(N)} = \frac{1}{2}\Big[\mathcal{I}_{\mathrm{Spin}(N)}(\chi = +1) + \mathcal{I}_{\mathrm{Spin}(N)}(\chi = -1)\Big]. \tag{2.3d}$$

The reader is referred to [44–49] for general background on the 3d $\mathcal{N} = 2$ supersymmetric index. More relevant for the purposes here, the index conventions for 3d $\mathcal{N} \geqslant 3$ theories with $\mathrm{SO}(n)$ vector multiplets have been given in [37].

**Hilbert series.** Another important tool for this work is the Coulomb branch Hilbert series for 3d $\mathcal{N} = 4$ non-CS gauge theories [50]. The complete treatment of the global forms associated with $\mathrm{SO}(n)$ gauge groups factors has been detailed in [35, 37]. Once the Hilbert series is computed with fugacities $\zeta$ and $\chi$, the Hilbert series for the other global forms can be computed analogously to (2.3).

# 3 Dualities for CSM theories

Dualities of Chern–Simons–matter (CSM) theories are central to the dynamics of three-dimensional gauge theories and often admit a transparent realisation in brane constructions. A key ingredient is the brane creation/annihilation process, first described in Type IIB $D5 - NS5 - D3$ setups [15], where the number of D3 branes stretched between adjacent 5-branes changes as they cross. This phenomenon extends to configurations with $(p, q)$ 5-branes, giving rise to the Giveon–Kutasov (GK) duality [16], and further generalises in the presence of O3 planes, leading to dualities of orthosymplectic gauge theories. In the following, GK duality for unitary theories is recalled (Section 3.1), its extensions to orthogonal and symplectic cases are reviewed (Section 3.2), and analogous dualities for orthosymplectic CSM quivers are studied (Section 3.3).

## 3.1 Giveon–Kutasov duality for unitary CSM theories

Based on arguments of preserving supersymmetry and conserving brane-charges, one finds the Giveon–Kutasov duality [16] showcased in Figure 1 to hold for a NS5 $- (1, \kappa) -$ D3 brane system.

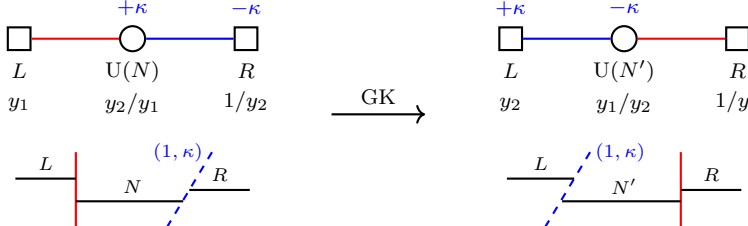

**Figure 1:** GK duality for a NS5 $- (1, \kappa) -$ D3 brane system. Taking the conservation of brane-charge and supersymmetry into account, one finds $N' = L + R - N + |\kappa|$. Below the gauge nodes, the corresponding topological fugacities have been written, while the fugacity reported under the flavour nodes are those of their background topological symmetries. These fugacity assignment is essential when one applies locally the GK duality (see [28, 51] for more details).

The GK duality can be locally implemented on a unitary CSM theory. For example, consider the two nodes quiver dubbed as (0) in Figure 2, one can apply the GK duality in Figure 1 on both of its nodes, obtaining the two dual theories named (1) and (2).

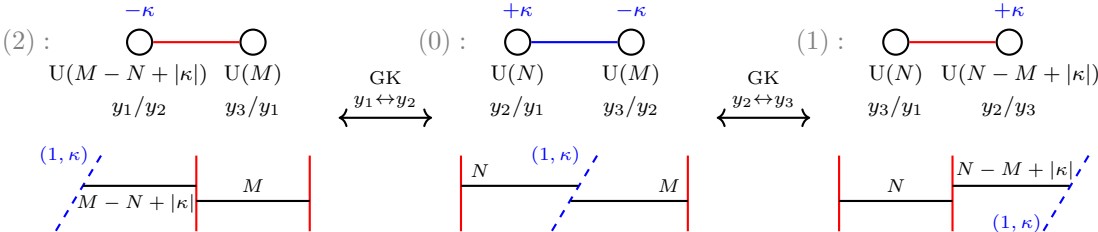

**Figure 2:** GK duality applied on a two nodes unitary CSM theory. Moving the $(1, \kappa)$ 5-brane to the right leads to the dual theory (1), while moving it to the left yields the dual theory (2). Close to each arrow, representing a GK transition, the associate fugacity map is indicated, as derived from the duality in Figure 1.

The GK duality for unitary CSM theories is employed as the basic construction principle for the magnetic quivers of this class of CSM theories, as discussed in [28] and reviewed in Section 4.1.

## 3.2 Dualities for orthogonal or symplectic CSM theories

The GK-duality has been extended to orthogonal and symplectic gauge groups in [17, 18]. For the purposes of this paper, recall the following $\mathcal{N} = 3$ field theory dualities (see also [19, 20]) involving one symplectic or orthogonal gauge node with a CS term:

$$
\begin{cases}
\text{if } q = 2\kappa \Rightarrow \quad {}^{N_c' = N_f - N_c + |\kappa| - 1}_{\quad G_{N_f} = D_{N_f}} \\[4pt]
\text{if } q = 2\kappa + 1 \Rightarrow \quad {}^{N_c' = N_f - N_c + |\kappa|}_{\quad G_{N_f} = B_{N_f}}
\end{cases}
\qquad , \qquad (3.1a)
$$

$$
\chi' = \chi\zeta \, , \ \zeta' = \zeta \qquad \qquad (3.1b)
$$
$$
N_c' = 2N_f - N_c + |q| + 2
$$

where below the orthogonal groups have been written the fugacities $\zeta$ and $\chi$ for the magnetic and charge conjugation symmetry respectively. These dualities manifest in the NS5 and $(1, q)$ 5-branes systems in the presence of O3 orientifolds in terms of D3 brane creation and annihilation; see Appendix A.2. Separating in $q = 2\kappa$ even (see Figure 3) and $q = 2\kappa + 1$ odd (see Figure 4), the dualities (3.1) are reproduced. Note that the $L$ red and $R$ blue flavours of Figures 3 and 4 can be seen as the splitting of the $N_f = L + R$ black flavours of (3.1), which are $\mathcal{N} = 3$ dualities and hence no axial charge is in the game. In Figures 3 and 4, some flavour nodes carry a background CS-level, determined by the $(1, q)$ 5-brane. This is useful whenever one works locally on a theory and can assign a CS term to a specific node, even if the node is non-dynamical.

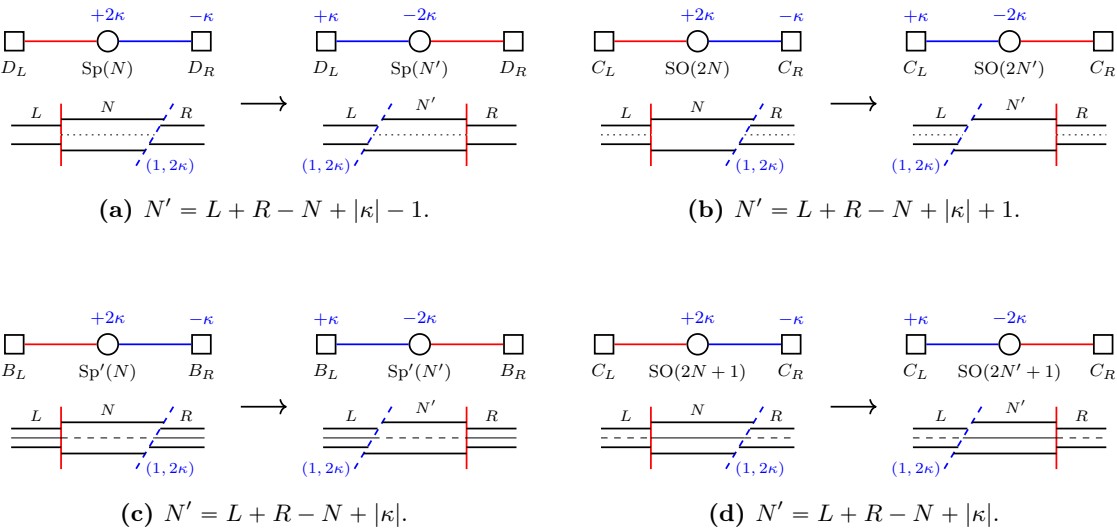

**(a)** $N' = L + R - N + |\kappa| - 1.$  **(b)** $N' = L + R - N + |\kappa| + 1.$

**(c)** $N' = L + R - N + |\kappa|.$  **(d)** $N' = L + R - N + |\kappa|.$

**Figure 3:** The basic OSp CSM dualities for the NS5 $-(1, 2\kappa)-$ D3 $\mathcal{N} = 4$ brane system with the four possible combinations of O3 planes.

## 3.3 Dualities for orthosymplectic CSM quivers

The $\mathcal{N} = 3$ dualities (3.1) can be seen as basic building blocks for dualities between 3d $\mathcal{N} \geqslant 3$ orthosymplectic CSM quivers. Here, such dualities between CSM quivers are considered and their fugacity maps are established in 2 and 3-nodes CSM quiver theories.

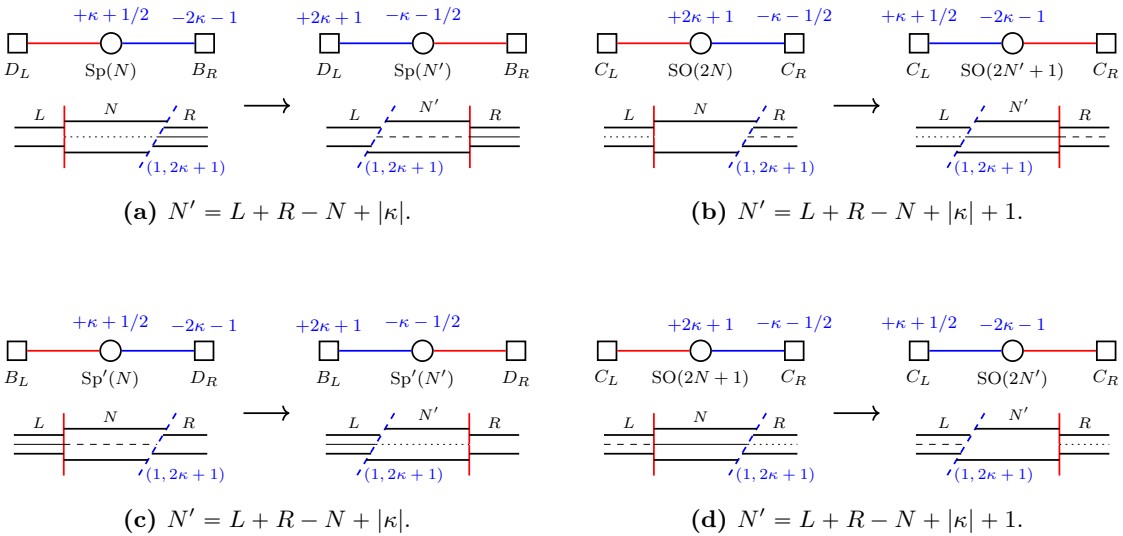

**(a)** $N' = L + R - N + |\kappa|$.

**(b)** $N' = L + R - N + |\kappa| + 1$.

**(c)** $N' = L + R - N + |\kappa|$.

**(d)** $N' = L + R - N + |\kappa| + 1$.

**Figure 4:** The basic OSp CSM dualities for the NS5 $-(1, 2\kappa + 1) -$ D3 $\mathcal{N} = 4$ brane system with the four possible combinations of O3 planes.

### 3.3.1 Linear 2-node quivers

Consider a quiver with two gauge nodes, one orthogonal and one symplectic, equipped with a CS-level. In the following, the case in which the two nodes have the same rank is considered. The four possible such 2-node CSM quivers are summarised in Figure 5. The generalisation to different ranks is obtained analogously and is discussed in Section 4.

**Sp-type duality.** Consider the $\mathrm{SO}_{+q} \times \mathrm{Sp}_{-\frac{q}{2}}$ CSM quivers labelled (0) in Figure 5. Moving the central $(1, q)$ 5-brane through the left NS5 leads to a dual $\mathrm{SO} \times \mathrm{Sp}_{+\frac{q}{2}}$ CSM theory (denoted (1)) that is obtained by applying locally[1] the Sp-type duality (3.1a). In contrast to the single gauge group case, the 2-node CSM quivers do have $\mathbb{Z}_2^{\mathcal{M}}$ and $\mathbb{Z}_2^{\mathcal{C}}$ that may be affected non-trivially by the duality.

Based on explicit index calculations for these dualities (see Table 4 in Appendix B.1), one finds the following fugacity map between (0) and (1):

$$\chi_1 = \chi_0 \,, \qquad \text{and} \qquad \zeta_1 = \zeta_0 \chi_0 \,, \tag{3.2}$$

wherein the subscripts indicate the theory to which the fugacities belong. As a consequence, the global forms of the orthogonal nodes are mapped non-trivially

$$\mathrm{SO} \leftrightarrow \mathrm{SO} \,, \quad \mathrm{O}^{\pm} \leftrightarrow \mathrm{O}^{\mp} \,, \quad \mathrm{Spin} \leftrightarrow \mathrm{Spin} \,, \quad \mathrm{Pin} \leftrightarrow \mathrm{Pin} \,, \tag{3.3}$$

under the duality $(0) \leftrightarrow (1)$.

---

[1]To apply the duality (3.1a) locally on the right gauge node of the theory $\mathrm{SO}_{+q} \times \mathrm{Sp}_{-\frac{q}{2}}$, add trivial $C_0$ and $D_0$ flavour nodes on the sides of the quiver, then freeze the gauge integration:

Now apply the duality (3.1a) on the $\mathrm{Sp}(N)_{-\kappa}$ gauge node and restore the gauge integration:

This corresponds to the $(1, 2\kappa)$ 5-brane passing through the NS5 brane. For more details on the local application of basic dualities to a quiver, see [51].

**SO-type duality.** Returning to the $\mathrm{SO}_{+q} \times \mathrm{Sp}_{-\frac{q}{2}}$ CSM quivers denoted (0) in Figure 5. The central $(1,q)$ 5-brane can also be moved through the right NS5 brane, which leads to a dual $\mathrm{SO}_{-q} \times \mathrm{Sp}$ CSM quiver, labelled by (2). This is equivalent to locally apply an SO-type duality (3.1b). As there are no additional discrete symmetry fugacities, compared to the single gauge group case, the symmetry map is that of [18–20]:

$$\chi_2 = \chi_0 \zeta_0 \,, \qquad \text{and} \qquad \zeta_2 = \zeta_0 \,, \tag{3.4}$$

where the subscripts indicate to which theory the $\mathbb{Z}_2^{\mathcal{M}}$ and $\mathbb{Z}_2^{\mathcal{C}}$ fugacities belong. This leads to the known map of the global forms of the orthogonal nodes

$$\mathrm{SO} \leftrightarrow \mathrm{SO} \,, \quad \mathrm{O}^+ \leftrightarrow \mathrm{O}^+ \,, \quad \mathrm{Spin} \leftrightarrow \mathrm{O}^- \,, \tag{3.5}$$

under the duality $(0) \leftrightarrow (2)$.

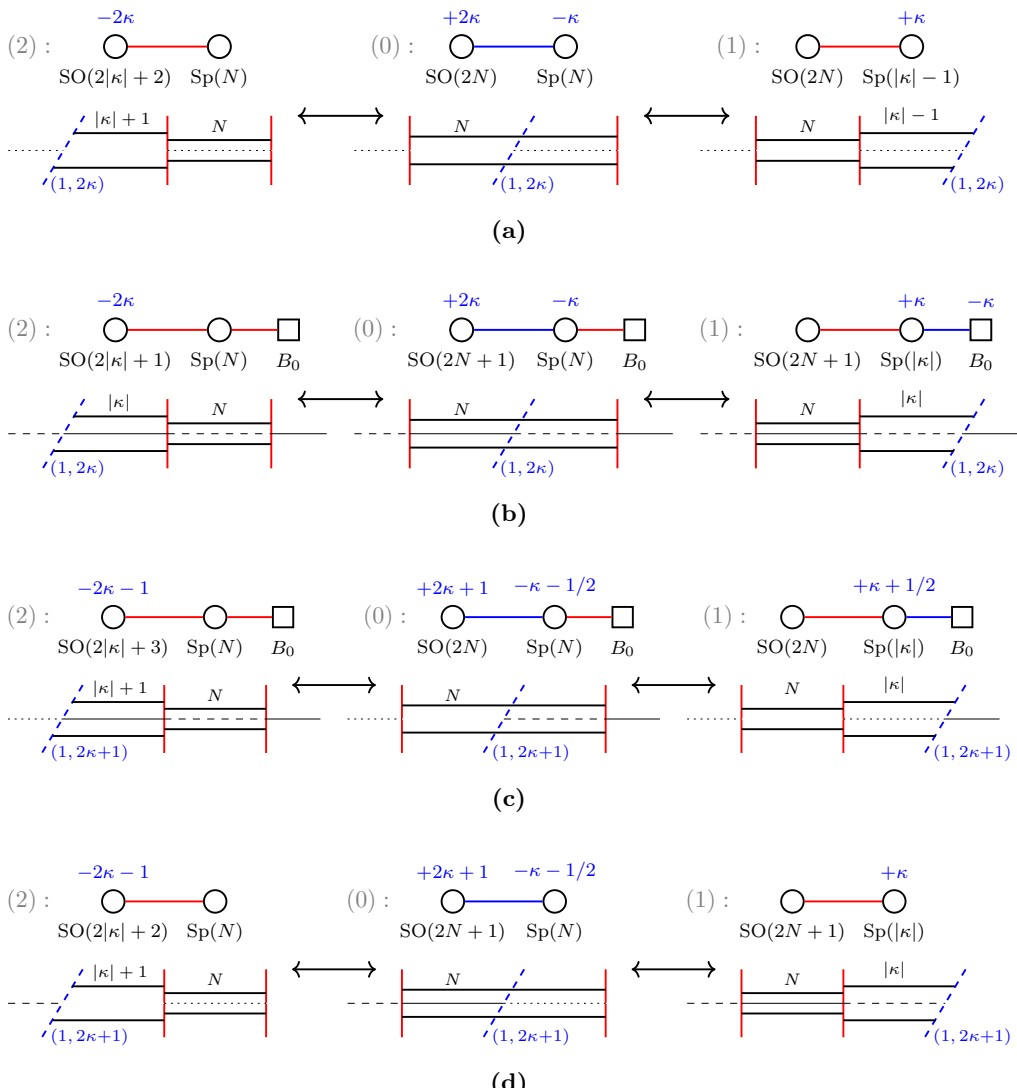

**Figure 5:** The four possible two node CSM theories (labelled (0)) realised from $N$ D3s in between two NS5s and one $(1,q)$ 5-brane in the presence of O3 planes. For all cases, moving the $(1,q)$ 5-brane through the right NS5, one realises the dual CSM theory labelled (1). This is essential a basic Sp-type duality (3.1a). One the other hand, moving the $(1,q)$ 5-brane through the left NS5, the dual CSM theory (2) is reached. Field-theoretically, this is an SO-type duality (3.1b).

### 3.3.2 Linear 3-node quivers

The 2-node quivers have revealed a non-trivial map of the discrete symmetries under an Sp-type duality, even though the single node case did not require any fugacity map. Therefore, the next logic step is to establish the symmetry maps across basic SO/Sp-type dualities in 3 nodes quivers.

**$\mathbf{Sp}(N)_{+\kappa} \times \mathbf{SO}(2N)_{-2\kappa} \times \mathbf{Sp}(N)_{+\kappa}$.** Consider the theory on top of Figure 6a. As shown there, there are three possible duality moves.

1. Applying the duality (3.1a) on the leftmost $\mathrm{Sp}(N)_{+\kappa}$ gauge node, results in the theory (1) of Figure 6a:

$$\tag{3.6}$$

with good condition $|\kappa| \geqslant N$ required on the central node of the dual theory.

The duality map reads

$$\chi_1 = \chi_0 , \qquad \text{and} \qquad \zeta_1 = \zeta_0 \chi_0 , \tag{3.7}$$

which is the same as the 2-node case (3.2). This is consistent and was to be expected, as there are no other symmetries that could alter the required map.

$1'$. Applying the duality (3.1a) on the rightmost $\mathrm{Sp}(N)_{+\kappa}$ gauge node, results on the theory ($1'$) of Figure 6a:

$$\tag{3.8}$$

with good condition $|\kappa| \geqslant N$ required on the central node of the dual theory[2].

Again, the symmetries map across the duality as before (3.2), i.e.

$$\chi_{1'} = \chi_0 , \qquad \text{and} \qquad \zeta_{1'} = \zeta_0 \chi_0 . \tag{3.9}$$

2. Applying the duality (3.1b) on the central $\mathrm{SO}(2N)_{-2\kappa}$ gauge node, leads to theory (2) of Figure 6a

$$\tag{3.10}$$

with good condition $|\kappa| \geqslant N$ required on the first and third nodes of the dual theory.

The required symmetry map turns out to be

$$\chi_2 = \chi_0 \zeta_0 , \qquad \text{and} \qquad \zeta_2 = \zeta_0 , \tag{3.11}$$

in other words, a standard SO-type map, see (3.1b) or (3.4).

Considering the other possible 3-node CSM quivers of the type $\mathrm{Sp}_{k/2} \times \mathrm{SO}_{-k} \times \mathrm{Sp}_{k/2}$ leads to the same conclusions about the required symmetry maps across dualities.

---

[2]The dualities in (3.6) and (3.8) coincide up to twisting of hypermultiplets and orientation (see Figure 6a). Theories (1) and ($1'$) are related by a reflection along the $x_3$ direction combined with $(\mathcal{T})^{2|\kappa|}$, under which NS5 $\to (1, -2\kappa)$ and $(1, 2\kappa) \to$ NS5. Since $2\kappa$ is even, the O3 configuration remains invariant (cf. (2.2b)). This reflects the symmetry of the brane arrangement, with alternating NS5 and $(1, 2\kappa)$ branes in theory (0), leading to isomorphic branches. See Section 4 for details.

$\mathbf{SO(2N)_{+2\kappa} \times Sp(N)_{-\kappa} \times SO(2N)_{+2\kappa}}$. Consider the theory on top of Figure 6b. As shown there, there exist three basic dualities.

1. Applying the duality (3.1a) on the central $Sp(N)_{-\kappa}$ gauge node, yields the theory (1) of Figure 6b:

$$
\begin{array}{c}
\overset{+2\kappa}{\circ} - \overset{-\kappa}{\circ} - \overset{+2\kappa}{\circ} \quad \longleftrightarrow \quad \circ - \overset{+\kappa}{\circ} - \circ \\
\underset{SO(2N)}{\chi_0^{(L)},\zeta_0^{(L)}} \quad Sp(N) \quad \underset{SO(2N)}{\chi_0^{(R)},\zeta_0^{(R)}} \qquad \underset{SO(2N)}{\chi_1^{(L)},\zeta_1^{(L)}} \; Sp(N+|\kappa|-1) \; \underset{SO(2N)}{\chi_1^{(R)},\zeta_1^{(R)}}
\end{array}
\qquad , \qquad (3.12)
$$

with good condition $|\kappa| \geqslant N$ required on the first and last nodes of the dual theory.

Here, there are two sets of discrete $\mathbb{Z}_2^{\mathcal{C}}$ and $\mathbb{Z}_2^{\mathcal{M}}$ fugacities. Several index computations (see Table 5 in Appendix B.1) suggest the following map

$$
\chi_1^{(L)} = \chi_0^{(L)} \;, \quad \zeta_1^{(L)} = \zeta_0^{(L)}\chi_0^{(L)}\chi_0^{(R)} \;, \quad \chi_1^{(R)} = \chi_0^{(R)} \;, \quad \zeta_1^{(R)} = \zeta_0^{(R)}\chi_0^{(R)}\chi_0^{(L)} \;. \quad (3.13)
$$

This is a new/extended symmetry map for an Sp-type duality in quivers where the Sp-node is connected to two orthogonal nodes. Note that this map reduces consistently to (3.2) if one of the orthogonal nodes is not dynamical.

2. Applying the duality (3.1b) on the leftmost $SO(2N)_{+2\kappa}$ gauge node, results in theory (2) of Figure 6b

$$
\begin{array}{c}
\overset{+2\kappa}{\circ} - \overset{-\kappa}{\circ} - \overset{+2\kappa}{\circ} \quad \longleftrightarrow \quad \overset{-2\kappa}{\circ} - \circ - \overset{+2\kappa}{\circ} \\
\underset{SO(2N)}{\chi_0^{(L)},\zeta_0^{(L)}} \quad Sp(N) \quad \underset{SO(2N)}{\chi_0^{(R)},\zeta_0^{(R)}} \qquad \underset{SO(2(|\kappa|+1))}{\chi_2^{(L)},\zeta_2^{(L)}} \; Sp(N) \; \underset{SO(2N)}{\chi_2^{(R)},\zeta_2^{(R)}}
\end{array}
\qquad , \qquad (3.14)
$$

with good condition $|\kappa| \geqslant N$ required on the central node of the dual theory.

The required symmetry map is

$$
\chi_2^{(L)} = \chi_0^{(L)}\zeta_0^{(L)} \;, \quad \zeta_2^{(L)} = \zeta_0^{(L)} \;, \quad \chi_2^{(R)} = \chi_0^{(R)} \;, \quad \zeta_2^{(R)} = \zeta_0^{(R)} \;, \quad (3.15)
$$

which is nothing but the standard SO-type map for a single node, see (3.1b) or (3.4).

2′. Applying the duality (3.1b) on the rightmost $SO(2N)_{+2\kappa}$ gauge node, leads to the theory (2′) of Figure 6b:

$$
\begin{array}{c}
\overset{+2\kappa}{\circ} - \overset{-\kappa}{\circ} - \overset{+2\kappa}{\circ} \quad \longleftrightarrow \quad \overset{+2\kappa}{\circ} - \circ - \overset{-2\kappa}{\circ} \\
\underset{SO(2N)}{\chi_0^{(L)},\zeta_0^{(L)}} \quad Sp(N) \quad \underset{SO(2N)}{\chi_0^{(R)},\zeta_0^{(R)}} \qquad \underset{SO(2N)}{\chi_{2'}^{(L)},\zeta_{2'}^{(L)}} \; Sp(N) \; \underset{SO(2(|\kappa|+1))}{\chi_{2'}^{(R)},\zeta_{2'}^{(R)}}
\end{array}
\qquad , \qquad (3.16)
$$

with good condition $|\kappa| \geqslant N$ required on the central node of the dual theory[3].

As expected, the fugacity map is simply of SO-type for the rightmost node:

$$
\chi_{2'}^{(L)} = \chi_0^{(L)} \;, \quad \zeta_{2'}^{(L)} = \zeta_0^{(L)} \;, \quad \chi_{2'}^{(R)} = \chi_0^{(R)}\zeta_0^{(R)} \;, \quad \zeta_{2'}^{(R)} = \zeta_0^{(R)} \;. \quad (3.17)
$$

---

[3]The dualities in (3.14) and (3.16) coincide up to hypermultiplet twisting and a CS-level sign (see Figure 6b). Theories (2) and (2′) are related by a reflection along the $x_3$ direction combined with $(\mathcal{T})^{2|\kappa|}$, under which the O3 configuration remains invariant since $2\kappa$ is even (cf. (2.2b)). This again reflects the symmetry of the brane arrangement, leading to isomorphic branches. The reader is referred to Section 4.

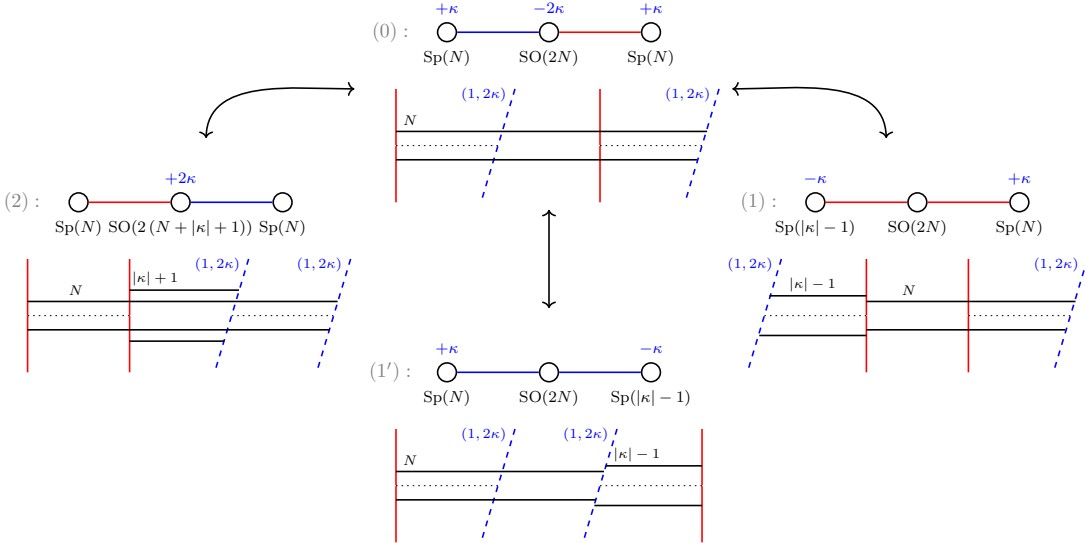

**(a)** The orthosymplectic dualities via brane moves in the $\mathrm{Sp}(N)_\kappa \times \mathrm{SO}(2N)_{-2\kappa} \times \mathrm{Sp}(N)_\kappa$ theory.

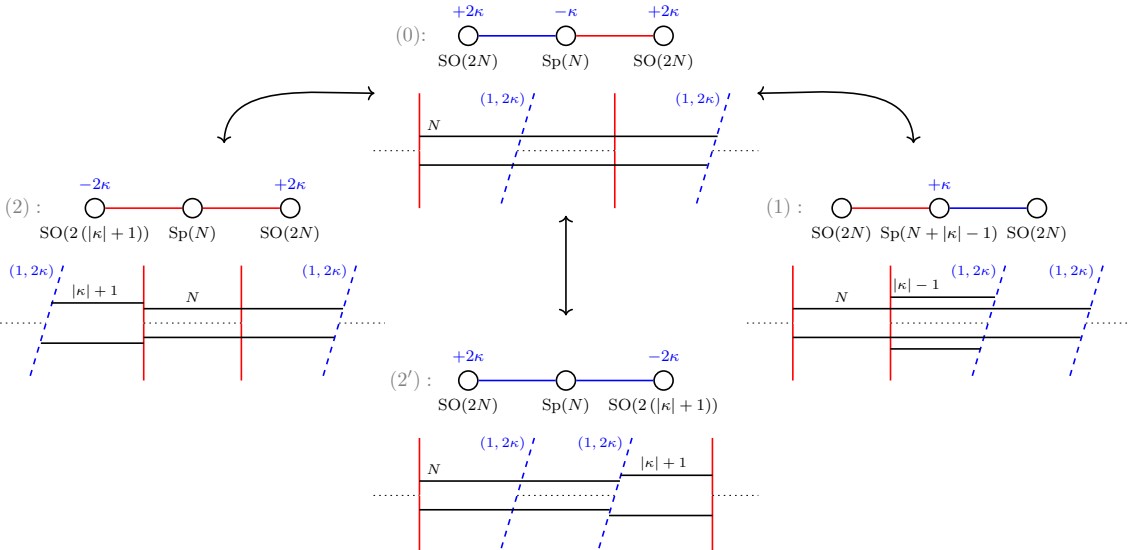

**(b)** The orthosymplectic dualities via brane moves in the $\mathrm{SO}(2N)_{2\kappa} \times \mathrm{Sp}(N)_{-\kappa} \times \mathrm{SO}(2N)_{2\kappa}$ theory.

**Figure 6:** Example 3-node orthosymplectic CSM quivers and their brane realisations. The dual world-volume CSM theories are deduced via brane creation/annihilation moves (see Appendix A.2).

### 3.3.3 General linear quivers

To recollect, starting from the known single node SO/Sp-type dualities in Section 3.2, their extension to two and three node orthosymplectic CSM quivers has been detailed in Sections 3.3.1 and 3.3.2, respectively.

For general linear (or circular) orthosymplectic CSM quivers, one expects that fugacity maps can be derived by an analogous logic. See for instance Appendix A.5 for a sequential application of these basic brane moves (and associated fugacity maps). The treatment of a general fugacity map is not relevant for this work, but would be an interesting extension.

# 4 Magnetic quivers for orthosymplectic CSM theories

In this section, a systematic method for constructing magnetic quivers that capture the NS5-branch and $(1,q)$-branch – $\mathcal{B}_{\mathrm{NS5}}$ and $\mathcal{B}_{(1,q)}$ respectively[4] – of a given orthosymplectic 3d $\mathcal{N} = 4$ $\mathrm{CSM}_q$ theory is introduced and demonstrated for several examples. Firstly, in Section 4.1, the proposal [28] for unitary 3d $\mathcal{N} = 4$ $\mathrm{CSM}_q$ theories is reviewed. Secondly, in Section 4.2 the method is extended to orthosymplectic 3d $\mathcal{N} = 4$ $\mathrm{CSM}_q$ theories; examples for linear quivers of this class are provided in Sections 4.2.1, 4.2.2 and 4.2.3, while in Section 4.2.4 selected examples of circular shape are discussed. Finally, in Section 4.3 predictions are made for the class of orthosymplectic 3d $\mathcal{N} = 3$ $\mathrm{CSM}_q$ theories, both in the Lagrangian and in the non-Lagrangian cases.

## 4.1 Review of magnetic quivers for unitary 3d $\mathcal{N} = 4$ CSM theories

The moduli space of a 3d $\mathcal{N} = 4$ $\mathrm{CSM}_\kappa$ theory possesses two maximal branches – the NS5-branch $\mathcal{B}_{\mathrm{NS5}}$ and the $(1,\kappa)$-branch $\mathcal{B}_{(1,\kappa)}$. For such theories realised in a Type IIB brane system, this corresponds to the movement of D3 branes in intervals of NS5 branes and intervals of $(1,\kappa)$ 5-branes, respectively. For unitary $\mathrm{CSM}_\kappa$ theories, their geometric structure can be captured [28] by a pair of magnetic quivers, $\mathsf{MQ}_{\mathrm{NS5}}$ and $\mathsf{MQ}_{(1,\kappa)}$, namely two non-CS unitary 3d $\mathcal{N} = 4$ theories whose Coulomb branches $\mathcal{C}$ reproduce $\mathcal{B}_{\mathrm{NS5}}$ and $\mathcal{B}_{(1,\kappa)}$ of the starting $\mathrm{CSM}_\kappa$ theory, respectively[5]:

$$\mathcal{B}_{\mathrm{NS5}}\left(\mathrm{CSM}_\kappa\right) \cong \mathcal{C}\left(\mathsf{MQ}_{\mathrm{NS5}}\right), \quad \mathcal{B}_{(1,\kappa)}\left(\mathrm{CSM}_\kappa\right) \cong \mathcal{C}\left(\mathsf{MQ}_{(1,\kappa)}\right). \tag{4.1}$$

The derivation of $\mathsf{MQ}_{\mathrm{NS5}}$ and $\mathsf{MQ}_{(1,\kappa)}$ employs the GK duality in Figure 1 and the generator $\mathcal{T}$ of the $\mathrm{SL}(2,\mathbb{Z})$ duality group, see (2.1b). In particular, to obtain $\mathsf{MQ}_{\mathrm{NS5}}$:

- Use the GK duality (see Figure 1) to bring the NS5 – $(1,\kappa)$ – D3 brane system into a phase defined by having the maximal possible number of D3s suspended between the various intervals of NS5s, i.e. a phase of the brane system in which the maximal number of moduli for $\mathcal{B}_{\mathrm{NS5}}$ are turned on.

- To read off the magnetic quiver $\mathsf{MQ}_{\mathrm{NS5}}$, count the D3s suspended between each NS5 interval. These give rise to $\mathcal{N} = 4$ gauge nodes in the magnetic quiver. Adjacent NS5 intervals give rise to bifundamental hypermultiplet edges between the associated gauge nodes. The amount of D5 charge (provided by each $(1,\kappa)$ 5-brane) in each NS5 interval results in a flavour node for the corresponding gauge node.

For concreteness, consider (assuming $|\kappa| \geqslant L$)

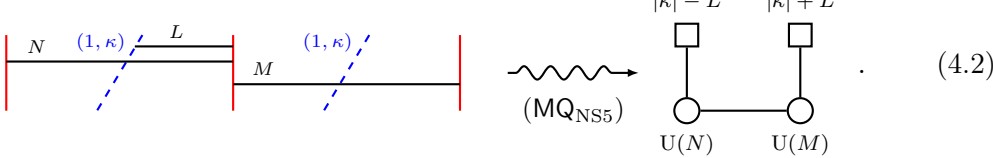

$$\tag{4.2}$$

The stack of $N$-many D3s suspended between the NS5 interval on the left yields a $\mathrm{U}(N)$ gauge node for the magnetic quiver, while the NS5 interval on the right yields a $\mathrm{U}(M)$ gauge node. The $(1,\kappa)$ 5-brane positioned inside the NS5 interval on the left contributes $|\kappa|$-many units of D5 charge to the brane system. The stack of $L$-many frozen D3s suspended between that $(1,\kappa)$ 5-brane and the NS5 brane in the middle

---

[4]For 3d $\mathcal{N} = 4$ $\mathrm{CSM}_q$ theories, in [28] the NS5/$(1,\kappa)$ branches are referred to as the A/B branches.

[5]Whilst this magnetic quivers approach works for any $|\kappa| \geqslant 1$, for $|\kappa| = 1$ one can also employ exact $\mathrm{SL}(2,\mathbb{Z})$ dualities. The $\mathrm{SL}(2,\mathbb{Z})$ dual theories are equivalent to the magnetic quivers $\mathsf{MQ}_{\mathrm{NS5}}$ and $\mathsf{MQ}_{(1,\kappa)}$.

freezes $L$-many units of D5 charge for the first NS5 interval. The unfrozen amount of D5 charge accessible for this NS5 interval counts $(|\kappa| - L)$-many units. Analogously, the NS5 interval on the right is sensitive to $(|\kappa| + L)$-many units of D5 charge[6].

On the other hand, to obtain $\mathsf{MQ}_{(1,\kappa)}$:

- Consider the $(\mathcal{T})^{|\kappa|}$ dual of the $\mathrm{CSM}_\kappa$ theory, yielding another unitary 3d $\mathcal{N} = 4$ $\mathrm{CSM}'_\kappa$ theory, whose maximal branches are swapped with respect to those of the starting $\mathrm{CSM}_\kappa$ theory.

- Apply the procedure to obtain the magnetic quiver $\mathsf{MQ}_{\mathrm{NS5}}$ of $\mathrm{CSM}'_\kappa$, which hence is the $\mathsf{MQ}_{(1,\kappa)}$ of the starting $\mathrm{CSM}_\kappa$ theory.

This strategy is summarised in Figure 7.

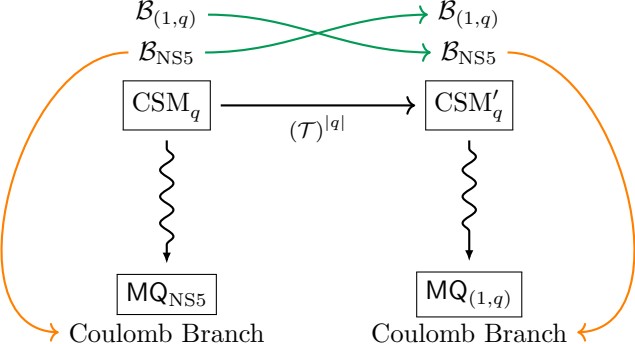

**Figure 7:** The strategy for the magnetic quivers $\mathsf{MQ}_{\mathrm{NS5}}$ and $\mathsf{MQ}_{(1,q)}$ capturing $\mathcal{B}_{\mathrm{NS5}}$ and $\mathcal{B}_{(1,q)}$, respectively, of a 3d $\mathcal{N} = 4$ CSM theory. For unitary theories $q = \kappa$ and for orthosymplectic theories $q = 2\kappa$ or $q = 2\kappa + 1$. The wiggly arrows represent the operation of reading off the $\mathsf{MQ}$.

Once the two magnetic quivers $\mathsf{MQ}_{\mathrm{NS5}}$ and $\mathsf{MQ}_{(1,\kappa)}$ have been derived, one can match their Coulomb branch Hilbert series with the Hilbert series for the $\mathcal{B}_{\mathrm{NS5}}$ and $\mathcal{B}_{(1,\kappa)}$ branches of the $\mathrm{CSM}_\kappa$ theory, obtained as a limit from the index [28, 52] as follows. First redefine inside the index the R-symmetry fugacity $x$ and the axial symmetry fugacity $t$ using two new parameters $a$ and $b$, which account for the $\mathcal{B}_{\mathrm{NS5}}$ (or A-branch) and $\mathcal{B}_{(1,\kappa)}$ (or B-branch) projections:

$$\mathcal{I}[a, b] = \mathcal{I}\left[x = (a\,b)^{\frac{1}{2}}, \, t = (b/a)^{\frac{1}{4}}\right]. \tag{4.3}$$

Then project the index onto the $\mathcal{B}_{\mathrm{NS5}}$ and $\mathcal{B}_{(1,\kappa)}$ branches:

$$\lim_{b \to 0} \mathcal{I}[a, b] = \mathrm{HS}_{\mathrm{NS5}}(\mathrm{CSM})[a], \tag{4.4a}$$

$$\lim_{a \to 0} \mathcal{I}[a, b] = \mathrm{HS}_{(1,\kappa)}(\mathrm{CSM})[b]. \tag{4.4b}$$

Therefore, the Hilbert series comparison reads

$$\mathrm{HS}_{\mathrm{NS5}}(\mathrm{CSM})[a] = \mathrm{HS}_{\mathcal{C}}(\mathsf{MQ}_{\mathrm{NS5}})[a], \tag{4.5a}$$

$$\mathrm{HS}_{(1,\kappa)}(\mathrm{CSM})[b] = \mathrm{HS}_{\mathcal{C}}(\mathsf{MQ}_{(1,\kappa)})[b]. \tag{4.5b}$$

---

[6]These rules are equivalent to replacing each $(1, \kappa)$ brane with $|\kappa|$-many D5 branes and using Hanany – Witten transitions [15] to move the brane system into a phase that captures the magnetic quiver [28]. In view of the extension to orthosymplectic theories, however, the approach presented here is better suited.

**Example.** Consider the $\text{CSM}_\kappa$ theory $\text{U}(N)_\kappa \times \text{U}(M)_{-\kappa}$ on the top left of Figure 8. Without loss of generality, assume $N \geqslant M$, for which the maximal number of D3s that can be suspended between the two NS5s is $M$. There are two distinct phases of the initial brane system that allow for such a configuration, referred to as (1) and (2). They are obtained as follows:

(1) Using GK duality (see Figure 1), move the $(1, \kappa)$ 5-brane through the NS5 brane on the left. The result is the brane system (1) in Figure 8, where $d := N - M$.

(2) Reconnect the D3s such that there is a stack of $M$-many D3s suspended between the two NS5s and a stack of $d$-many D3s suspended between the NS5 brane on the left and the $(1, \kappa)$ 5-brane. The result is the brane system (2) in Figure 8.

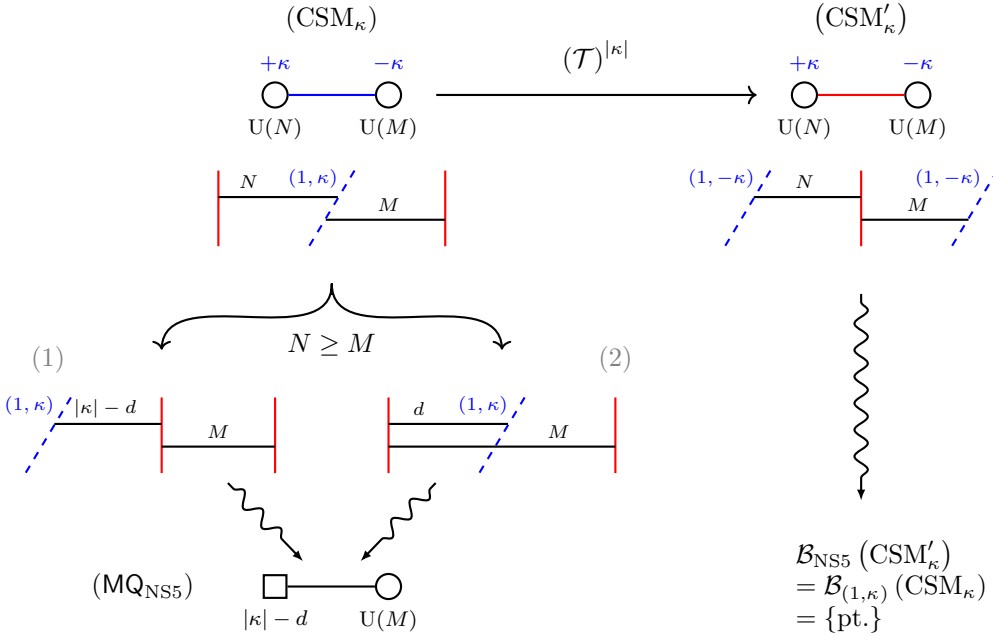

**Figure 8:** Capturing $\mathcal{B}_{\text{NS5}}$ and $\mathcal{B}_{(1,\kappa)}$ of the unitary 3d $\mathcal{N} = 4$ $\text{CSM}_\kappa$ theory $\text{U}(N)_\kappa \times \text{U}(M)_{-\kappa}$ with the magnetic quiver $\text{MQ}_{\text{NS5}}$, where $d := N - M \geqslant 0$. For the starting brane system to be a supersymmetric configuration, assume $|\kappa| \geqslant N, M$. The case $N \leqslant M$ works analogously. The wiggly arrows represent the operation of reading off the MQ.

Reading off[7] the magnetic quivers as described above, these two distinct phases of the same brane system yield the same $\text{MQ}_{\text{NS5}}$ (see on the bottom of Figure 8).

In a similar fashion, one can construct a magnetic quiver $\text{MQ}_{(1,\kappa)}$ that captures $\mathcal{B}_{(1,\kappa)}$ of the starting $\text{CSM}_\kappa$ theory. However, for the example considered here, $\mathcal{B}_{(1,\kappa)}$ is trivial, as can be seen from the $(\mathcal{T})^{|\kappa|}$ dual $\text{CSM}'_\kappa$ in the right part of Figure 8, where

$$\text{NS5} \xrightarrow{(\mathcal{T})^{|\kappa|}} (1, -\kappa), \qquad (1, \kappa) \xrightarrow{(\mathcal{T})^{|\kappa|}} \text{NS5}, \tag{4.6}$$

has been used (see (2.1b)). Finally, the goodness of the starting $\text{CSM}_\kappa$ is provided by the goodness condition of the magnetic quivers $\text{MQ}_{\text{NS5}}$ and $\text{MQ}_{(1,\kappa)}$, here $|\kappa| \geqslant N + M$.

The above framework is applicable to the class of 3d $\mathcal{N} = 3$ $\text{CSM}_\kappa$ theories, where one has more than two maximal branches [28].

---

[7]Another perspective is the operators map between the frames (0), (1), and (2), as illustrated in Appendix A.4. This shows that the magnetic quiver, by construction, encodes the maximal branch operators.

## 4.2 Magnetic quivers for orthosymplectic 3d $\mathcal{N} = 4$ CSM theories

In the following, the magnetic quiver method is extended to a class of orthosymplectic CSM quivers that are realised in certain Type IIB brane configurations. Firstly, the setup is defined; secondly, the method is spelled out.

**The setup.** To apply the magnetic quiver method to probe a maximal branch of a CSM brane system, the class of admissible brane setups must be suitably restricted. For concreteness, focus on the NS5 branch. The required suitable phase is defined by the following conditions:

1. All non-NS5 (i.e., $(p, q)$) 5-branes can be moved between NS5 intervals such that:

   - The maximal number of D3-branes are free to move along NS5s.
   - All other D3s are frozen between an NS5 and a $(p, q)$ 5-brane, obeying the generalized s-rule (cf. Appendix A.2).
   - No D3 segments are allowed to move freely along $(p, q)$ 5-branes; otherwise, the configuration would not represent a maximal branch phase.

2. Each $(p, q)$ 5-brane is connected to at most one NS5 via frozen D3-branes. This minimises the number of frozen D3s and ensures that each $(p, q)$ 5-brane can only undergo brane creation/annihilation with the NS5 it is connected to.

   This phase closely resembles the standard Coulomb branch phase of a D3–D5–NS5 system, extended to include $(p, q)$ 5-branes.

3. Using the residual freedom of brane creation/annihilation as in (2), one must be able to reach a phase in which each NS5 interval contains a single, well-defined O3 plane, provided all $(p, q)$ 5-branes in the interval are coincident.

**Magnetic quiver derivation.** Next, the method for constructing magnetic quivers capturing $\mathcal{B}_{\mathrm{NS5}}$ and $\mathcal{B}_{(1,2\kappa)}$ is extended to orthosymplectic 3d $\mathcal{N} = 4$ $\mathrm{CSM}_{2\kappa}$ theories. For convenience, the CS-level is assumed to be even. Nonetheless, the discussion can be generalised to odd CS-level. In order to realise such models as world-volume theories in a Type IIB string theory brane system, one includes O3 planes. In analogy to the unitary case, the derivation of $\mathsf{MQ}_{\mathrm{NS5}}$ and $\mathsf{MQ}_{(1,2\kappa)}$ employs the orthosymplectic dualities in Figure 3 and the $\mathcal{T}$ generator of $\mathrm{SL}(2, \mathbb{Z})$, see (2.1b). In particular, to obtain $\mathsf{MQ}_{\mathrm{NS5}}$:

- Use the orthosymplectic dualities (see Figure 3) to bring the NS5 – $(1, 2\kappa)$ – D3 brane system into a phase defined by having the maximal possible number of D3s suspended between the various intervals of NS5s, i.e. the phase of the brane system in which the maximal number of moduli for $\mathcal{B}_{\mathrm{NS5}}$ are turned on.

- To read off the magnetic quiver $\mathsf{MQ}_{\mathrm{NS5}}$, count the D3s suspended between each NS5 interval. These give rise to $\mathcal{N} = 4$ gauge nodes in the magnetic quiver, with the gauge algebra defined by the O3 plane (see Tables 1 and 3) spanned between the NS5 interval. Adjacent NS5 intervals give rise to $\mathcal{N} = 4$ half-hypermultiplet connecting the associated gauge nodes. Each $(1, 2\kappa)$ 5-brane contributes D5 charge to the NS5 intervals, resulting in flavour nodes for the corresponding gauge nodes. The flavour algebra is determined by the type of O3 plane spanned between the 5-branes.

For example, consider (assuming $|\kappa| + 1 \geqslant L$)

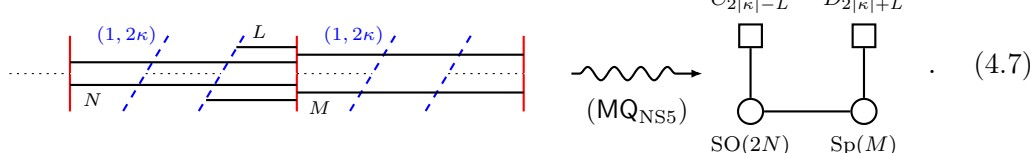

$$. \quad (4.7)$$

The stack of $N$-many D3s suspended inside the NS5 interval on the left is mirrored across an O3$^-$ plane, yielding a SO$(2N)$ gauge node for the magnetic quiver. For this to work, it is necessary to have an even amount of half D5 charge positioned inside the NS5 interval. Analogously, the stack of $M$-many D3s suspended between the second NS5 interval and mirrored across an O3$^+$ plane yields a Sp$(M)$ gauge node for the magnetic quiver.

The two $(1, 2\kappa)$ 5-branes positioned inside the NS5 interval on the left contribute $2|\kappa|$-many units of D5 charge to the brane system. The stack of $L$-many D3s suspended between the second $(1, 2\kappa)$ 5-brane and the NS5 brane in the middle freezes $L$-many units of D5 charge for the first NS5 interval. In total, the unfrozen amount of D5 charge accessible for this NS5 interval counts $(2|\kappa| - L)$-many units and yields a $C_{2|\kappa|-L}$ flavour node for the magnetic quiver. Analogously, the NS5 interval on the right is sensitive to a total amount of $(2|\kappa| + L)$-many units of D5 charge, mirrored across a O3$^-$ plane, yielding a $D_{2|\kappa|+L}$ flavour node for the magnetic quiver.

The derivation procedure holds for any O3 planes inserted in the brane system (4.7). For the brane system to be a supersymmetric configuration, in the presence of a O3$^\pm$ mirroring the stack of $L$-many D3s one requires $|\kappa| \mp 1 \geqslant L$, while in the case of $\widetilde{O3}^\pm$ one has to assume $|\kappa| \geqslant L$. Furthermore, when semi-infinite $\widetilde{O3}^-$ planes connected to an NS5 interval are present, there are additional stuck D5 half-branes that contribute to the unfrozen D5 charge.

On the other hand, to obtain $\mathsf{MQ}_{(1,2\kappa)}$:

- Consider the $(\mathcal{T})^{2|\kappa|}$ dual of the CSM$_{2\kappa}$ theory, resulting in another 3d $\mathcal{N} = 4$ CSM$'_{2\kappa}$ theory whose maximal branches are swapped with respect to those of the starting CSM$_{2\kappa}$ theory.

- Apply the procedure to obtain the magnetic quiver $\mathsf{MQ}_{NS5}$ of CSM$'_{2\kappa}$, which corresponds to the $\mathsf{MQ}_{(1,\kappa)}$ of the starting CSM$_{2\kappa}$ theory.

Again, this strategy is summarised in Figure 7. In the rest of this section, this framework is demonstrated for two examples, differing in the type of O3 planes that are spanned throughout the brane systems.

### 4.2.1 Linear examples with 2 nodes

In the following, the magnetic quivers construction is demonstrated on linear orthosymplectic 3d $\mathcal{N} = 4$ CSM$_{2\kappa}$ theories with 2 gauge nodes.

**Example with O3$^\pm$ planes.** Consider the CSM$_{2\kappa}$ theory SO$(2N)_{2\kappa} \times$ Sp$(M)_{-\kappa}$ on the top left of Figure 9, where only O3$^\pm$ orientifolds are present in the brane system. Depending on the number of D3s suspended between the 5-branes, there are different phases of the initial brane system that capture the maximum amount of possible $\mathcal{B}_{NS5}$ moduli. One has two possibilities:

(1) Assume $N \geqslant M$, for which the maximal number of D3s that can be suspended between the two NS5s is $M$. Using the orthosymplectic duality shown in Figure 3b, move the $(1, 2\kappa)$ 5-brane through the NS5 brane on the left. The result is the brane system (1) in Figure 9, giving rise to the magnetic quiver $\mathsf{MQ}_{\mathrm{NS5}}^{\mathrm{Sp}}$, which consists of a $\mathrm{Sp}(M)$ gauge node and a $D_{|\kappa|+1-d}$ flavour node, where $d \coloneqq N - M$.

(2) Alternatively, assume $N \leqslant M$, i.e. the maximal number of D3 moduli between the two NS5s being $N$. Using orthosymplectic duality (see Figure 3a), move the $(1, 2\kappa)$ 5-brane through the NS5 brane on the right. The result is the brane system (2) in Figure 9, associated with the magnetic quiver $\mathsf{MQ}_{\mathrm{NS5}}^{\mathrm{SO}}$, consisting of a $\mathrm{SO}(N)$ gauge node and a $C_{|\kappa|-1-\ell}$ flavour node, where $\ell \coloneqq M - N$.

When $N = M$, the two above brane systems are both viable and thus one get two valid magnetic quivers[8] with identical Coulomb branch Hilbert series, see Figure 10.

In contrast to unitary theories, extending the framework introduces additional subtleties. The $\mathsf{MQ}_{\mathrm{NS5}}^{\mathrm{Sp}}$ and $\mathsf{MQ}_{\mathrm{NS5}}^{\mathrm{SO}}$ magnetic quivers capture different global gauge forms of the $\mathrm{CSM}_{2\kappa}$ theory. In particular, based on the map of the topological symmetry $\mathbb{Z}_2^{\mathcal{M}}$ (with fugacity $\zeta$) and charge conjugation symmetry $\mathbb{Z}_2^{\mathcal{C}}$ (with fugacity $\chi$) across the orthosymplectic dualities as discussed throughout Section 3.2 (see (3.1b)), one finds that the Hilbert series $\mathrm{HS}_{\mathrm{NS5}}(\mathrm{CSM}_{2\kappa})\left[a; \chi, \zeta\right]$ for the $\mathrm{CSM}_{2\kappa}$ theory and the Coulomb branch Hilbert series $\mathrm{HS}_{\mathcal{C}}(\mathsf{MQ}_{\mathrm{NS5}}^{\mathrm{SO}})\left[a; \widetilde{\chi}, \widetilde{\zeta}\right]$ and $\mathrm{HS}_{\mathcal{C}}(\mathsf{MQ}_{\mathrm{NS5}}^{\mathrm{Sp}})\left[a\right]$ of the magnetic quivers match as follows:

$$\mathrm{HS}_{\mathrm{NS5}}(\mathrm{CSM}_{2\kappa})\left[a; \chi, \zeta\right] = \mathrm{HS}_{\mathcal{C}}(\mathsf{MQ}_{\mathrm{NS5}}^{\mathrm{SO}})\left[a; \widetilde{\chi} = \chi, \widetilde{\zeta} = \zeta\chi\right], \qquad (4.8a)$$

$$\mathrm{HS}_{\mathrm{NS5}}([\mathrm{CSM}_{2\kappa}]^{\mathrm{O}^-})\left[a; \zeta = 1\right] = \mathrm{HS}_{\mathcal{C}}(\mathsf{MQ}_{\mathrm{NS5}}^{\mathrm{Sp}})\left[a\right], \qquad (4.8b)$$

where $[\mathrm{CSM}_{2\kappa}]^{\mathrm{O}^-}$ indicates the $\mathrm{O}^-$ global form of $\mathrm{CSM}_{2\kappa}$, and $\mathrm{HS}_{\mathrm{NS5}}([\mathrm{CSM}_{2\kappa}]^{\mathrm{O}^-})\left[a; \zeta\right]$ is obtained by applying (2.3b) to the index of the $\mathrm{CSM}_{2\kappa}$ theory. As a consequence, for the magnetic quiver $\mathsf{MQ}_{\mathrm{NS5}}^{\mathrm{SO}}$ one can match the different global forms as follows:

$$\mathrm{HS}_{\mathrm{NS5}}([\mathrm{CSM}_{2\kappa}]^{\mathrm{O}^{\pm}})\left[a; \zeta\right] = \mathrm{HS}_{\mathcal{C}}([\mathsf{MQ}_{\mathrm{NS5}}^{\mathrm{SO}}]^{\mathrm{O}^{\mp}})\left[a; \widetilde{\zeta} = \zeta\right], \qquad (4.9a)$$

$$\mathrm{HS}_{\mathrm{NS5}}([\mathrm{CSM}_{2\kappa}]^{\mathrm{Spin}})\left[a; \chi\right] = \mathrm{HS}_{\mathcal{C}}([\mathsf{MQ}_{\mathrm{NS5}}^{\mathrm{SO}}]^{\mathrm{Spin}})\left[a; \widetilde{\chi} = \chi\right], \qquad (4.9b)$$

$$\mathrm{HS}_{\mathrm{NS5}}([\mathrm{CSM}_{2\kappa}]^{\mathrm{Pin}})\left[a\right] = \mathrm{HS}_{\mathcal{C}}([\mathsf{MQ}_{\mathrm{NS5}}^{\mathrm{SO}}]^{\mathrm{Pin}})\left[a\right], \qquad (4.9c)$$

where, for example, $[\mathsf{MQ}_{\mathrm{NS5}}^{\mathrm{SO}}]^{\mathrm{O}^{\pm}}$ indicates the $\mathrm{O}^{\pm}$ global form of $\mathsf{MQ}_{\mathrm{NS5}}^{\mathrm{SO}}$. Note that, consistently with the $N = M$ case, equations (4.8b) and (4.9a) (with $\zeta = 1$) are reflected by the Hilbert series equalities in Figure 10. Concretely, the differing behaviour for the two magnetic quivers is due to $\mathsf{MQ}_{\mathrm{NS5}}^{\mathrm{SO}}$ having a pair of discrete symmetries for which its Coulomb branch is sensitive to, i.e. the topological symmetry $\mathbb{Z}_2^{\mathcal{M}}$ counted with fugacity $\widetilde{\zeta}$ and the charge conjugation symmetry $\mathbb{Z}_2^{\mathcal{C}}$ counted with fugacity $\widetilde{\chi}$, whilst $\mathsf{MQ}_{\mathrm{NS5}}^{\mathrm{Sp}}$ has no such discrete symmetries.

One can also construct a magnetic quiver that captures $\mathcal{B}_{(1,2\kappa)}$ of the orthosymplectic 3d $\mathcal{N} = 4$ $\mathrm{CSM}_{2\kappa}$. The procedure requires to $(\mathcal{T})^{2|\kappa|}$ dualise the $\mathrm{CSM}_{2\kappa}$ theory, using

$$\mathrm{NS5} \xrightarrow{(\mathcal{T})^{2|\kappa|}} (1, -2\kappa), \qquad (1, 2\kappa) \xrightarrow{(\mathcal{T})^{2|\kappa|}} \mathrm{NS5}. \qquad (4.10)$$

The O3 planes configuration is left invariant, since $2\kappa$ is even (recall the transformation rules (2.2b) of O3 planes under $\mathcal{T}$). The result is a $\mathrm{CSM}_{2\kappa}'$ theory for which $\mathcal{B}_{\mathrm{NS5}}$ and

---

[8]Another perspective comes from the operator map between the initial system and frames (1) and (2), as shown in Appendix A.4. By construction, the magnetic quivers encode the maximal branch operators.

$\mathcal{B}_{(1,2\kappa)}$ are swapped with respect to those of the starting model. Then, by constructing the magnetic quiver $\mathsf{MQ}_{NS5}$ for $CSM'_{2\kappa}$, one automatically obtains the magnetic quiver $\mathsf{MQ}_{(1,2\kappa)}$ of $CSM_{2\kappa}$. For the chosen example, however, $\mathcal{B}_{(1,2\kappa)}$ is trivial, as depicted on the right part of Figure 9.

In conclusion, depending on the ranks $N$ and $M$, for $\mathcal{B}_{NS5}$ one has two possible magnetic quivers, $\mathsf{MQ}_{NS5}^{Sp}$ and $\mathsf{MQ}_{NS5}^{SO}$, which are good theories provided that the condition

$$|\kappa| \geqslant N + M \tag{4.11}$$

is met. Furthermore, since $\mathcal{B}_{(1,2\kappa)}$ is trivial in the example, no further goodness constraint is imposed and the starting $CSM_{2\kappa}$ theory is good when (4.11) is satisfied.

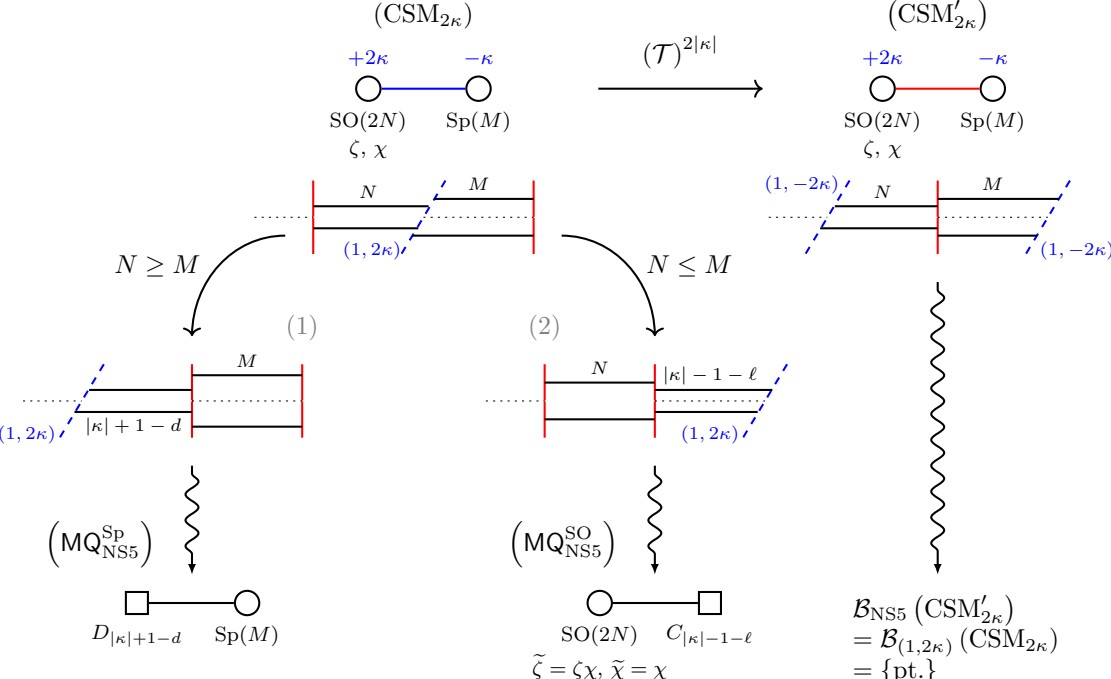

**Figure 9:** Capturing $\mathcal{B}_{NS5}$ and $\mathcal{B}_{(1,2\kappa)}$ of the orthosymplectic 3d $\mathcal{N} = 4$ $CSM_{2\kappa}$ theory $SO(2N)_{2\kappa} \times Sp(M)_{-\kappa}$ with magnetic quivers $\mathsf{MQ}_{NS5}^{Sp}$ and $\mathsf{MQ}_{NS5}^{SO}$, where $d := N - M \geqslant 0$ (for $N \geqslant M$) and $\ell := M - N \geqslant 0$ (for $N \leqslant M$). The $(1, 2\kappa)$ branch of the $CSM_{2\kappa}$ is trivial. The wiggly arrows represent the operation of reading off the $\mathsf{MQ}$.

$$\mathrm{HS}_\mathcal{C}\left( \begin{smallmatrix} C_M \\ \square \\ | \\ \bigcirc \\ O(2N)^+ \end{smallmatrix} \right) = \mathrm{HS}_\mathcal{C}\left( \begin{smallmatrix} C_{M+1} \\ \square \\ | \\ \bigcirc \\ O(2N+1)^+ \end{smallmatrix} \right) = \mathrm{HS}_\mathcal{C}\left( \begin{smallmatrix} C_{M+1} \\ \square \\ | \\ \bigcirc \\ SO(2N+1)^+ \end{smallmatrix} \right) = \mathrm{HS}_\mathcal{C}\left( \begin{smallmatrix} D_{M+2} \\ \square \\ | \\ \bigcirc \\ Sp(N) \end{smallmatrix} \right)$$

**Figure 10:** Equality of unrefined Hilbert series for quivers consisting of symplectic and orthogonal nodes, based on comparison of conformal dimension of the monopole operators, the Weyl group and dressing factors [43].

The discussion here presented for generic ranks and CS-level is accompanied by calculations for explicit values of $N$, $M$ and $\kappa$, see Table 6 in Appendix B.2. In the table, below each index one finds the Hilbert series for the $\mathcal{B}_{NS5}$ and $\mathcal{B}_{(1,2\kappa)}$ branches, which are obtained as a limit from the index, see (4.3) and (4.4). As $\mathrm{HS}_{\mathcal{B}_{NS5}}$ for this example is

trivial, one only has to compare the NS5 branch Hilbert series of the CSM theory with the Coulomb branch Hilbert series of the magnetic quivers $\mathsf{MQ}^{\mathrm{Sp}}_{\mathrm{NS5}}$ and $\mathsf{MQ}^{\mathrm{SO}}_{\mathrm{NS5}}$, provided that the fugacity mapping (4.8) is implemented.

**Example with $\widetilde{\mathrm{O3}}^{\pm}$ planes.** For a second example of a $\mathrm{CSM}_{2\kappa}$ theory with 2 nodes, consider the brane system on the top left of Figure 11, where $\widetilde{\mathrm{O3}}^{\pm}$ planes are included. As in the previous example, there are two possibilities to capture the maximum amount of possible $\mathcal{B}_{\mathrm{NS5}}$ moduli:

(1) Assume $N \geqslant M$, for which the maximal number of D3s that can be suspended between the two NS5s is $M$. After moving the $(1, 2\kappa)$ 5-brane through the NS5 brane on the left (see Figure 3d). The initial brane system changes into system (1) in Figure 11, where $d := N - M$. The associated magnetic quiver $\mathsf{MQ}^{\mathrm{Sp}}_{\mathrm{NS5}}$ consists of a $\mathrm{Sp}(M)$ gauge node and a $D_{|\kappa|+1-d}$ flavour node, where the semi-infinite $\widetilde{\mathrm{O3}}^{-}$ planes are accounted for, giving an additional flavour.

(2) On the other hand, for $N \leqslant M$, the maximal number of D3 moduli between the two NS5s is $N$. Move the $(1, 2\kappa)$ 5-brane through the NS5 brane on the right (Figure 3c), resulting in the brane system (2) in Figure 11, where $\ell := M - N$,. The magnetic quiver $\mathsf{MQ}^{\mathrm{SO}}_{\mathrm{NS5}}$ consists of a $\mathrm{SO}(N)$ gauge node and a $C_{|\kappa|-\ell}$ flavour node.

As before, when $N = M$, the two above brane systems are both viable and give rise to two magnetic quivers with identical Coulomb branch Hilbert series, see Figure 10.

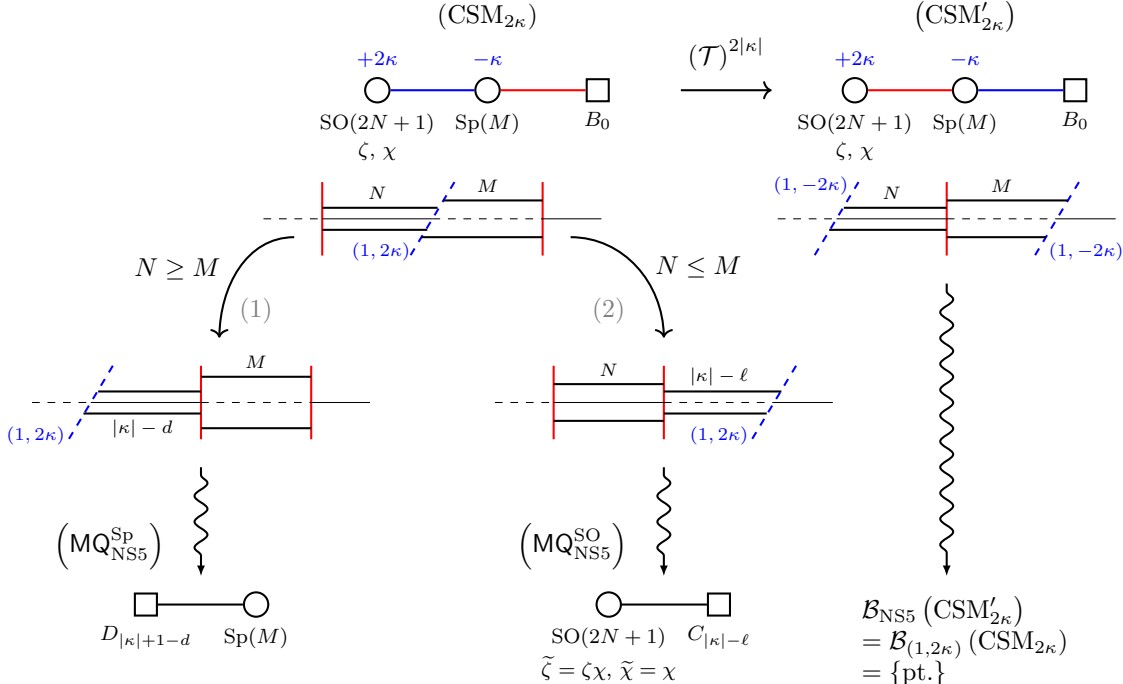

**Figure 11:** Capturing $\mathcal{B}_{\mathrm{NS5}}$ and $\mathcal{B}_{(1,2\kappa)}$ of the orthosymplectic 3d $\mathcal{N} = 4$ $\mathrm{CSM}_{2\kappa}$ theory $\mathrm{SO}(2N + 1)_{2\kappa} \times \mathrm{Sp}(M)_{-\kappa}$ with magnetic quivers $\mathsf{MQ}^{\mathrm{Sp}}_{\mathrm{NS5}}$ and $\mathsf{MQ}^{\mathrm{SO}}_{\mathrm{NS5}}$, where $d := N - M$ (for $N \geqslant M$) and $\ell := M - N \geqslant 0$ (for $N \leqslant M$). The $(1, 2\kappa)$ branch of the $\mathrm{CSM}_{2\kappa}$ theory is trivial. The wiggly arrows represent the operation of reading off the $\mathsf{MQ}$.

Analogous to the previous example, taking the mapping of the discrete symmetries across the orthosymplectic dualities (see Section 3) into account, the Hilbert series for

$\mathcal{B}_{\mathrm{NS5}}$ of the OSp $\mathrm{CSM}_{2\kappa}$ theory and for the Coulomb branches of the two different magnetic quivers satisfies (4.8) and (4.9). The goodness condition for both magnetic quivers reads $|\kappa| \geqslant N + M$.

As in the previous example, note that $\mathcal{B}_{(1,2\kappa)}$ for $\mathrm{CSM}_\kappa$ is trivial, as depicted on the right of Figure 11. Therefore, the $\mathrm{CSM}_{2\kappa}$ theory has the same goodness condition as the magnetic quivers $\mathsf{MQ}^{\mathrm{SO}}_{\mathrm{NS5}}$ and $\mathsf{MQ}^{\mathrm{Sp}}_{\mathrm{NS5}}$.

Again, the discussion here presented for generic ranks and CS-level is accompanied by calculations for explicit values of $N$, $M$ and $\kappa$, see Table 7 in Appendix B.2. In that table is presented the index for the linear $\mathrm{SO}(2N + 1)_{+2\kappa} \times \mathrm{Sp}(N)_{-\kappa}$ CSM theory, together with the Hilbert series for the $\mathcal{B}_{\mathrm{NS5}}$ and $\mathcal{B}_{(1,2\kappa)}$ branches, which are obtained as a limit from the index as explained in (4.3) and (4.4). The $\mathrm{HS}_{\mathcal{B}_{\mathrm{NS5}}}$ for this example is trivial, therefore, one only has to compare the NS5 branch Hilbert series of the CSM theory with the Coulomb branch Hilbert series of the magnetic quivers $\mathsf{MQ}^{\mathrm{Sp}}_{\mathrm{NS5}}$ and $\mathsf{MQ}^{\mathrm{SO}}_{\mathrm{NS5}}$, provided that the fugacity mapping (4.8) is implemented.

### 4.2.2 Linear examples with 3 nodes

In this section, the magnetic quiver construction is applied to several example $\mathrm{CSM}_{2\kappa}$ theories with three gauge nodes. In the process, both the construction and the associated fugacity maps are discussed in detail.

$\mathbf{SO(2N)_{+2\kappa} \times Sp(M)_{-\kappa} \times SO(2N)_{+2\kappa}}$. As a first example of a 3 node quiver, consider the orthosymplectic 3d $\mathcal{N} = 4$ $\mathrm{CSM}_{2\kappa}$ theory depicted on the top left of Figure 12. Depending on the number of D3 branes suspended between the various 5-branes, there are different phases of the brane system that capture the maximal branch $\mathcal{B}_{\mathrm{NS5}}$:

(1) If $N \leqslant M$, use the orthosymplectic duality in Figure 3a to move the $(1, 2\kappa)$ 5-brane on the left through the NS5 brane on the right, resulting in a stack of $(|\kappa| - 1 - \ell + N)$-many D3s (with $\ell \coloneqq M - N$) suspended between the two 5-branes. In order to reach a configuration that is supersymmetric and captures the maximal amount of turned-on moduli for $\mathcal{B}_{\mathrm{NS5}}$, reconnect the D3s such that a stack of $(|\kappa| - 1 - \ell)$-many D3s is suspended between the NS5 brane on the right and the $(1, 2\kappa)$ 5-brane that has been moved through. The two $(1, 2\kappa)$ 5-branes to the right of the NS5 interval can be stacked on top of each other, such that the $(|\kappa| - 1 - \ell)$-many units of unfrozen D5 charge are mirrored across a O3$^+$ plane. The result is the brane setup labelled (1) in Figure 12.

(2) If $N \geqslant M$, one can reach the brane system (2) in Figure 12 analogously to what has been shown for the examples of 2-nodes quivers in Section 4.2.

(3) If $N = M$, move the $(1, 2\kappa)$ 5-brane on the right through the NS5 brane on the right (see Figure 3b), resulting in a stack of $(|\kappa| + 1)$-many D3s suspended between those two 5-branes. For a stable configuration, reconnect the D3s into a stack of $(|\kappa| + 1 - N)$-many D3s suspended between the NS5 5-brane on the right and the $(1, 2\kappa)$ 5-brane on the right and a stack of $N$-many D3s suspended between the two NS5s, see brane system (3) in Figure 12. For the magnetic quiver the flavour count is $2|\kappa| - (|\kappa| + 1 - N) = |\kappa| - 1 + N$.

Different to the examples considered so far, note that $\mathcal{B}_{(1,2\kappa)}$ for the starting $\mathrm{CSM}_{2\kappa}$ is non-trivial. Explicitly, as can be directly followed from the $(\mathcal{T})^{2|\kappa|}$ dual (see the right half of Figure 12), the two maximal branches are isomorphic, i.e. $\mathcal{B}_{\mathrm{NS5}} \cong \mathcal{B}_{(1,2\kappa)}$. Both

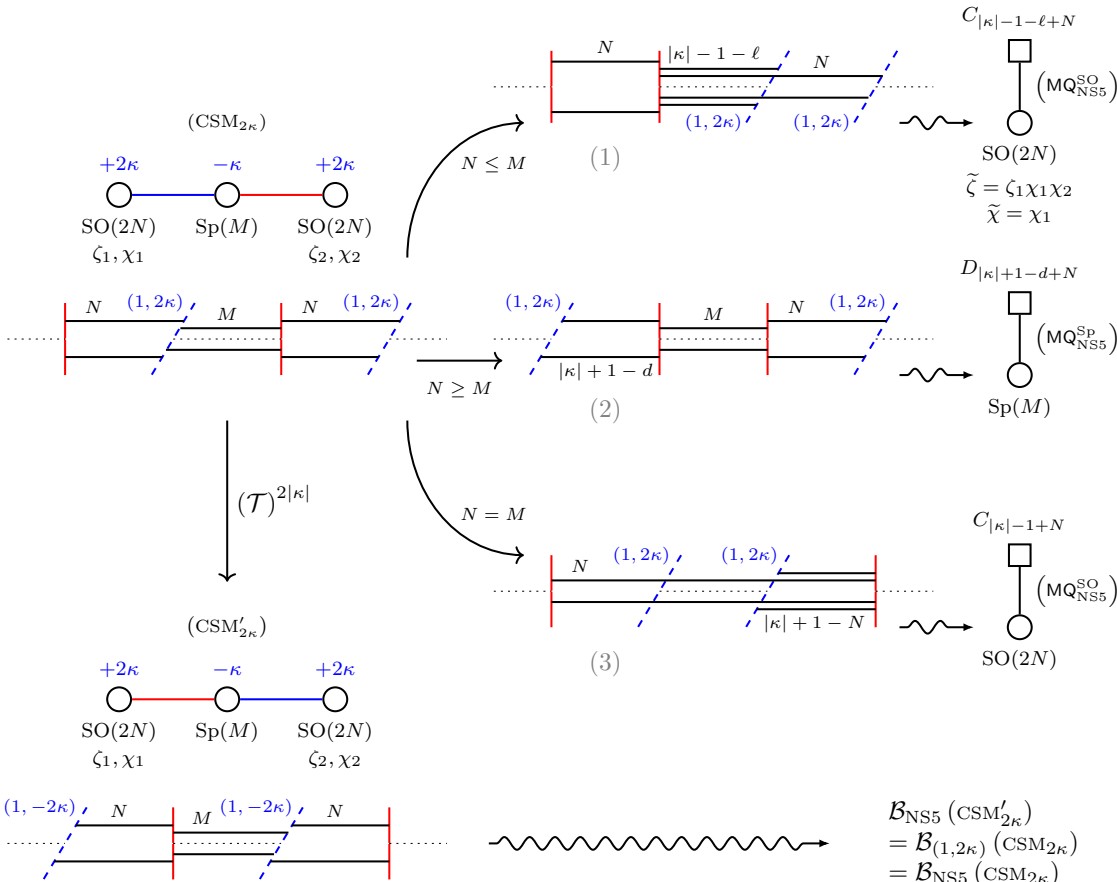

**Figure 12:** Capturing $\mathcal{B}_{\mathrm{NS5}}$ and $\mathcal{B}_{(1,2\kappa)}$ of the 3d $\mathcal{N} = 4$ $\mathrm{SO}(2N)_{+2\kappa} \times \mathrm{Sp}(M)_{-\kappa} \times \mathrm{SO}(2N)_{+2\kappa}$ theory with magnetic quivers $\mathsf{MQ}_{\mathrm{NS5}}^{\mathrm{SO}}$ and $\mathsf{MQ}_{\mathrm{NS5}}^{\mathrm{Sp}}$, where $d := N - M \geqslant 0$ (for $N \geqslant M$) and $\ell := M - N \geqslant 0$ (for $N \leqslant M$). Note that $\mathcal{B}_{\mathrm{NS5}}(\mathrm{CSM}_{2\kappa}) \cong \mathcal{B}_{(1,2\kappa)}(\mathrm{CSM}_{2\kappa})$

magnetic quivers yield the same goodness condition, i.e. $|\kappa| \geqslant M$, which therefore defines the goodness of the starting $\mathrm{CSM}_{2\kappa}$ theory.

Following the local application of the orthosymplectic dualities, the fugacities of the starting $\mathrm{CSM}_{2\kappa}$ theory map to the magnetic quivers as follows (see (3.13)):

$$\mathrm{HS}_{\mathrm{NS5}}(\mathrm{CSM}_{2\kappa})\left[a; [\chi_1, \zeta_1], [\chi_2, \zeta_2]\right] = \mathrm{HS}_{\mathcal{C}}(\mathsf{MQ}_{\mathrm{NS5}}^{\mathrm{SO}})\left[a; \widetilde{\chi} = \chi_1, \widetilde{\zeta} = \zeta_1\chi_1\chi_2\right], \quad (4.12a)$$

$$\mathrm{HS}_{(1,2\kappa)}(\mathrm{CSM}_{2\kappa})\left[b; [\chi_1, \zeta_1], [\chi_2, \zeta_2]\right] = \mathrm{HS}_{\mathcal{C}}(\mathsf{MQ}_{(1,2\kappa)}^{\mathrm{SO}})\left[b; \widetilde{\chi} = \chi_2, \widetilde{\zeta} = \zeta_2\chi_2\chi_1\right]. \quad (4.12b)$$

Note that the magnetic quivers $\mathsf{MQ}_{\mathrm{NS5}}^{\mathrm{SO}}$ and $\mathsf{MQ}_{(1,2\kappa)}^{\mathrm{SO}}$ are independent of the fugacities $\zeta_2$ and $\zeta_1$, respectively, consistently with the NS5 intervals that gives rise to the orthogonal gauge nodes. Furthermore, even though the two phases (1) and (3) of the initial brane system $\mathrm{CSM}_{2\kappa}$ yield the same magnetic quiver $\mathsf{MQ}_{\mathrm{NS5}}^{\mathrm{SO}}$, only the system (1) allows for a consistent way of following the fugacity map.

The discussion here presented for generic ranks and CS-level is accompanied by calculations for explicit values of $N$, $M$ and $\kappa$, see Table 8 in Appendix B.3. Moreover, in Table 9 one can find the index expansion for the theory $\mathrm{SO}(2N+1)_{+2\kappa} \times \mathrm{Sp}(M)_{-\kappa} \times \mathrm{SO}(2N+1)_{+2\kappa}$, which is analogous to the one just studied.

**$\mathrm{Sp}(N)_{+\kappa} \times \mathrm{SO}(2M+1) \times \mathrm{Sp}(N)_{-\kappa}$.** For an example containing $\widetilde{\mathrm{O3}}^{\pm}$ planes, see the orthosymplectic 3d $\mathcal{N} = 4$ $\mathrm{CSM}_\kappa$ theory depicted on the top left of Figure 13. Consider

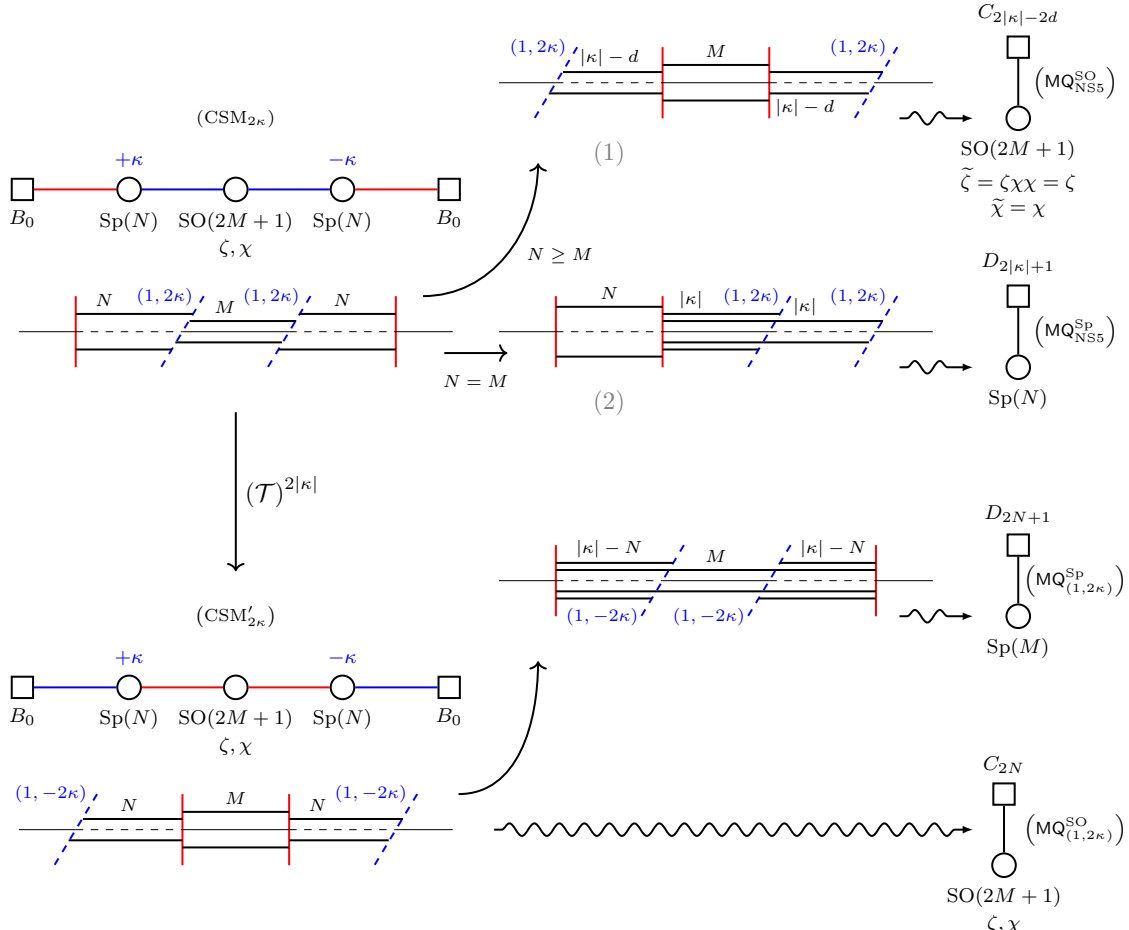

**Figure 13:** Capturing $\mathcal{B}_{\mathrm{NS5}}$ and $\mathcal{B}_{(1,2\kappa)}$ of the 3d $\mathcal{N} = 4$ $\mathrm{Sp}(N)_{+\kappa} \times \mathrm{SO}(2M+1) \times \mathrm{Sp}(N)_{-\kappa}$ theory with the pairs of magnetic quivers $\mathsf{MQ}^{\mathrm{SO}}_{\mathrm{NS5}}$, $\mathsf{MQ}^{\mathrm{Sp}}_{\mathrm{NS5}}$ and $\mathsf{MQ}^{\mathrm{SO}}_{(1,2\kappa)}$, $\mathsf{MQ}^{\mathrm{Sp}}_{(1,2\kappa)}$, respectively, where $d \coloneqq N - M \geqslant 0$ (for $N \geqslant M$).

the two different scenarios:

(1) If $N \geqslant M$, one can reach the brane system (2) in Figure 13 analogously to what has been shown for the examples of 2-nodes quivers in Section 4.2.

(2) If $N = M$, using orthosymplectic duality (see Figures 3c and 3d), move both $(1, 2\kappa)$ 5-branes to the right through the NS5 brane on the right, resulting in a stack of $2|\kappa|$-many D3s suspended between the NS5 and the first $(1, 2\kappa)$, and a stack of $|\kappa|$-many D3s between the two $(1, 2\kappa)$s. For a stable configuration, reconnect the D3s such that a stack of $|\kappa|$-many D3s is suspended between the NS5 on the right and the two $(1, 2\kappa)$ respectively. The result is the brane system (2) in Figure 13 and an associated magnetic quiver equal to the one of case (1).

Moreover, $(\mathcal{T})^{2|\kappa|}$-dualising the starting $\mathrm{CSM}_{2\kappa}$, one gets the $\mathrm{CSM}'_{2\kappa}$ theory on the bottom left part of Figure 13. It can be used to construct the $\mathsf{MQ}^{\mathrm{SO}}_{(1,2\kappa)}$ and $\mathsf{MQ}^{\mathrm{Sp}}_{(1,2\kappa)}$ magnetic quivers, whose derivation is analogous to what has been discussed so far.

The goodness condition for the magnetic quivers capturing $\mathcal{B}_{\mathrm{NS5}}$ and $\mathcal{B}_{(1,2\kappa)}$ reads $|\kappa| \geqslant N$ and $N \geqslant M$, respectively. The later is reflected in the absence of a brane phase of the initial $\mathrm{CSM}_{2\kappa}$ that is suited to read out a magnetic quiver $\mathsf{MQ}_{\mathrm{NS5}}$ for $N \leqslant M$.

Different to the previous example, the fugacity map (a sequential application of (3.13))

is trivial, i.e.

$$\text{HS}_{\text{NS5}}(\text{CSM}_{2\kappa})\left[a; \chi, \zeta\right] = \text{HS}_{\mathcal{C}}(\text{MQ}_{\text{NS5}}^{\text{SO}})\left[a; \widetilde{\chi} = \chi, \widetilde{\zeta} = \zeta\right], \tag{4.13a}$$

$$\text{HS}_{(1,2\kappa)}(\text{CSM}_{2\kappa})\left[b; \chi, \zeta\right] = \text{HS}_{\mathcal{C}}(\text{MQ}_{(1,2\kappa)}^{\text{SO}})\left[b; \widetilde{\chi} = \chi, \widetilde{\zeta} = \zeta\right]. \tag{4.13b}$$

Note that for the starting theory $\text{CSM}_{2\kappa}$ with $N \geqslant M$ one can immediately find the magnetic quiver $\text{MQ}_{\text{NS5}}^{\text{SO}}$, as well as the fugacity map.

The discussion here presented for generic ranks and CS-level is accompanied by calculations for explicit values of $N$, $M$ and $\kappa$, see Table 10 in Appendix B.3.

### 4.2.3 Generalisation to linear quivers with $n$ nodes

In order to illustrate the application of the magnetic quiver method to a linear CSM quiver theory of length $n$, consider the $\text{CSM}_{2\kappa}$ theory on top of Figure 14.

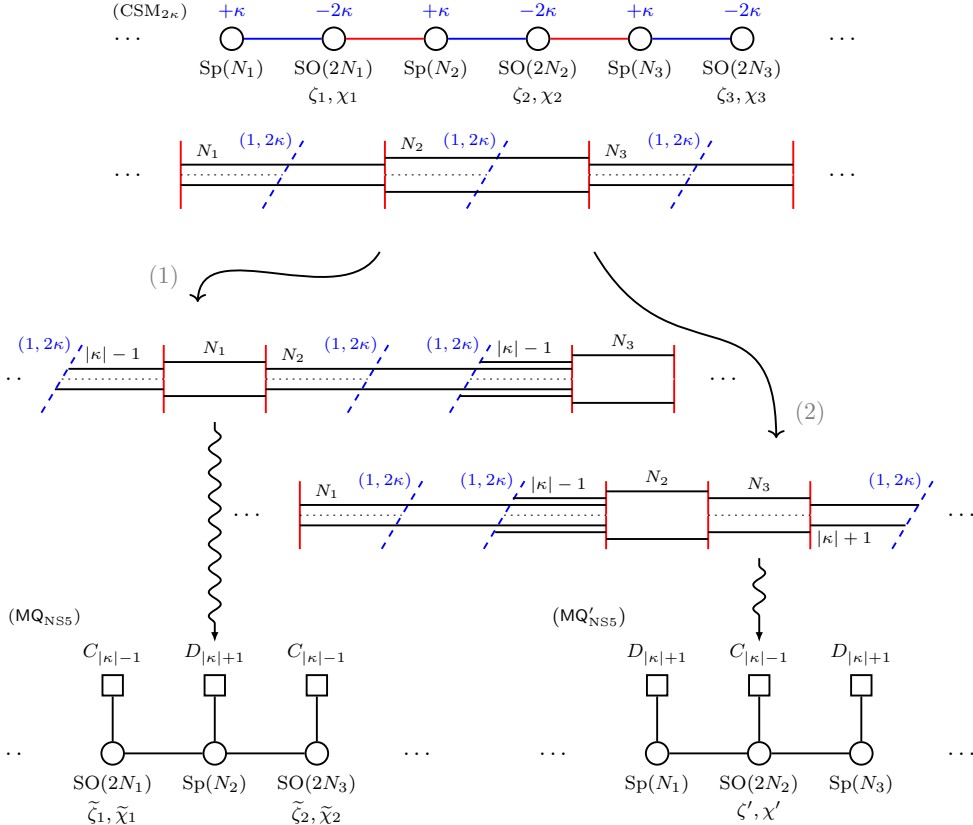

**Figure 14:** Capturing $\mathcal{B}_{\text{NS5}}$ of a linear orthosymplectic 3d $\mathcal{N} = 4$ $\text{CSM}_{2\kappa}$ theory with magnetic quivers $\text{MQ}_{\text{NS5}}$ and $\text{MQ}'_{\text{NS5}}$.

Analogous to the previous examples, using orthosymplectic dualities (see Figure 3), one can move the initial brane system $\text{CSM}_{2\kappa}$ into the two distinct phases labelled (1) and (2), giving rise to the two different magnetic quivers $\text{MQ}_{\text{NS5}}$ and $\text{MQ}'_{\text{NS5}}$. Establishing the fugacity maps between the $\{\zeta_i, \chi_i\}$ of the CSM quiver and $\{\widetilde{\zeta}_i, \widetilde{\chi}_i\}$ of $\text{MQ}_{\text{NS5}}$ (resp. $\{\zeta'_i, \chi'_i\}$ of $\text{MQ}'_{\text{NS5}}$) is left for future analysis, because of current limitations in verifying such maps via index computations. The expectation, however, is that analogous arguments as in Section 3 apply. Besides these technical points, such analysis would shed light in the interplay between global forms of the CSM quiver theory and the required global forms

in the magnetic quiver to describe the maximal branch. This raises the open question on what information on $\mathcal{B}_{\mathrm{NS5}}$ is captured in $\mathsf{MQ}_{\mathrm{NS5}}$ and $\mathsf{MQ}'_{\mathrm{NS5}}$, how do they differ, and what is encoded in both.

### 4.2.4 Circular examples

So far, the magnetic quiver method has been applied to linear orthosymplectic 3d $\mathcal{N} = 4$ quiver theories. In this section, the analysis is showcased in circular orthosymplectic CSM quiver theories. This serves as proof-of-concept, but detailed fugacity maps and index computations are not feasible yet.

**Example with O3$^\pm$ planes.** As a first example, consider the circular orthosymplectic 3d $\mathcal{N} = 4$ CSM$_{2\kappa}$ theory on top of Figure 15.

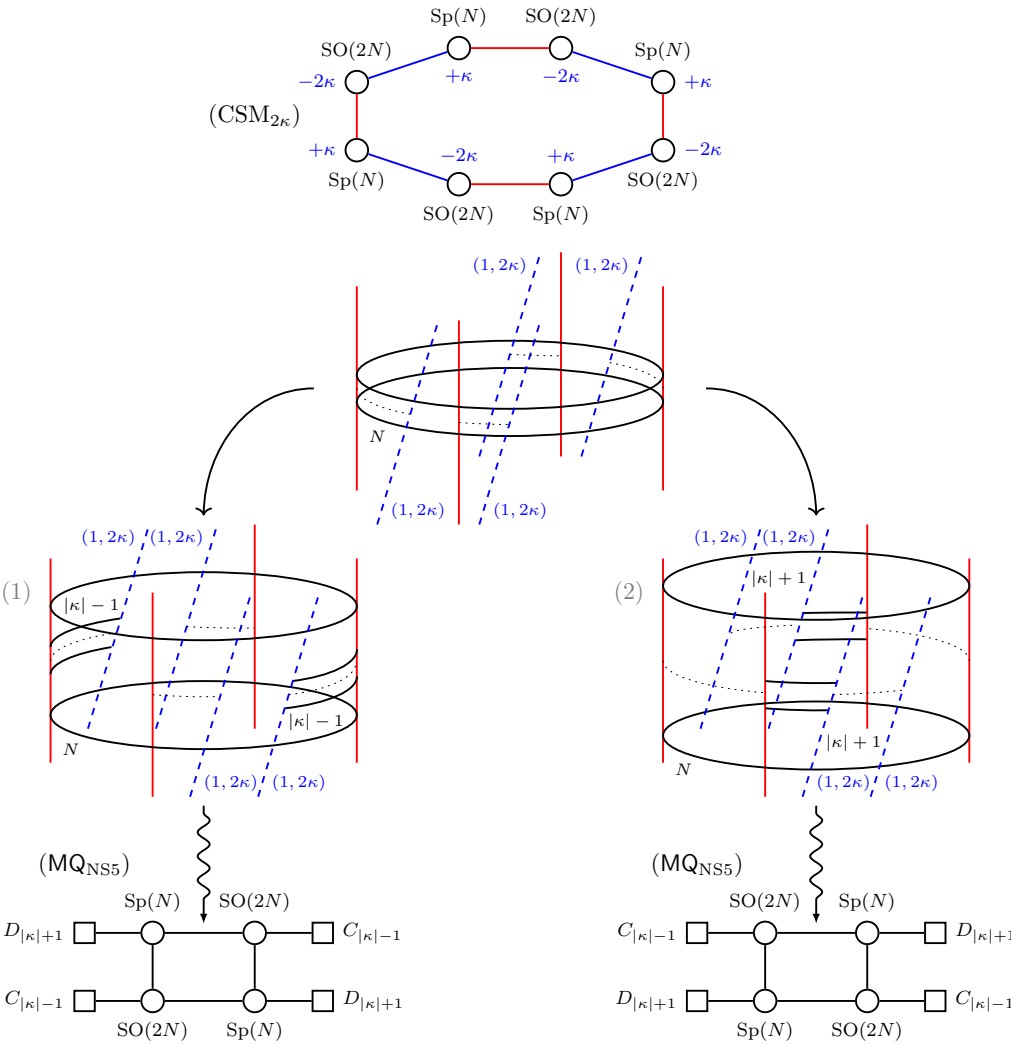

**Figure 15:** Circular brane system with four half NS5 and four $(1, 2\kappa)$ 5-branes, intersected by O3$^\pm$ orientifolds. The $\mathcal{B}_{\mathrm{NS5}}$ branch is captured by magnetic quivers $\mathsf{MQ}_{\mathrm{NS5}}$, derived from a canonical brane phase. Note that $\mathcal{B}_{(1,2\kappa)} \cong \mathcal{B}_{\mathrm{NS5}}$ due to the symmetry of the brane configuration.

Analogously to the linear quiver examples considered so far, one can move the initial brane system CSM$_{2\kappa}$ into the two phases labelled (1) and (2) in Figure 15. Despite the two phases being different from each other, applying the magnetic quiver method results in the same magnetic quiver $\mathsf{MQ}_{\mathrm{NS5}}$. Note, however, that the gauge and flavour nodes arise from different NS5 intervals and differently positioned $(1, 2\kappa)$ 5-branes.

Moreover, the $\mathcal{B}_{(1,2\kappa)}$ branch is isomorphic to $\mathcal{B}_{\text{NS5}}$, which is also manifest from the symmetric brane arrangement.

**Example with O3$^\pm$ and $\widetilde{\text{O3}}^\pm$ planes.** For the second example consider a brane system involving $(1, 2\kappa + 1)$ 5-branes, i.e. an odd amount of half D5 charge, see the brane system $\text{CSM}_{2\kappa+1}$ on top of Figure 16.

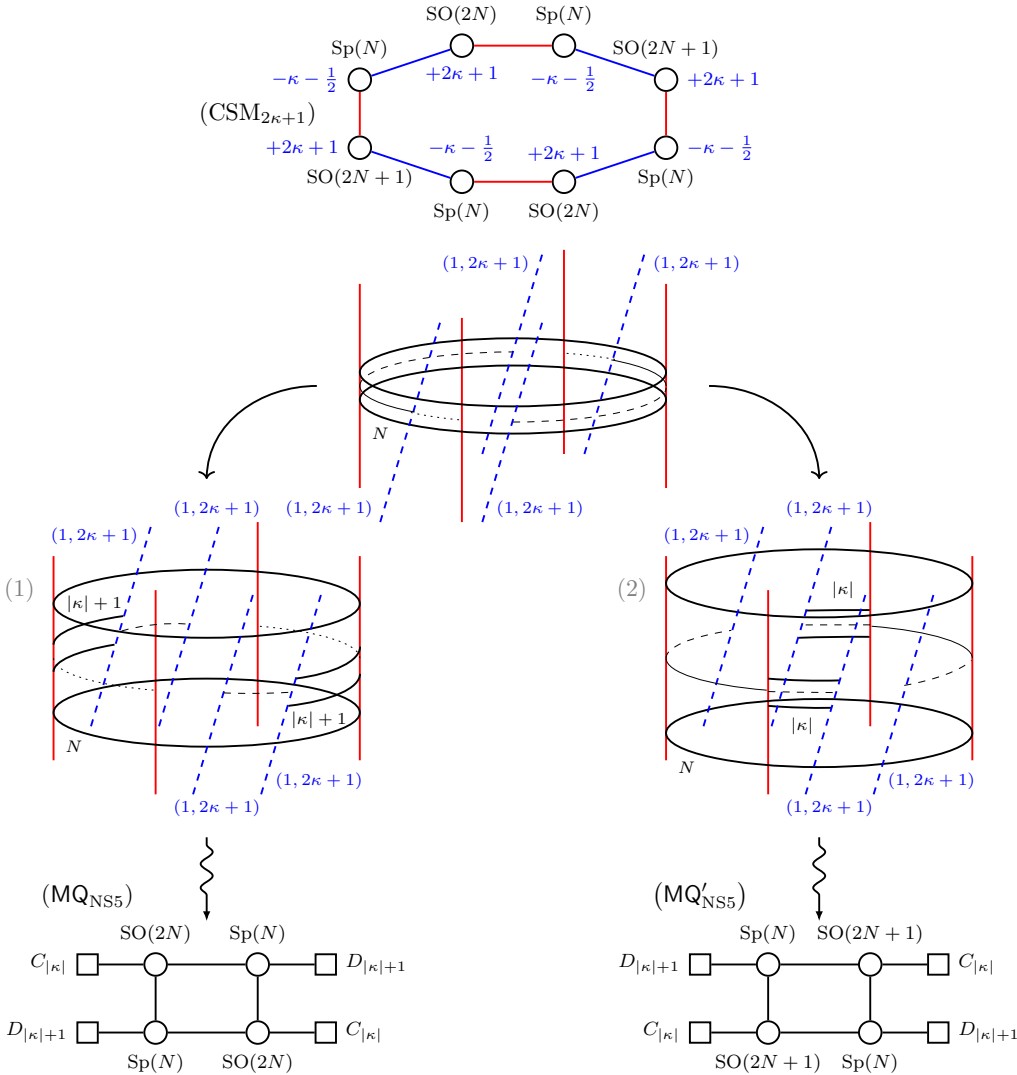

**Figure 16:** Circular brane system with four NS5 and four $(1, 2\kappa + 1)$ 5-branes, intersected by O3 planes. The $\mathcal{B}_{\text{NS5}}$ branch is captured by magnetic quivers $\text{MQ}_{\text{NS5}}$ and $\text{MQ}'_{\text{NS5}}$, derived from inequivalent canonical brane phases. Also, $\mathcal{B}_{(1,2\kappa+1)} \cong \mathcal{B}_{\text{NS5}}$ due to the symmetric configuration.

Similar to the previous example, moving the $(1, 2\kappa+1)$ 5-branes and taking into account the D3 brane creation, one finds the two distinct phases (1) and (2) that the initial brane system $\text{CSM}_{2\kappa+1}$ can move into. Both phases give rise to two distinct magnetic quivers, $\text{MQ}_{\text{NS5}}$ and $\text{MQ}'_{\text{NS5}}$, respectively. For a detailed analysis of the magnetic quivers, i.e. what global forms of the CSM quiver is captured, one needs to establish the fugacity map. This is left as an open question, as these theories are computationally demanding.

## 4.3 Magnetic Quivers for Orthosymplectic 3d $\mathcal{N} = 3$ CSM theories

The brane systems in Section 4.2 have only two types of 5-branes: hence, giving rise to 3d $\mathcal{N} = 4$ SCFTs on the D3 world-volume. In terms of the magnetic quiver proposal, this has the advantage that the Coulomb branch Hilbert series of the magnetic quivers can be compared against the maximal branch limits of the superconformal index. Based on the consistency checks and previous insights, an exploration of more general brane configurations is now undertaken. These include NS5s, (one or several types of) $(1, q)$ 5-branes, and possibly D5 branes. The D3 world-volume theory is generically a 3d $\mathcal{N} = 3$ Chern–Simons matter theory with as many distinct maximal branches as there are distinct type of 5-branes. In this section, the magnetic quiver proposal is tested and found to access all of these maximal hyper-Kähler branches in the considered cases, as long as conditions 1–3 are satisfied. In these examples, no checks via limits of the superconformal index are currently available, so the magnetic quivers should be regarded as predictions. While the associated fugacity maps are expected to follow similar patterns as in the $\mathcal{N} = 4$ case, they are not worked out here and are deferred to future work, once index computations become feasible.

**Example 1: CSM with fundamental flavours.** Consider the following example (with $F_1, F_3 > N$ for concreteness)

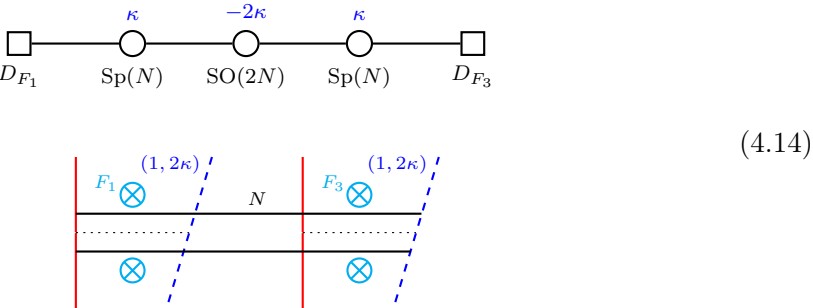

$$(4.14)$$

which exhibits two stacks of D5 branes (in cyan). The numbers $F_{1,3}$ count full D5s. As there are three distinct types of 5-branes, there are three maximal branches — denoted as $\mathcal{B}_{\mathrm{NS5}}$, $\mathcal{B}_{(1,2\kappa)}$, and $\mathcal{B}_{\mathrm{D5}}$. The magnetic quivers are straightforwardly derived as follows:

- $\mathcal{B}_{\mathrm{NS5}}$: There are two canonical brane phases. For instance, move all $2F_1$ many half D5s through the left-most half NS5, paying attention to brane creation, see Appendix A.2. Thereafter, move the left $(1, 2k)$ 5-brane through the left half NS5, again with D3 creation. The resulting brane phase allows to read off an orthogonal SQCD type $\mathsf{MQ}_{\mathrm{NS5}}$. Alternatively, start again from (4.14), but move the left $(1, 2k)$ 5-brane through the right half NS5. This create a phase that allows to read off a symplectic SQCD type $\mathsf{MQ}_{\mathrm{NS5}}$. In summary, one finds.

$$C_{N+F_1+|\kappa|-1} \qquad\qquad D_{N+F_1+|\kappa|+1}$$

$$\square \qquad\qquad\qquad\qquad \square$$
$$\Big| \qquad , \qquad \text{and} \qquad \Big| \qquad . \qquad (4.15)$$
$$\bigcirc \qquad\qquad\qquad\qquad \bigcirc$$
$$SO(2N) \qquad\qquad\qquad\qquad Sp(N)$$

These quivers fit with expected equivalence of Figure 10. However, in contrast to $\mathcal{N} = 4$ CSM theories, here one cannot compare the Hilbert series of the various global form of (4.15) with some limit of the index.

- $\mathcal{B}_{(1,2\kappa)}$: After the $(\mathcal{T})^{2\kappa}$ duality that swaps NS5s with $(1, 2k)$ 5-branes and leaves the D5s invariant, one realises that the brane systems is that of (4.14) with $F_1$ and $F_3$

swapped. (This follows as all O3s are invariant under $(\mathcal{T})^{2\kappa}$, see (2.2b).) Hence, the magnetic quivers for this branch are obtained from (4.15) by replacing $F_1$ with $F_3$.

- $\mathcal{B}_{\mathrm{D5}}$: To turn D5s into NS5s, the $\mathcal{S}$ map is the suitable duality, which then leads to a brane systems that has two stacks of NS5s: on the left is a stack of $2F_1$ half NS5 and on the right are $2F_3$ many half NS5s. The two original half NS5s in (4.14) are mapped to two half D5s. Lastly, the $(1, 2k)$ turn into $(2k, -1)$ 5-branes. Importantly, as $\mathcal{S}$ acts non-trivially on the O3s, see (2.2a), one finds

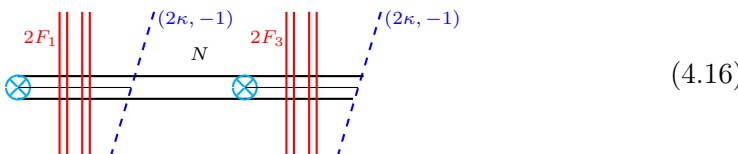

$$(4.16)$$

To transition into the canonical phase for the maximal branch, one has to move the half D5 on the left-hand side through sufficiently many of the $2F_1$ half NS5s. This standard D3 creation/annihilation terminates after $2N$ moves, because by assumption $F_1 > N$. Next, one needs to move the $(2\kappa, -1)$ 5-brane through the some half NS5s on the right-side as well. Here, however, D3 creation/annihilation proceeds as in the D5-NS5 case on the left, because $|\det \left( \begin{smallmatrix} 1 & 2\kappa \\ 0 & -1 \end{smallmatrix} \right)| = 1$. After standard brane moves, one finds

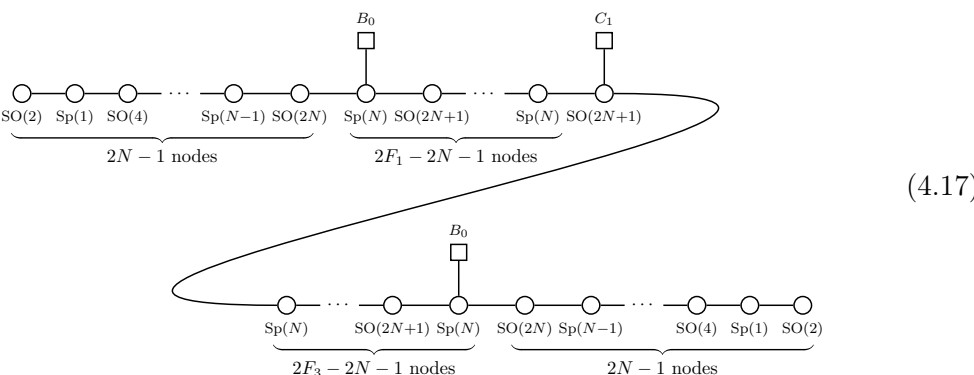

$$(4.17)$$

This magnetic quiver exhibits two long subsets of balanced nodes, cf. Appendix A.3. To see this, recall that a $\ldots - \mathrm{SO}(2\ell) - \mathrm{Sp}(\ell) - \mathrm{SO}(2\ell + 2) - \mathrm{Sp}(\ell + 1) - \ldots$ segment is balanced for each node, and likewise segments of the form $\ldots - \mathrm{SO}(2\ell + 1) - \mathrm{Sp}(\ell) - \mathrm{SO}(2\ell + 1) - \mathrm{Sp}(\ell) - \ldots$. Thus, the entire left chain is balanced up to the central $\mathrm{SO}(2N + 1)$ nodes, which is good, but not balanced. The length of this balanced chain is $2F_1 - 2$, which leads to a $\mathfrak{so}(2F_1)$ global symmetry factor. The analogous arguments holds for the right hand side. Thus, the overall $\mathcal{B}_{\mathrm{D5}}$ isometry algebra read off from the magnetic quiver is $\mathfrak{so}(2F_1) \oplus \mathfrak{so}(2F_3)$. This is exactly what the brane systems (4.14) exhibits.

As comment, the brane system (4.16) is technically already outside the class of theories considered so far, as it contains the $(p, q)$ 5-brane with $|p| > 1$. This, however, just shows that there is no reason to restrain the setup to $(1, q)$ 5-branes, and this extension is subject of Section 4.4.

**Example 2: CSM with different CS-levels.** Consider the following example (with $\kappa_2 > \kappa_1 > 0$ for concreteness, and $N \leqslant \kappa_2 + 1$).

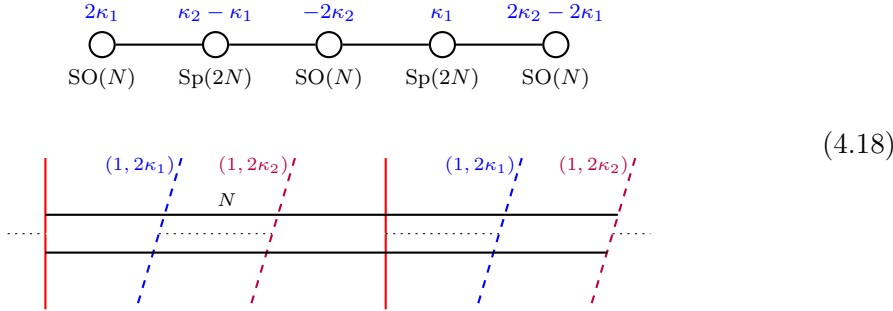

$$(4.18)$$

The orthosymplectic CSM quiver features three different CS-levels. Based on the three different 5-branes, there exist again three maximal branches. The magnetic quivers are derived as follows:

- $\mathcal{B}_{\text{NS5}}$: By assumption, $N \leqslant \kappa_2 - 1$ and $\kappa_2 > \kappa_1$, which ensure that all $N$ D3s to the right of the second NS5 are frozen between the NS5 and the $(1, 2\kappa_{1,2})$ 5-branes.

- $\mathcal{B}_{(1,2\kappa_1)}$: The convenient brane configuration is reached after a $(\mathcal{T})^{2\kappa_1}$ and standard brane moves. Note that NS5s and $(1, 2\kappa_1)$ 5-branes are swapped, while $(1, 2\kappa_2)$ is mapped to a $(1, 2\kappa_2 - 2\kappa_1)$ 5-brane.

- $\mathcal{B}_{(1,2\kappa_2)}$: The magnetic quiver is read off after a $(\mathcal{T})^{2\kappa_2}$.

In conclusion, the maximal branches are captured by the following magnetic quivers:

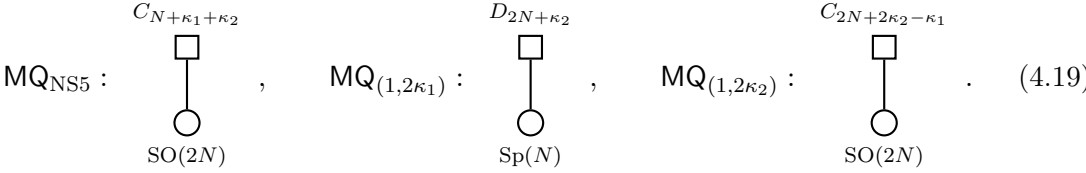

$$(4.19)$$

Here, only one magnetic quiver for each branch has been shown, but by previous arguments, one finds also an SO or Sp-type magnetic quiver as long as the number of D3s is the same throughout all 5-brane intervals.

## 4.4 Magnetic quivers for non-Lagrangian CSM theories

So far, the brane configurations allowed for NS5s, D5s, and any number of $(1, q_i)$ 5-branes. The D3 world-volume theories are Lagrangian orthosymplectic Chern–Simons theories with matter. However, one can equally well consider brane configurations with $(p, q)$ 5-branes (with $p > 0$, $q \neq 0$, and $p$, $q$ coprime integers). While the resulting 3d $\mathcal{N} \geqslant 3$ SCFTs lack a known Lagrangian description, their maximal branches remain amenable to analysis via the magnetic quiver proposal. This is illustrated through two examples below. In these cases, no consistency checks via superconformal index limits are currently available, nor is it feasible to construct fugacity maps due to the non-Lagrangian nature of the theories. As such, the proposed magnetic quivers should be viewed as predictions, which may require refinement in future work as further tools become available.

**Example 1: $\mathcal{N} = 4$ CSM with $(p, q)$ 5-branes.** Consider a system with NS5s and $(p, q)$ 5-branes (with $p, q$ coprime, and $p = 2\ell + 1$, $q = 2\kappa$ for concreteness)

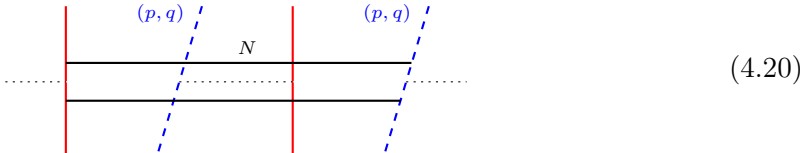

$$(4.20)$$

which admits no Lagrangian UV description. Nonetheless, the magnetic quiver proposal is applicable and one straightforwardly finds the following:

- $\mathcal{B}_{\text{NS5}}$: Moving a $(p, q)$ 5-brane through an NS5 brane creates or annihilates $\left| \det \left( \begin{smallmatrix} 1 & p \\ 0 & q \end{smallmatrix} \right) \right| = |q|$ many (half) D3s (up to $\pm 2$ from O3$^{\mp}$ effects). Hence, assuming that $N \leqslant \kappa$, one finds two magnetic quivers, depending on which $(p, q)$ 5-branes is moved:

$$
\begin{array}{ccc}
D_{N+\kappa+1} & & C_{N+\kappa-1} \\
\square & & \square \\
| & \text{and} & | \\
\bigcirc & & \bigcirc \\
\text{Sp}(N) & & \text{SO}(2N)
\end{array}
\qquad (4.21)
$$

- $\mathcal{B}_{(p,q)}$: Here, one utilises an SL$(2, \mathbb{Z})$ map $M$ of the following form:

$$
M = \left( \begin{smallmatrix} p & * \\ q & \star \end{smallmatrix} \right) \quad \text{s.t.} \quad M.(1,0) = (p, q) \quad \text{and} \quad M^{-1} = \left( \begin{smallmatrix} \star & -* \\ -q & p \end{smallmatrix} \right) \qquad (4.22)
$$

applying this onto the two types of 5-branes in (4.20), yields

$$
M^{-1}.(p, q) = (1, 0) = \text{NS5} \quad \text{and} \quad M^{-1}.(1, 0) = (\star, -q) \,. \qquad (4.23)
$$

In fact, one can deduce that $\star$ is an odd integer as follows: recall that $M \in \text{SL}(2, \mathbb{Z})$ requires $\det(M) = p \cdot \star - q \cdot * \overset{!}{=} 1$. By assumption, $p$ is odd and $q$ is even. Thus, $q \cdot *$ is always even. Therefore, $p \cdot \star$ has to be odd, which implies that $\star$ is odd.

In addition, one has to consider the effect of the $M^{-1}$ transformation on the O3 planes. Using (negative) continued fractions, the following holds (cf. [53]):

$$
M = (-1)^{r-1} \mathcal{T}^{k_1} \mathcal{S} \mathcal{T}^{k_2} \mathcal{S} \cdot \ldots \cdot \mathcal{S} \mathcal{T}^{k_r} \qquad \text{with} \quad \frac{p}{q} = \cfrac{1}{k_1 - \cfrac{1}{k_2 - \cfrac{1}{\ldots - \frac{1}{k_r}}}} \,. \qquad (4.24)
$$

Recalling $\mathcal{S}^2 = -\mathbb{1}$ and $\mathcal{T}^{-k} \mathcal{T}^k = \mathbb{1}$ (for any $k \in \mathbb{N}$), one arrives at

$$
M^{-1} = \mathcal{T}^{-k_1} \mathcal{S} \cdot \ldots \cdot \mathcal{S} \mathcal{T}^{-k_2} \mathcal{S} \mathcal{T}^{-k_1} \,. \qquad (4.25)
$$

Based on the actions of $\mathcal{T}$ and $\mathcal{S}$ on O3s (see (2.2b) and (2.2a)), one observes that $M^{-1}$ leaves O3$^-$ invariant. This is enough the construct the $M^{-1}$ dual brane system of (4.20), by consistency of O3s passing through NS5 as well as $(\star, -q)$ 5-brane (with $\star$ odd and $q$ even):

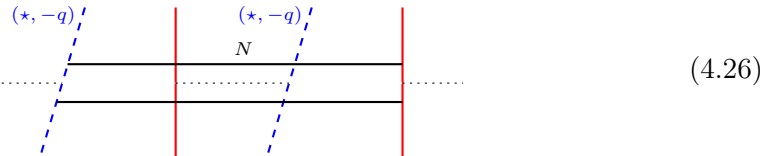

$$(4.26)$$

The magnetic quiver for (4.26) is derived to be

$$
\begin{array}{c}
D_{N+\kappa+1} \\
\square \\
| \\
\bigcirc \\
\mathrm{Sp}(N)
\end{array}
\tag{4.27}
$$

wherein one needs to assume that $N \leqslant \kappa$. Moreover, the two maximal branches are in fact isomorphic, as expected from the symmetric brane arrangement (4.20).

**Example 2: $\mathcal{N} = 3$ CSM with $(p, q)$ 5-branes.** Consider the $\mathcal{N} = 3$ theory that results from the following brane system

$$\tag{4.28}$$

where the cyan branes denote two stacks of $2F_1$ and $2F_3$ half D5s. For concreteness, assume that $p, q$ are coprime, $p = 2\ell + 1$ odd, $q = 2\kappa$ even, and $F_{1,3} > N$. This is a non-Lagrangian generalisation of the case in (4.14), obtained by replacing $(1, 2\kappa)$ 5-branes by $(p, q)$ 5-branes.

Despite the non-Lagrangian nature, the magnetic quivers for the three maximal branches are within reach. One finds:

- $\mathcal{B}_{\mathrm{NS5}}$: By standard brane moves, one finds two phases which give rise to an orthogonal and a symplectic $\mathsf{MQ}_{\mathrm{NS5}}$:

$$
\begin{array}{ccc}
\begin{array}{c}
D_{F_1+\kappa+1+N} \\
\square \\
| \\
\bigcirc \\
\mathrm{Sp}(N)
\end{array}
& \quad , \qquad \text{and} \qquad &
\begin{array}{c}
C_{F_1+\kappa-1+N} \\
\square \\
| \\
\bigcirc \\
\mathrm{SO}(2N)
\end{array}
\end{array}
\tag{4.29}
$$

  Comparing to (4.15) shows that the NS5 branch is not effected by replacing $(1, 2\kappa)$ with $(2\ell + 1, 2\kappa)$.

- $\mathcal{B}_{(p,q)}$: The relevant $\mathrm{SL}(2, \mathbb{Z})$ transformation is (4.22). The transformed NS5 and $(p, q)$ 5-branes are given by (4.23); in addition, the D5 branes becomes

$$
M^{-1}.(0, 1) = (-*, p) \, .
\tag{4.30}
$$

  Note that $*$ can both be even or odd integer, in contrast to $\star$, which has to be odd. Nonetheless, the duality map $M^{-1}$ leaves the $\mathrm{O3}^-$ plane invariant, which is sufficient to construct a magnetic quiver. One finds:

$$
\begin{array}{c}
D_{p \cdot F_3+\kappa+1+N} \\
\square \\
| \\
\bigcirc \\
\mathrm{Sp}(N)
\end{array}
\tag{4.31}
$$

  Compared to the $\mathsf{MQ}_{(1,2\kappa)}$ of (4.14), the $\mathsf{MQ}_{(p,q)}$ is not obtained from $\mathsf{MQ}_{\mathrm{NS5}}$ by just replacing $F_1$ with $F_3$. Here, due to the modified $p$ charge, one needs to replace $F_1$ with $p \cdot F_3$. This also implies that even for $F_1 = F_3$ the two branches $\mathcal{B}_{\mathrm{NS5}}$ and $\mathcal{B}_{(p,q)}$ are not isomorphic, despite the symmetric looking brane arrangement (4.28).

- $\mathcal{B}_{\mathrm{D5}}$: After applying $\mathcal{S}$ on (4.28), one arrives at

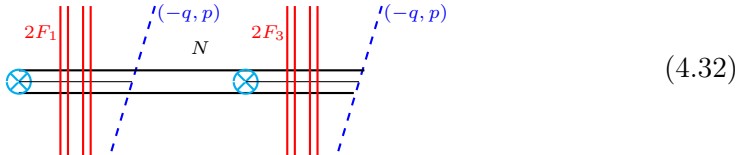

$$(4.32)$$

which manifests a two stacks of $2F_1$ and $2F_3$ half NS5s, respectively. The left half D5 brane has $N$ full D3s ending in it, which all can end on some of the $2F_1$ half NS5s, provided $N < F_1$. These are standard brane moves. The $(-q, p)$ 5-brane on the right-hand side can undergo brane creation/annihilation moves in alternating steps of $\ell$ D3s and $\ell + 1$ D3s, see Appendix A.2. Hence, there are several cases that need to be discussed separately.

If $N \leqslant \frac{p-1}{2} \equiv \ell$, then the right $(-q, p)$ 5-brane accommodates all $N$ D3s, and no brane move is necessary. Consequently, $\mathsf{MQ}_{\mathrm{D5}}$ reads

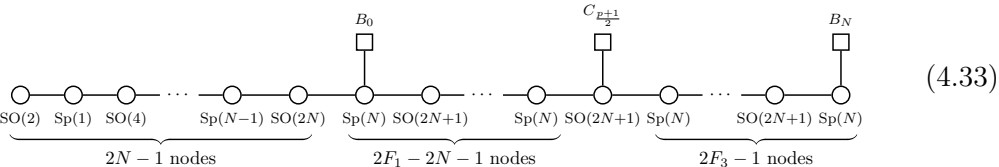

$$(4.33)$$

and one observes two linear connected sets of balanced nodes. Analogously to (4.17), the left $2F_1 - 2$ nodes are all balanced and give rise to an $\mathfrak{so}(2F_1)$ symmetry factor, see Appendix A.3. Moreover, the right chain of $2F_3 - 1$ nodes are also balanced, giving rise to an enhanced $\mathfrak{so}(2F_3)$ global symmetry due to monopole operators.

If $N > \frac{p-1}{2} \equiv \ell$, then not all $N$ D3s can end on the $(-q, p)$ 5-brane and one has to move the the $(-q, p)$ 5-brane through sufficiently many half NS5s. There exist two separate cases, depending on how many elementary brane moves can be done. If $N = a \cdot p + \ell + b$ with $a \in \mathbb{N}_0$, $b \in \{0, 1, \dots, \ell - 1\}$, then the $(-q, p)$ 5-branes moves through $2a + 1$ many half D5s during which it creates/annihilates first $\ell$, then $\ell + 1$, again $\ell$, and so forth many D3s. The magnetic quiver then reads

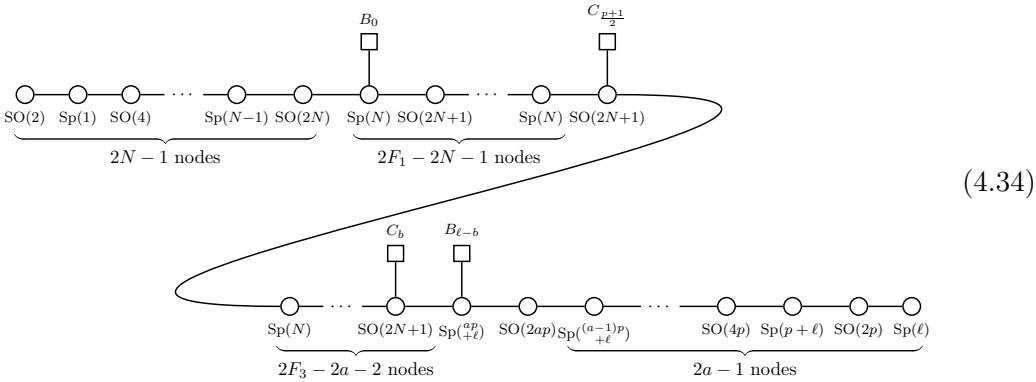

$$(4.34)$$

Again, the left chain of balanced nodes displays the $\mathfrak{so}(2F_1)$ symmetry algebra, while the right side still exhibits a chain of $2F_3 - 1$ many balanced node; hence, showcasing the $\mathfrak{so}(2F_3)$ symmetry.

If $N = a \cdot p + b > \frac{p-1}{2} \equiv \ell$ with $a \in \mathbb{N}_{>0}$, $b \in \{0, 1, \dots, \ell - 1\}$, the $(-q, p)$ 5-brane passes through $2a$ many half NS5, during which it creates/annihilates alternatingly

$\ell$ and $\ell + 1$ many D3s. One then derives the following $\mathsf{MQ}_{D5}$:

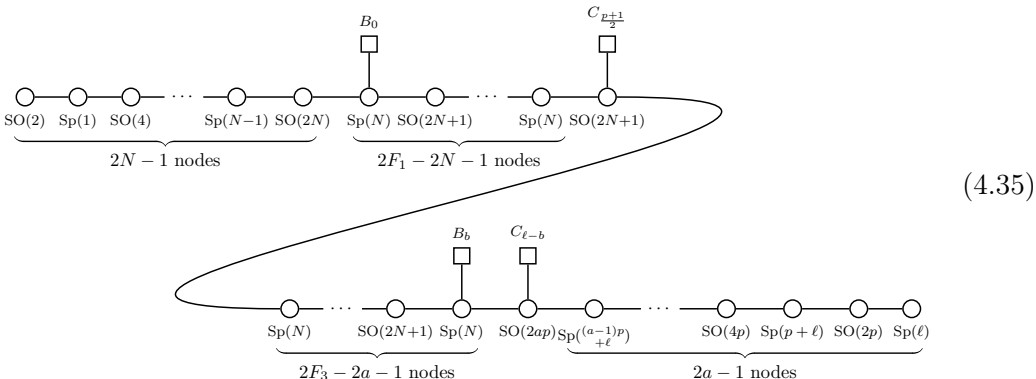

$$(4.35)$$

The analysis of balanced still produces the $\mathfrak{so}(2F_1) \oplus \mathfrak{so}(2F_3)$ symmetry algebra, as expected from the brane system (4.28). As a remark, the $\mathcal{B}_{NS5}$ branch (4.32) is substantially modified by the higher $p$-value, compare to the $\mathcal{B}_{NS5}$ branch (4.16) with magnetic quiver (4.17).

# 5   A puzzle

There are brane systems that do not transition into a suitable phase through simple brane creation or annihilation moves, i.e., they do not satisfy conditions 1–3. In such cases, the derivation of a magnetic quiver becomes opaque. Here, this puzzle is addressed, and an operator counting is proposed.

**Example 1.**   For instance, consider the following configuration:

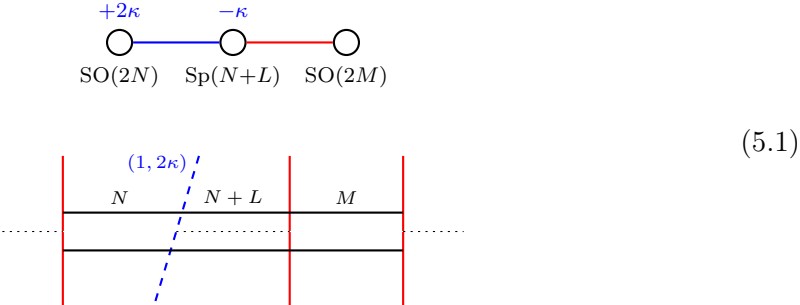

$$(5.1)$$

The $(1, 2\kappa)$ 5-brane has extra $L$ D3 branes suspended between itself and the central NS5. This implies that one should at most pass the $(1, 2k)$ 5-brane through the central NS5, in order to keep the number of frozen D3s on the NS5-branch minimal. However, this does not lead to a brane configuration in which a magnetic quiver can be read off, because the orientifold type in a single NS5 interval changes between O3$^+$ to O3$^-$.

This presents a puzzle. If one tries to move the $(1, 2k)$ 5-brane all the way to the left (or all the way to the right), the putative magnetic quivers fail to match the NS5-branch limit of the CSM index. This is expected, as already the 2-node CSM quivers with unequal ranks did not give rise to valid magnetic quivers if the $(p, q)$ 5-branes where moved against the "natural" direction.

In an attempt to overcome this puzzle, one can try to construct a consistent counting of the NS5-branch operators of (5.1). To do so, proceed in three steps: firstly, recall that the right-most NS5 interval already contributes an $\mathcal{N} = 4$ gauge node in the CSM theory. Hence, to capture these NS5-branch operator one needs to include a SO($2M$) gauge node with number of flavours given by $N + L$. Secondly, the next ingredient is the 2-node CSM

subquiver that stems from the left NS5 interval. Here, based on previous sections, the NS5-branch operators are counted by an SO($2N$) gauge node with $\kappa - 1 - L$ many flavours (not yet accounting for the $M$ D3s in the adjacent interval). Thirdly, combining these two leads to following quiver-like auxiliary object:

$$
\begin{array}{ccc}
C_{\kappa-1-L} & & C_L \\
\square & & \square \\
| & & | \\
\underset{\text{SO}(2N)}{\bigcirc} & \!\!\!\!\!\!\!\!\!\overset{\frac{1}{2}}{\rule{3.5cm}{0.4pt}}\!\!\!\!\!\!\!\! & \underset{\text{SO}(2M)}{\bigcirc}
\end{array}
\tag{5.2}
$$

The claim is that a monopole-formula computation of (5.2) matches the NS5-branch limit of the CSM index. This, however, needs some explanation. First, the GNO lattice and dressing factor of SO($2N$) and SO($2M$) are the standard ones. The links between the SO-nodes and the $C$-type flavours contribute as half-hypermultiplets, as usual. The only new ingredient is the link between the two SO-nodes. It transforms as bi-vector in the product group SO($2N$) × SO($2M$); recall that the vector representation is real, hence the product is also a real representation. Thus, the flavour symmetry of such a bi-vector representation is of Sp-type. This implies that the minimal such matter field really yields a bi-vector hypermultiplet (because the fundamental of Sp(1) is 2-dimensional). However, including a bi-vector hypermultiplet in the monopole formula of (5.2) does not yield the correct counting, because the SO($2M$) node would see $2N + L$ flavours. Thus, to correct the counting, one imposes *by hand* that only half of the bi-vector contribution is counted.

Appendix B.4 showcases that this construction allows to match with the Hilbert series limit of the index. However, the physical interpretation of (5.2) remains unclear for now.

**Example 2.** Next, consider a different configuration:

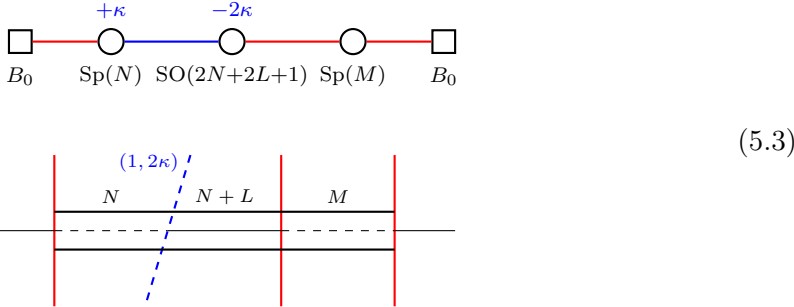

$$\tag{5.3}$$

which exhibits the same problem.

Here, one can try to repeat the same construction principle of a quiver-like object that captures the operator counting on the NS5-brane of the CSM theory (5.3). The three steps are as follows: Firstly, the right-most NS5 interval yields a $\mathcal{N} = 4$ Sp($M$) node with $\frac{1}{2}(2N+2L+1)+\frac{1}{2} = N+L+1$ many flavours. Secondly, the left NS5 interval is reminiscent of the 2-node CSM systems considered before. Consequently, one would assign an Sp($N$) gauge node with $\frac{1}{2}(2\kappa - 2L + 1) + \frac{1}{2} = \kappa - L + 1$ fundamental flavours (not yet accounting for the $M$ D3s in the adjacent interval). Thirdly, combining both results in:

$$
\begin{array}{ccc}
D_{\kappa+1-L} & & D_{L+1} \\
\square & & \square \\
| & & | \\
\underset{\text{Sp}(N)}{\bigcirc} & \!\!\!\!\!\!\!\!\!\overset{\frac{1}{2}}{\rule{3.5cm}{0.4pt}}\!\!\!\!\!\!\!\! & \underset{\text{Sp}(M)}{\bigcirc}
\end{array}
\tag{5.4}
$$

whose monopole formula reproduces the operator count on the NS5-branch of (5.4). For this, the ingredients are as follows: the GNO lattice and dressing factors of Sp nodes are the standard ones. The same is true for the half-hypermultiplets between the Sp gauge nodes and the $D$-type flavours. The link between the Sp nodes require a modification: it transforms in the bi-vector of $\mathrm{Sp}(N) \times \mathrm{Sp}(M)$. Since the product of two pseudo-real representations is a real representation, the associated flavour symmetry is of Sp-type. Hence, the bi-vector is again a full hypermultiplet. However, this would spoil the numbers of flavours the $\mathrm{Sp}(M)$ node perceives, and results in an incorrect operator counting. Thus, one imposes *by hand* that the bi-vector contribution is weighted with a half.

Appendix B.4 showcases that (5.4) leads to a consistent counting, albeit the physical interpretation of (5.4) is yet to be understood.

**Remark.** In principle, the constructed quiver-like objects (5.2) and (5.4) could be combined with the "regular" magnetic quivers of Section 4 to yield predictions for CSM brane systems beyond the conditions 1–3. In view of the unclear physical meaning, this is left for future work.

# 6 Conclusions and outlook

The 3d $\mathcal{N} \geqslant 3$ SCFTs realised by Type IIB brane configurations with D3 branes on O3 orientifolds and intersecting various $(p, q)$ 5-branes exhibit two central features: (i) dualities sensitive to discrete $\mathbb{Z}_2$-valued symmetry fugacities, and (ii) maximal branches of the vacuum moduli space.

The first main result concerns the two basic dualities for orthogonal and symplectic Chern–Simons matter (CSM) theories. While the dualities follow naturally from brane creation/annihilation processes (Appendix A.2), the corresponding maps of $\mathbb{Z}_2^{\mathcal{M}}$ and $\mathbb{Z}_2^{\mathcal{C}}$ symmetry fugacities must be specified separately. The orthogonal dualities (3.4) and (3.11) reproduce known results [18–20], while the symplectic dualities (3.2) and (3.13) are, to the best of the author's knowledge, derived here for the first time. These two dualities form elementary building blocks: any duality sequence in a linear or circular CSM quiver arising from a brane configuration is expected to decompose into them. However, direct validation in longer quivers remains computationally out of reach.

The second main result is the development of a magnetic quiver framework for orthosymplectic $\mathcal{N} \geqslant 3$ CSM quivers (Section 4.2). The construction proceeds directly from the brane system via 5-brane moves and D3 creation/annihilation, and features two notable aspects:

- In general, multiple suitable phases of a given brane system may exist, each leading to a different magnetic quiver (see Figures 14 and 16). These quivers, though distinct, are expected to encode the maximal branch geometry for specific (possibly distinct) global forms of the original gauge group — providing an internal consistency check.

- Using the symmetry maps from the basic dualities, a fugacity map can be established between the CSM theory and its magnetic quiver(s). This renders the magnetic quiver partially sensitive to the global form of the CSM gauge group. In the $\mathcal{N} = 4$ setting, this can be (and has been) tested via Hilbert series limits of the supersymmetric index for small CSM quivers (up to three nodes).

**Open questions.** A distinctive feature of magnetic quivers for 3d orthosymplectic Chern-Simons matter theories is the availability of exact operator counting via supersymmetric indices and Hilbert series. This makes it possible, and indeed necessary, to explicitly test the proposed magnetic quivers. Such cross-checks are not typically feasible for magnetic

quivers of higher-dimensional SCFTs. While the approach has been validated for theories with a small number of gauge nodes, its extension to larger quivers presents new challenges.

In particular, the expected fugacity maps become increasingly intricate and raise foundational questions about the relationship between the global form of the CSM gauge group and that of the magnetic quiver required to describe a given maximal branch. Future work must address questions such as: What aspects of the maximal branch geometry are captured by a magnetic quiver? When multiple magnetic quivers arise from different brane phases, what information is shared, and what distinguishes them?

These questions echo long-standing ambiguities in 3d mirror symmetry for theories with orthogonal and symplectic gauge nodes (see, for instance, [35–38]), many of which similarly trace back to subtleties in global structure and orientifold effects.

**Higgsing.** Access to both magnetic quivers and brane constructions typically enables further analysis of maximal branch geometries. In the unitary CSM case, renormalisation group (RG) flows along the maximal branch can be studied through brane moves and through decay and fission processes [54] of the associated magnetic quivers. For the orthosymplectic CSM theories considered here, the resulting magnetic quivers are of the $T_\rho^\sigma[G]$ type [23], with $G$ in the $B$, $C$, or $D$ series. In such cases, RG flows are expected to follow the dominance ordering of partitions $\rho$ and $\sigma$, cf. [23, 55].

However, $T_\rho^\sigma[G]$ theories with orthogonal and symplectic groups are known to exhibit "bad" gauge nodes. This complication is anticipated to arise in the present context as well. It is therefore natural to ask: do "bad" nodes occur in the Higgsing flows of orthosymplectic CSM magnetic quivers? If so, to what extent do they appear, and what are their implications for interpreting the associated moduli spaces?

Some preliminary results on orthosymplectic decay and fission in magnetic quivers are available [56], but a more systematic analysis is required to clarify the structure of RG flows in orthosymplectic CSM theories.

**Other classes of CSM theories.** Naturally, many 3d $\mathcal{N} \geqslant 3$ Chern–Simons matter theories lie beyond the scope of the present work. These may be grouped into two broad classes: (i) CSM quiver theories arising as world-volume theories of D3–NS5–$(1, \kappa)$ brane systems in the presence of orientifold and/or orbifold 5-planes; (ii) more exotic CSM theories that cannot be realised in known brane constructions, such as those built from $T_N$ building blocks [27, 57, 58].

Class (i) appears to be within reach of the magnetic quiver framework in the near future, with related studies already initiated in [33, 59]. Class (ii), while more speculative, points toward a broader landscape of non-Lagrangian 3d CSM theories, whose structure remains largely unexplored and could motivate the development of new magnetic quiver techniques.

**Higher symmetries and symmetry webs.** Orthosymplectic CSM quivers with alternating $\mathrm{SO}(2n_i) \times \mathrm{USp}(2m_i)$ gauge nodes exhibit a $\mathbb{Z}_2$ 1-form symmetry, corresponding to the diagonal subgroup of the center symmetries of the individual gauge factors. In this work, the 1-form symmetry remains ungauged. However, it is well known that gauging such a symmetry in $2 + 1$ dimensions introduces a dual $\mathbb{Z}_2$ 0-form symmetry, which may non-trivially extend the $\mathbb{Z}_2^{\mathcal{C}} \times \mathbb{Z}_2^{\mathcal{M}}$-type global symmetries [37, 60, 61].

It would be natural to extend the present analysis to include gauged 1-form symmetries and to investigate how this modifies the associated magnetic quivers. This may involve the use of refined indices and could build upon prior results on 1-form symmetry gauging in orthosymplectic magnetic quivers [37, 43, 62].

## Acknowledgments

The authors would like to thank Amihay Hanany, William Harding, and Noppadol Meka­reeya for useful discussions. The work of FM, SMS and MS is supported by the Austrian Science Fund (FWF), START project "Phases of quantum field theories: symmetries and vacua" STA 73-N [grant DOI: 10.55776/STA73]. FM, SMS and MS also acknowledge sup­port from the Faculty of Physics, University of Vienna. SMS acknowledges the financial support by the Vienna Doctoral School in Physics (VDSP). MS gratefully acknowledges the Simons Center for Geometry and Physics, Stony Brook University for the hospitality and the partial support during various stages of this work at the Workshop "Symplectic Singularities, Supersymmetric QFT, and Geometric Representation Theory" (Mar 31 - Apr 4, 2025) and the 2025 Simons Physics Summer Workshop. Some results of this work have been presented prior publication at the "Quivers, Symmetries and SCFTs" workshop at ICTP Trieste (Sep 1-5, 2025). The authors acknowledge partial financial support and are grateful for the stimulating environment during the last stage of this work.

## A  Background material

### A.1  Brane configurations

The brane realisation of 3d $\mathcal{N} \geqslant 3$ CSM SCFTs in Type IIB superstring theory involves D3 branes, NS5 branes, and (one or several) $(1, \kappa)$ 5-branes; see for instance [5, 7, 53]. Orientifold 3-planes are included to realise orthogonal and symplectic gauge groups.

- D3s extend along directions 0123, with the 3 direction compact. The O3 planes span the same directions as the D3 branes.

- NS5s span directions 012789.

- $(1, \kappa)$ 5-branes span direction $012[4, 7]_\theta [5, 8]_\theta [6, 9]_\theta$, with $[i, j]_\theta$ denotes the tilted direction $\cos\theta x_i + \sin\theta x_j$ in the $(x_i, x_j)$ plane. The angle $\theta$ is a fixed function of $\kappa$ to preserve $\mathcal{N} = 3$ supersymmetry [11, 14]: $\tan\theta = \kappa$.

| brane | 0 | 1 | 2 | 3 | 4 | 5 | 6 | 7 | 8 | 9 |
|---|---|---|---|---|---|---|---|---|---|---|
| D3 ( − ) | × | × | × | × | | | | | | |
| NS5 ( | ) | × | × | × | | × | × | × | | | |
| $(1, \kappa)$-5 ( ╱ ) | × | × | × | | | $[4, 7]_\theta [5, 8]_\theta [6, 9]_\theta$ | | | | |

**Table 2:** Space-time occupation and notation for CSM brane systems. For linear CSM quiver theories, the $x^3$ direction is non-compact, but the D3 segments are finite in $x^3$ as they end on the 5-branes. For circular CSM quiver theories, the $x^3$ direction is compactified to a circle.

Next, the CMS quiver theory to a brane configuration of a sequence of NS5s and $(1, \kappa)$ 5-branes along the $x^3$ direction is specified as shown in Table 3.

### A.2  Brane creation and annihilation

Similar to the brane creation/annihilation effect [15] of an NS5 brane moving through a D5 brane, there is exists D3 brane creation/annihilation whenever an NS5 moves through a $(1, \kappa)$ 5-brane [11]. In the presence of O3 orientifolds, the brane creation/annihilation process needs to be modified to take the O3 charges into account. In units of the physical D3 branes, the orientifold charges are as follows [39, 63]:

$$\text{charge}(O3^\pm) = \pm\frac{1}{4}, \quad \text{charge}(\widetilde{O3}^-) = \frac{1}{2} - \frac{1}{4}, \quad \text{charge}(\widetilde{O3}^+) = \frac{1}{4}. \qquad (A.1)$$

| Supermultiplet | Brane configuration | Quiver |
|---|---|---|
| $\mathcal{N} = 4$ vector multiplet |  | $\bigcirc$ G($N$) |
| $\mathcal{N} = 4$ twisted vector multiplet |  $(1, \kappa)$ | $\bigcirc$ G($N$) |
| $\mathcal{N} = 2$ vector multiplet with CS-level $+\kappa$ |  $(1, \kappa)$ | $+\kappa$ $\bigcirc$ G($N$) |
| $\mathcal{N} = 2$ vector multiplet with CS-level $-\kappa$ |  $(1, \kappa)$ | $-\kappa$ $\bigcirc$ G($N$) |
| $\mathcal{N} = 4$ bifundamental (half-)hypermultiplet |  | $)$———$($ |
| $\mathcal{N} = 4$ bifundamental twisted (half-)hypermultiplet |  $(1, \kappa)$ | $)$———$($ |
| $\mathcal{N} = 4$ fundamental (half-)hypermultiplet |  | $)$——$\square$ 1 |
| $\mathcal{N} = 4$ fundamental twisted (half-)hypermultiplet |  $(1, \kappa)$ | $)$——$\square$ 1 |

**Table 3:** Conventions for the 3d multiplets that compose the D3 world-volume theory of the Type IIB brane systems involving D3 brane in between 5-branes. The branes follow the convention of Table 3. Here, the black horizontal lines are assumed to be a stack of $N$ D3s branes, while the 5-branes appear as single branes. The gray line represents the possible presence of a O3 plane. The wiggly lines represent fundamental strings, from which the world-volume theory is deduced. The quiver notation is given with respect to 3d $\mathcal{N} = 4$. When a CS-level is present, the gauge node represents only a 3d $\mathcal{N} = 2$ vector multiplet, without the $\mathcal{N} = 2$ adjoint chiral that is integrated out by the CS superpotential term. If no O3 plane is inserted in the system, one has G = U, while in the presence of an O3 plane, the group G reads according to Table 1.

Moreover, recall that whenever a half 5-brane crosses an orientifold 3-plan, the type changes depending in the charges as follows [39, 42, 63, 64]:

- O3$^{\pm}$ (or $\widetilde{\text{O3}}^{\pm}$) becomes O3$^{\mp}$ (or $\widetilde{\text{O3}}^{\mp}$) when passing through a half NS5 brane;

- O3$^{\pm}$ becomes $\widetilde{\text{O3}}^{\pm}$ (and viceversa) when passing through a half D5 brane.

Using these rules, one can obtain the O3 planes transformation rules when passing through a $(p, q)$ 5-brane, which carries both NS5 and D5 charges.

To derive the number of created/annihilated D3 branes, one defines linking numbers for half NS5s and half D5s, see [15]. Requiring the linking numbers to be the same before and after the half NS5 moves through the half D5 leads to the D3 brane creation/annihilation, see [29, App. A.1] for a summary.

To extend this reasoning to the case of an NS5 passing through an $(1, q)$ 5-brane in the presence of O3 planes, requires to also take into account what happens if two NS5 brane pass each other. As shown in [8, Sec. 3], there are the following cases:

- If the orientifold between the two half NS5 branes is an O3$^-$, a single D3 is created.

- If the orientifold is an O3$^+$, a single D3 is annihilated.

- If the orientifold is an $\widetilde{\text{O3}}^-$, no brane is created or annihilated.

Again, this follows from linking number conservation.

Combining both effects, on readily deduced the brane creation/annihilation for an NS5 brane passing through an $(1, q)$ 5-brane with an O3 in between. Due to the changes of the O3s, depending in $q$ being even or odd, the results need to be separated for $q = 2k$ and $q = 2k + 1$, see Figure 17.

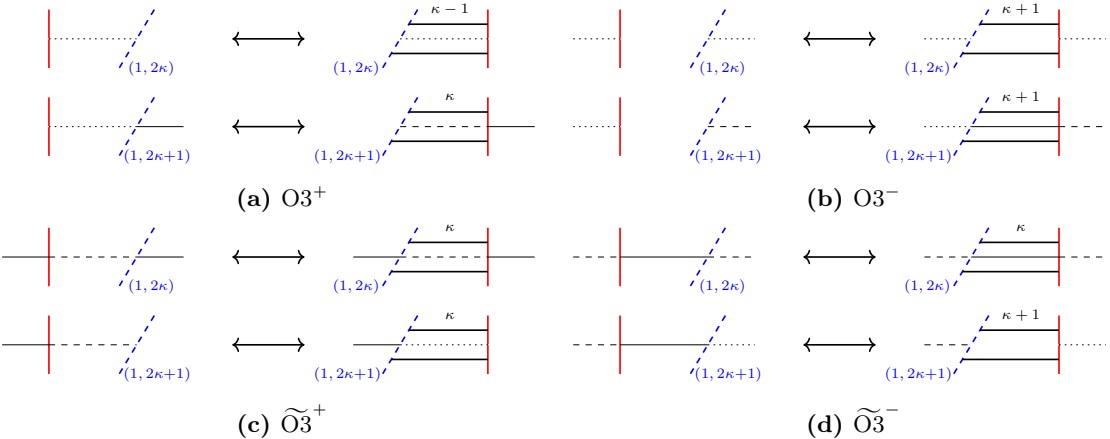

**Figure 17:** Brane creation and annihilation for an NS5 moving through an $(1, q)$ 5-brane with various types of O3 orientifolds in between.

## A.3   Good, bad, and ugly

An important criterion for 3d $\mathcal{N} = 4$ SCFTs is whether they are *good*, *bad*, or *ugly* [23], which indicates whether the R-charges in the UV and IR coincide or not. For 3d $\mathcal{N} = 4$ non-CS gauge theories, a single $G$ gauge node is denoted as *good* if the following holds:

- A U$(N)$ gauge group coupled to $N_f$ fundamental hypermultiplets is

$$good \text{ if } N_f \geqslant 2N\,, \qquad \text{and } balanced \text{ if } N_f = 2N\,. \tag{A.2a}$$

- An Sp$(N) \equiv$ USp$(2N)$ gauge group coupled to fundamental hypermultiplets with SO$(2N_f)$ (or O$(2N_f)$) flavour symmetry is called

$$good \text{ if } N_f \geqslant 2N + 1\,, \qquad \text{and } balanced \text{ if } N_f = 2N + 1\,. \tag{A.2b}$$

- An SO$(N)$ (or O$(N)$) gauge theory coupled to fundamental hypermultiplets with Sp$(N_f)$ flavour group is called

$$good \text{ if } N_f \geqslant N - 1\,, \qquad \text{and } balanced \text{ if } N_f = N - 1\,. \tag{A.2c}$$

The *balance* condition is particularly relevant for symmetry due to monopole operators, cf. [23, 65, 66].

Determining whether a 3d $\mathcal{N} = 4$ Chern–Simons matter theories is *good* is less straightforward. Some partial results have been obtained in specific models in [67, 68], while the magnetic quiver approach of [28] allows for a systematic analysis based on (A.2).

## A.4 Example operator spectroscopy

As with each duality, it is insightful to keep track of the mapping between gauge invariant operators. This is also useful to understand why the magnetic quiver method applies.

**Unitary CSM quivers.** To illustrate the point, consider how some gauge invariant operators map across the duality in Figure 2. One can introduce the following notation:

- $Q_{i,j}$ denotes the $\mathcal{N} = 4$ hypermultiplet between the $i$-th and the $j$-th gauge groups. Using $x$ and $t$ as the fugacities for the R-symmetry and the axial symmetry respectively, the $\mathcal{N} = 4$ un-twisted hypermultiplet (red edge in the figures) has charges $x^{1/2}t^{+1}$, while the $\mathcal{N} = 4$ twisted hypermultiplet (blue edge in the figures) has $x^{1/2}t^{-1}$.

- $A_i$ denotes the $\mathcal{N} = 2$ adjoint chiral on the $i$-th Sp/SO gauge group (if it is $\mathcal{N} = 4$). Due to the $\mathcal{N} = 4$ superpotential term $Q_{j,i}A_iQ_{i,\ell}$, the charge of $A_i$ is

$$[A_i] = xt^{+2} \quad \text{if} \quad [Q_{j,i}] = [Q_{i,\ell}] = x^{1/2}t^{-1}, \tag{A.3a}$$

$$[A_i] = xt^{-2} \quad \text{if} \quad [Q_{j,i}] = [Q_{i,\ell}] = x^{1/2}t^{+1}. \tag{A.3b}$$

- $\mathfrak{M}_{(I)}^{(\vec{f}_1,\vec{f}_2,\dots,\vec{f}_n)}$ denotes the monopole operator of theory $(I)$ with fluxes $\vec{f}_1,\vec{f}_2,\dots,\vec{f}_n$ under the gauge groups $1,2,\dots,n$ respectively.

Then, considering the theories $(0)$, $(1)$ and $(2)$ in Figure 2, one can establish the following map for the NS5-branch gauge-invariant operators (the $(1,\kappa)$-branch is instead trivial):

$$\left\{\mathfrak{M}_{(0)}^{(\{+,0,\dots,0\},\{+,0,\dots,0\})}\big(\widetilde{Q}_{1,2}\big)^{|\kappa|},\ \mathfrak{M}_{(0)}^{(\{-,0,\dots,0\},\{-,0,\dots,0\})}\big(\widetilde{Q}_{1,2}\big)^{|\kappa|}\right\}$$
$$\updownarrow$$
$$\left\{\mathfrak{M}_{(1)}^{(\{+,0,\dots,0\},\{0,0,\dots,0\})},\ \mathfrak{M}_{(1)}^{(\{-,0,\dots,0\},\{0,0,\dots,0\})}\right\} \tag{A.4}$$
$$\updownarrow$$
$$\left\{\mathfrak{M}_{(2)}^{(\{0,0,\dots,0\},\{+,0,\dots,0\})},\ \mathfrak{M}_{(2)}^{(\{0,0,\dots,0\},\{-,0,\dots,0\})}\right\}.$$

If $N = M$ and $\kappa = 2N$, the topological symmetry enhances to SU(2) and one can map across the duality its adjoint representation as follows:

$$\left\{\mathfrak{M}_{(0)}^{(\{+,0,\dots,0\},\{+,0,\dots,0\})}\big(\widetilde{Q}_{1,2}\big)^{|\kappa|},\ \mathfrak{M}_{(0)}^{(\{-,0,\dots,0\},\{-,0,\dots,0\})}\big(\widetilde{Q}_{1,2}\big)^{|\kappa|},\ \text{Tr}\left[Q_{1,2}\widetilde{Q}_{1,2}\right]\right\}$$
$$\updownarrow$$
$$\left\{\mathfrak{M}_{(1)}^{(\{+,0,\dots,0\},\{0,0,\dots,0\})},\ \mathfrak{M}_{(1)}^{(\{-,0,\dots,0\},\{0,0,\dots,0\})},\ \text{Tr}\left[A_1\right]\right\} \tag{A.5}$$
$$\updownarrow$$
$$\left\{\mathfrak{M}_{(2)}^{(\{0,0,\dots,0\},\{+,0,\dots,0\})},\ \mathfrak{M}_{(2)}^{(\{0,0,\dots,0\},\{-,0,\dots,0\})},\ \text{Tr}\left[A_2\right]\right\}.$$

This then sheds light on why the magnetic quiver, derived as in Figure 8, works. It isolates exactly the maximal branch operators, as detailed in [28].

**Orthosymplectic CSM quiver.** Also for orthosymplectic CSM theories is a mapping of gauge invariant operators across the duality insightful. Using the notation introduced above, consider the theories $(0)$, $(1)$ and $(2)$ for the examples in Figure 5a, for which

one can establish the following map for the NS5-branch gauge-invariant operators (the $(1,\kappa)$-branch is trivial here):

$$\left\{ \mathfrak{M}_{(0)}^{(\{+,0,\dots,0\},\{+,0,\dots,0\})} \left(\tilde{Q}_{1,2}\right)^{2|\kappa|} \right\}$$

$$\updownarrow$$

$$\left\{ \mathfrak{M}_{(1)}^{(\{+,0,\dots,0\},\{0,0,\dots,0\})} \right\} \tag{A.6}$$

$$\updownarrow$$

$$\left\{ \mathfrak{M}_{(2)}^{(\{0,0,\dots,0\},\{+,0,\dots,0\})} \right\} .$$

If $\kappa = 2N$, one has an enhanced $SO(2)$ global symmetry and hence finds the above operators at order $xt^{-2}$ in the index.

Again, this map illustrates why the magnetic quivers of Figure 9 yield the correct maximal branch operator count.

## A.5 Example fugacity map

To illustrate the general scheme of the fugacity map of Section 3.3.3, consider the following brane configuration

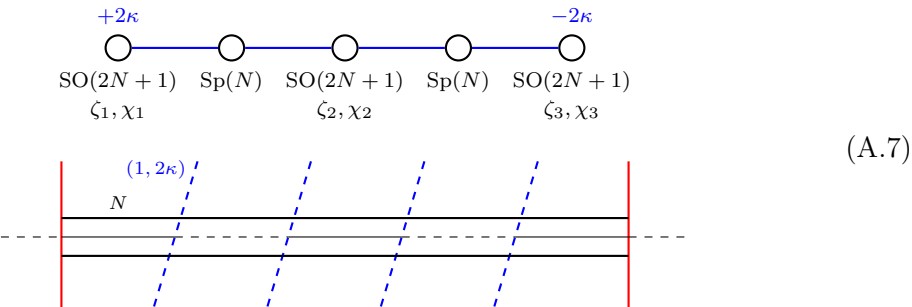

$$\tag{A.7}$$

which realises a 3d $\mathcal{N} = 4$ CSM theory with three orthogonal gauge nodes, each with fugacities $(\zeta_i, \chi_i)$. After a sequence of orthogonal (3.4), (3.11) and symplectic (3.2), (3.13) brane moves, one arrives at a equivalent brane system in a suitable phase

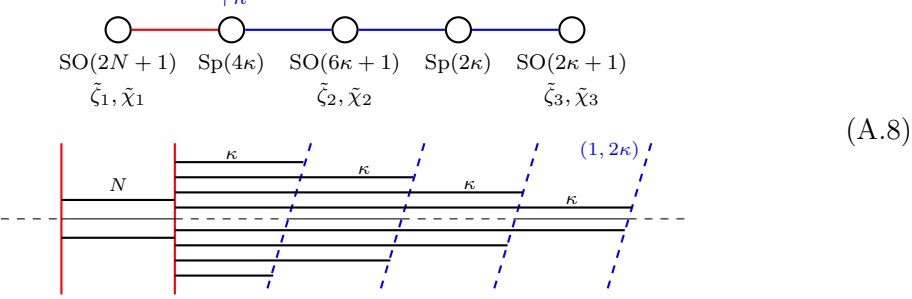

$$\tag{A.8}$$

from which one derives the following $\mathsf{MQ}_{\mathrm{NS5}}$:

$$C_{4|\kappa|}$$
$$\square$$
$$|$$
$$\bigcirc$$
$$SO(2N+1)$$
$$\zeta, \chi$$

$$\tag{A.9}$$

To map the Hilbert series of (A.9) with the $\mathcal{B}_{\mathrm{NS5}}$ limit of the index of (A.7), one utilises the following map: firstly, identify $(\zeta, \chi)$ of (A.9) with the fugacities $(\tilde{\zeta}_1, \tilde{\chi}_1)$ of the dual

setup (A.8). Secondly, relate the fugacities of (A.8) with that of (A.7), via sequential application of the maps (3.11) and (3.13). One finds

$$\zeta \equiv \tilde{\zeta}_1 = \zeta_1\zeta_2\zeta_3\chi_1\chi_3 \,, \quad \chi \equiv \tilde{\chi}_1 = \chi_1 \,. \tag{A.10}$$

# B  Index expansions

In this appendix have been collected some examples of the expanded 3d $\mathcal{N} = 2$ superconformal index for some of the theories discussed in the main text. The conventions followed are those of [20, 37].

## B.1  Orthosymplectic CSM dualities

In the following tables are showcased the index expansions for some examples of the 2 and 3 nodes CSM theories considered in Section 3. In particular, the reported results are for the starting theories (dubbed as (0)) and for their dual (1), obtained by implementing the Sp-type duality. This provides evidence for the new fugacity mapping proposed in Section 3 (see (3.2), (3.7) and (3.13)).

| $\{N, M\}$ | $\kappa$ | Index |
|---|---|---|
| $\{1,1\}$ | 2 | $\mathcal{I}^{(0)}(\chi_0, \zeta_0) = 1$ |
| | | $\quad + x^1(t^{-2}(\zeta_0\chi_0 + \zeta_0 + \chi_0))$ |
| | | $\quad + x^2(-\zeta_0\chi_0 - \zeta_0 - \chi_0 - 1 + t^{-4}(\zeta_0\chi_0 + \zeta_0 + \chi_0 + 2))$ |
| | | $\quad + x^3(t^2 + t^{-6}(2\zeta_0\chi_0 + 2\zeta_0 + 2\chi_0 + 1) + t^{-2}(-\zeta_0\chi_0 - \zeta_0 - \chi_0 - 1))$ |
| | | $\quad + x^4(-\zeta_0\chi_0 - \zeta_0 - \chi_0 - 2 + t^{-8}(2\zeta_0\chi_0 + 2\zeta_0 + 2\chi_0 + 3)$ |
| | | $\qquad + t^{-4}(-\zeta_0\chi_0 - \zeta_0 - \chi_0 - 1))$ |
| | | $\quad + x^5(t^2(\zeta_0\chi_0 + \zeta_0 + \chi_0 + 1) + t^{-10}(3\zeta_0\chi_0 + 3\zeta_0 + 3\chi_0 + 2)$ |
| | | $\quad + t^{-6}(-\zeta_0\chi_0 - \zeta_0 - \chi_0 - 1) + t^{-2}(-2\zeta_0\chi_0 - 2\zeta_0 - 2\chi_0 - 2))$ |
| | | $\quad + O(x^6)$ |
| | | $= \mathcal{I}^{(1)}(\chi_1, \zeta_1)$ |
| $\{1,1\}$ | 3 | $\mathcal{I}^{(0)}(\chi_0, \zeta_0) = 1$ |
| | | $\quad + x^1(t^{-2}\chi_0)$ |
| | | $\quad + x^2(t^{-4}(\zeta_0\chi_0 + \zeta_0 + 1) - \chi_0 - 1)$ |
| | | $\quad + x^3(t^{-6}(\zeta_0\chi_0 + \zeta_0 + \chi_0) + t^{-2}(-\zeta_0\chi_0 - \zeta_0) + t^2)$ |
| | | $\quad + x^4(t^{-8}(\zeta_0\chi_0 + \zeta_0 + \chi_0 + 2) + t^{-4}(-\zeta_0\chi_0 - \zeta_0) - 2)$ |
| | | $\quad + O(x^5)$ |
| | | $= \mathcal{I}^{(1)}(\chi_1, \zeta_1)$ |
| $\{1,2\}$ | 3 | $\mathcal{I}^{(0)}(\chi_0, \zeta_0) = 1$ |
| | | $\quad + x^1(t^{-2}(\zeta_0\chi_0 + \zeta_0 + \chi_0))$ |
| | | $\quad + x^2(t^{-4}(\zeta_0\chi_0 + \zeta_0 + \chi_0 + 2) - \zeta_0\chi_0 - \zeta_0 - \chi_0 - 1)$ |
| | | $\quad + x^3(t^{-6}(2\zeta_0\chi_0 + 2\zeta_0 + 2\chi_0 + 1) + t^{-2}(-\zeta_0\chi_0 - \zeta_0 - \chi_0 - 1) + t^2)$ |
| | | $\quad + x^4(t^{-8}(2\zeta_0\chi_0 + 2\zeta_0 + 2\chi_0 + 3) + t^{-4}(-\zeta_0\chi_0 - \zeta_0 - \chi_0 - 1) - 2)$ |
| | | $\quad + O(x^5)$ |
| | | $= \mathcal{I}^{(1)}(\chi_1, \zeta_1)$ |
| $\{1,2\}$ | 4 | $\mathcal{I}^{(0)}(\chi_0, \zeta_0) = 1$ |
| | | $\quad + x^1(t^{-2}\chi_0)$ |
| | | $\quad + x^2(t^{-4}(\zeta_0\chi_0 + \zeta_0 + 1) - \chi_0 - 1)$ |
| | | $\quad + x^3(t^{-6}(\zeta_0\chi_0 + \zeta_0 + \chi_0) + t^{-2}(-\zeta_0\chi_0 - \zeta_0) + t^2)$ |
| | | $\quad + x^4(t^{-8}(\zeta_0\chi_0 + \zeta_0 + \chi_0 + 2) + t^{-4}(-\zeta_0\chi_0 - \zeta_0) + \chi_0 - 2)$ |
| | | $\quad + O(x^5)$ |

$$= \mathcal{I}^{(1)}(\chi_1, \zeta_1)$$

| $\{N,M\}$ | $\kappa$ | |
|---|---|---|
| $\{2,2\}$ | 4 | $\mathcal{I}^{(0)}(\chi_0, \zeta_0) = 1$ |
| | | $\quad + x^1(t^{-2}\zeta_0\chi_0)$ |
| | | $\quad + x^2(t^{-4}(2 + \zeta_0 + \chi_0 + \zeta_0\chi_0) - \zeta_0\chi_0 - 1)$ |
| | | $\quad + x^3(t^{-6}(1 + \zeta_0 + \chi_0 + 3\zeta_0\chi_0) + t^{-2}(-1 - \zeta_0 - \chi_0 - \zeta_0\chi_0) + t^2)$ |
| | | $\quad + O(x^4)$ |
| | | |
| | | $\mathcal{I}^{(1)}(\chi_1, \zeta_1) = 1$ |
| | | $\quad + x^1(t^{-2}\zeta_1)$ |
| | | $\quad + x^2(t^{-4}(\zeta_1\chi_1 + \zeta_1 + \chi_1 + 2) - \zeta_1 - 1)$ |
| | | $\quad + x^3(t^{-6}(\zeta_1\chi_1 + 3\zeta_1 + \chi_1 + 1) + t^{-2}(-\zeta_1\chi_1 - \zeta_1 - \chi_1 - 1) + t^2)$ |
| | | $\quad + O(x^4)$ |
| $\{2,2\}$ | 5 | $\mathcal{I}^{(0)}(\chi_0, \zeta_0) = 1$ |
| | | $\quad + x^2(t^{-4}(\zeta_0\chi_0 + \chi_0 + 1) - 1)$ |
| | | $\quad + x^3(t^{-6}(\zeta_0\chi_0 + \zeta_0) + t^{-2}(-\zeta_0\chi_0 - \chi_0) + t^2)$ |
| | | $\quad + O(x^4)$ |
| | | |
| | | $\mathcal{I}^{(1)}(\chi_1, \zeta_1) = 1$ |
| | | $\quad + x^2(t^{-4}(\zeta_1 + \chi_1 + 1) - 1)$ |
| | | $\quad + x^3(t^{-6}(\zeta_1\chi_1 + \zeta_1) + t^{-2}(-\zeta_1 - \chi_1) + t^2)$ |
| | | $\quad + O(x^4)$ |

**Table 4:** Expanded index for the linear $\mathrm{SO}(2N)_{+2\kappa} \times \mathrm{Sp}(M)_{-\kappa}$ CSM theory (0) and for its dual (1) in Figure 5a, for some values of $N$, $M$ and $|\kappa| \geqslant N + M$ (for which the theory is good, see Section 4.2.1). Notice that in some cases the indices of the dual theories match both with and without the fugacity mapping (3.2).

| $\{N,M\}$ | $\kappa$ | Index of theories (0) and (1) |
|---|---|---|
| $\{1,1\}$ | 2 | $\mathcal{I}^{(0)}\big(\chi_0^{(L)}, \zeta_0^{(L)}, \chi_0^{(R)}, \zeta_0^{(L)}\big) = 1$ |
| | | $\quad + x^1(t^{-2}\chi_0^{(L)} + t^2\chi_0^{(R)})$ |
| | | $\quad + x^2(t^{-4}(\zeta_0^{(L)}\chi_0^{(L)}\chi_0^{(R)} + \zeta_0^{(L)}\chi_0^{(R)} + 1) + t^4(\zeta_0^{(R)}\chi_0^{(L)}\chi_0^{(R)} + \zeta_0^{(R)}\chi_0^{(L)} + 1)$ |
| | | $\qquad + \chi_0^{(L)}\chi_0^{(R)} - \chi_0^{(L)} - \chi_0^{(R)} - 1)$ |
| | | $\quad + x^3(t^{-6}(\zeta_0^{(L)}\chi_0^{(L)}\chi_0^{(R)} + \zeta_0^{(L)}\chi_0^{(R)} + \chi_0^{(L)})$ |
| | | $\qquad + t^{-2}(-\zeta_0^{(L)}\chi_0^{(L)}\chi_0^{(R)} - \zeta_0^{(L)}\chi_0^{(R)} - \chi_0^{(L)}\chi_0^{(R)})$ |
| | | $\qquad + t^2(-\zeta_0^{(R)}\chi_0^{(L)}\chi_0^{(R)} - \zeta_0^{(R)}\chi_0^{(L)} - \chi_0^{(L)}\chi_0^{(R)})$ |
| | | $\qquad + t^6(\zeta_0^{(R)}\chi_0^{(L)}\chi_0^{(R)} + \zeta_0^{(R)}\chi_0^{(L)} + \chi_0^{(R)}))$ |
| | | $\quad + O(x^4)$ |
| | | |
| | | $\mathcal{I}^{(1)}\big(\chi_1^{(L)}, \zeta_1^{(L)}, \chi_1^{(R)}, \zeta_1^{(L)}\big) = 1$ |
| | | $\quad + x^1(t^{-2}\chi_1^{(L)} + t^2\chi_1^{(R)})$ |
| | | $\quad + x^2(t^{-4}(\zeta_1^{(L)}\chi_1^{(L)} + \zeta_1^{(L)} + 1) + t^4(\zeta_1^{(R)}\chi_1^{(R)} + \zeta_1^{(R)} + 1)$ |
| | | $\qquad + \chi_1^{(L)}\chi_1^{(R)} - \chi_1^{(L)} - \chi_1^{(R)} - 1)$ |
| | | $\quad + x^3(t^{-6}(\zeta_1^{(L)}\chi_1^{(L)} + \zeta_1^{(L)} + \chi_1^{(L)}) + t^{-2}(-\zeta_1^{(L)}\chi_1^{(L)} - \zeta_1^{(L)} - \chi_1^{(L)}\chi_1^{(R)})$ |
| | | $\qquad + t^2(-\zeta_1^{(R)}\chi_1^{(R)} - \zeta_1^{(R)} - \chi_1^{(L)}\chi_1^{(R)}) + t^6(\zeta_1^{(R)}\chi_1^{(R)} + \zeta_1^{(R)} + \chi_1^{(R)}))$ |
| | | $\quad + O(x^4)$ |
| $\{1,1\}$ | 3 | $\mathcal{I}^{(0)}\big(\chi_0^{(L)}, \zeta_0^{(L)}, \chi_0^{(R)}, \zeta_0^{(L)}\big) = 1$ |
| | | $\quad + x^1(t^{-2}\chi_0^{(L)} + t^2\chi_0^{(R)})$ |
| | | $\quad + x^2(t^{-4} + t^4 + \chi_0^{(L)}\chi_0^{(R)} - \chi_0^{(L)} - \chi_0^{(R)} - 1)$ |
| | | $\quad + x^3(t^{-6}(\zeta_0^{(L)}\chi_0^{(L)}\chi_0^{(R)} + \zeta_0^{(L)}\chi_0^{(R)} + \chi_0^{(L)}) + t^{-2}(-\chi_0^{(L)}\chi_0^{(R)})$ |
| | | $\qquad + t^6(\zeta_0^{(R)}\chi_0^{(L)}\chi_0^{(R)} + \zeta_0^{(R)}\chi_0^{(L)} + \chi_0^{(R)}) - t^2\chi_0^{(L)}\chi_0^{(R)})$ |
| | | $\quad + O(x^3)$ |

$$\mathcal{I}^{(1)}\big(\chi_1^{(L)},\zeta_1^{(L)},\chi_1^{(R)},\zeta_1^{(L)}\big) = 1$$
$$+ x^1(t^{-2}\chi_1^{(L)} + t^2\chi_1^{(R)})$$
$$+ x^2(t^{-4} + t^4 + \chi_1^{(L)}\chi_1^{(R)} - \chi_1^{(L)} - \chi_1^{(R)} - 1)$$
$$+ x^3(t^{-6}(\zeta_1^{(L)}\chi_1^{(L)} + \zeta_1^{(L)} + \chi_1^{(L)}) + t^{-2}(-\chi_1^{(L)}\chi_1^{(R)}) + t^6(\zeta_1^{(R)}\chi_1^{(R)} + \zeta_1^{(R)} + \chi_1^{(R)})$$
$$-t^2\chi_1^{(L)}\chi_1^{(R)})$$
$$+ O(x^3)$$

| $\{N,M\}$ | $\kappa$ | |
|---|---|---|
| $\{1,2\}$ | $2$ | |

$$\mathcal{I}^{(0)}\big(\chi_0^{(L)},\zeta_0^{(L)},\chi_0^{(R)},\zeta_0^{(L)}\big) = 1$$
$$+ x^1(t^{-2}(\zeta_0^{(L)}\chi_0^{(L)}\chi_0^{(R)} + \zeta_0^{(L)}\chi_0^{(R)} + \chi_0^{(L)}) + t^2(\zeta_0^{(R)}\chi_0^{(L)}\chi_0^{(R)} + \zeta_0^{(R)}\chi_0^{(L)} + \chi_0^{(R)}))$$
$$+ x^2(t^{-4}(\zeta_0^{(L)}\chi_0^{(L)}\chi_0^{(R)} + \zeta_0^{(L)}\chi_0^{(R)} + \chi_0^{(L)} + 2)$$
$$+t^4(\zeta_0^{(R)}\chi_0^{(L)}\chi_0^{(R)} + \zeta_0^{(R)}\chi_0^{(L)} + \chi_0^{(R)} + 2) + \zeta_0^{(L)}\zeta_0^{(R)}\chi_0^{(L)}\chi_0^{(R)} + \zeta_0^{(L)}\zeta_0^{(R)}\chi_0^{(L)}$$
$$+\zeta_0^{(L)}\zeta_0^{(R)}\chi_0^{(R)} + \zeta_0^{(L)}\zeta_0^{(R)} - \zeta_0^{(L)}\chi_0^{(L)}\chi_0^{(R)} + \zeta_0^{(L)}\chi_0^{(L)} - \zeta_0^{(L)}\chi_0^{(R)} + \zeta_0^{(L)}$$
$$-\zeta_0^{(R)}\chi_0^{(L)}\chi_0^{(R)} - \zeta_0^{(R)}\chi_0^{(L)} + \zeta_0^{(R)}\chi_0^{(R)} + \zeta_0^{(R)} + \chi_0^{(L)}\chi_0^{(R)} - \chi_0^{(L)} - \chi_0^{(R)} - 1))$$
$$+ O(x^3)$$

$$\mathcal{I}^{(1)}\big(\chi_1^{(L)},\zeta_1^{(L)},\chi_1^{(R)},\zeta_1^{(L)}\big) = 1$$
$$+ x^1(t^{-2}(\zeta_1^{(L)}\chi_1^{(L)} + \zeta_1^{(L)} + \chi_1^{(L)}) + t^2(\zeta_1^{(R)}\chi_1^{(R)} + \zeta_1^{(R)} + \chi_1^{(R)}))$$
$$+ x^2(t^{-4}(\zeta_1^{(L)}\chi_1^{(L)} + \zeta_1^{(L)} + \chi_1^{(L)} + 2) + t^4(\zeta_1^{(R)}\chi_1^{(R)} + \zeta_1^{(R)} + \chi_1^{(R)} + 2)$$
$$+\zeta_1^{(L)}\zeta_1^{(R)}\chi_1^{(L)}\chi_1^{(R)} + \zeta_1^{(L)}\zeta_1^{(R)}\chi_1^{(L)} + \zeta_1^{(L)}\zeta_1^{(R)}\chi_1^{(R)} + \zeta_1^{(L)}\chi_1^{(R)} - \zeta_1^{(L)}$$
$$+\zeta_1^{(R)}\chi_1^{(L)}\chi_1^{(R)} + \zeta_1^{(R)}\chi_1^{(L)} - \zeta_1^{(R)}\chi_1^{(R)} - \zeta_1^{(R)} + \chi_1^{(L)}\chi_1^{(R)} - \chi_1^{(L)} - \chi_1^{(R)} - 1))$$
$$+ O(x^3)$$

**Table 5:** Expanded index for the linear $\mathrm{SO}(2N)_{+2\kappa} \times \mathrm{Sp}(M)_{-\kappa} \times \mathrm{SO}(2N)_{+2\kappa}$ CSM theory (0) and its dual (1) as in (3.12) for some values of $N$, $M$ and $|\kappa| \geqslant M$ (for which the theory is good, see Section 4.2.2). Notice that the indices of the dual theories match if (3.13) holds.

## B.2 Linear examples with 2 nodes

In the following tables the index expansions for some instances of the 2 nodes examples of Section 4.2.1 have been reported. In particular, in Table 6 one can find the expanded index and the Hilbert series for the linear $\mathrm{SO}(2N_1)_{+2\kappa} \times \mathrm{Sp}(N)_{-\kappa}$ CSM theory for some values of $N$, $M$ and $\kappa$, while in Table 7 the same is shown for the $\mathrm{SO}(2N + 1)_{+2\kappa} \times \mathrm{Sp}(M)_{-\kappa}$ CSM theory.

| $\{N,M\}$ | $\kappa$ | Index and Hilbert series for CSM $= \mathrm{SO}(2N)_{+2\kappa} \times \mathrm{Sp}(M)_{-\kappa}$ |
|---|---|---|
| $\{1,1\}$ | $2$ | |

$$\mathcal{I}(\mathrm{CSM}) = 1$$
$$+ x^1((\zeta\chi + \zeta + \chi)t^{-2})$$
$$+ x^2(-\zeta\chi - \zeta + (\zeta\chi + \zeta + \chi + 2)t^{-4} - \chi - 1)$$
$$+ x^3((2\zeta\chi + 2\zeta + 2\chi + 1)t^{-6} + (-\zeta\chi - \zeta - \chi - 1)t^{-2} + t^2)$$
$$+ x^4(-\zeta\chi - \zeta + (2\zeta\chi + 2\zeta + 2\chi + 3)t^{-8} + (-\zeta\chi - \zeta - \chi - 1)t^{-4} - \chi - 2)$$
$$+ x^5((3\zeta\chi + 3\zeta + 3\chi + 2)t^{-10} + (-\zeta\chi - \zeta - \chi - 1)t^{-6} + (\zeta\chi + \zeta + \chi + 1)t^2$$
$$+(-2\zeta\chi - 2\zeta - 2\chi - 2)t^{-2})$$
$$+ x^6(\zeta\chi + \zeta + (3\zeta\chi + 3\zeta + 3\chi + 4)t^{-12} + (-\zeta\chi - \zeta - \chi - 1)t^{-8}$$
$$+(-2\zeta\chi - 2\zeta - 2\chi - 2)t^{-4} + \chi + 1)$$
$$+ x^7((4\zeta\chi + 4\zeta + 4\chi + 3)t^{-14} + (-\zeta\chi - \zeta - \chi - 1)t^{-10}$$
$$+(-2\zeta\chi - 2\zeta - 2\chi - 2)t^{-6} + (-2\zeta\chi - 2\zeta - 2\chi - 2)t^{-2} + t^2)$$
$$+ x^8(4\zeta\chi + 4\zeta + (4\zeta\chi + 4\zeta + 4\chi + 5)t^{-16} + (-\zeta\chi - \zeta - \chi - 1)t^{-12}$$
$$+(-2\zeta\chi - 2\zeta - 2\chi - 2)t^{-8} + (-2\zeta\chi - 2\zeta - 2\chi - 2)t^{-4} + 4\chi + 2)$$
$$+ O(x^9)$$

$$\mathrm{HS}_{\mathrm{NS5}}(\mathrm{CSM}) = 1 + a(\zeta\chi + \zeta + \chi) + a^2(\zeta\chi + \zeta + \chi + 2) + a^3(2\zeta\chi + 2\zeta + 2\chi + 1)$$
$$+ a^4(2\zeta\chi + 2\zeta + 2\chi + 3) + a^5(3\zeta\chi + 3\zeta + 3\chi + 2) + a^6(3\zeta\chi + 3\zeta + 3\chi + 4)$$

$\qquad + a^7(4\zeta\chi + 4\zeta + 4\chi + 3) + a^8(4\zeta\chi + 4\zeta + 4\chi + 5) + O(a^{17/2}) + O(a^9)$

$\qquad \text{HS}_{(1,2\kappa)}(\text{CSM}) = 1 + O(b^9)$

| | | |
|---|---|---|
| $\{1,1\}$ | 3 | $\mathcal{I}(\text{CSM}) = 1$ |

$\qquad + x^1(t^{-2}\chi)$

$\qquad + x^2((\zeta\chi + \zeta + 1)t^{-4} - \chi - 1)$

$\qquad + x^3((\zeta\chi + \zeta + \chi)t^{-6} + (-\zeta\chi - \zeta)t^{-2} + t^2)$

$\qquad + x^4((\zeta\chi + \zeta + \chi + 2)t^{-8} + (-\zeta\chi - \zeta)t^{-4} - 2)$

$\qquad + x^5((\zeta\chi + \zeta + 2\chi + 1)t^{-10} + (-\chi - 1)t^{-6} + (-\zeta\chi - \zeta - 2\chi - 1)t^{-2})$

$\qquad + x^6(\zeta\chi + \zeta + (2\zeta\chi + 2\zeta + \chi + 2)t^{-12} + (-\chi - 1)t^{-8} + (-2\zeta\chi - 2\zeta)t^{-4}$

$\qquad\qquad + t^4 + 2\chi + 2)$

$\qquad + x^7((2\zeta\chi + 2\zeta + 2\chi + 1)t^{-14} + (-\zeta\chi - \zeta)t^{-10} + (-\zeta\chi - \zeta - \chi - 1)t^{-6}$

$\qquad\qquad + (\zeta\chi + \zeta - 3\chi - 2)t^{-2} + (-2\chi - 2)t^2)$

$\qquad + x^8((2\zeta\chi + 2\zeta + 2\chi + 3)t^{-16} + (-\zeta\chi - \zeta)t^{-12} + (-2\chi - 2)t^{-8}$

$\qquad\qquad + (-3\zeta\chi - 3\zeta - 1)t^{-4} + (\chi + 1)t^4 + 6\chi + 3)$

$\qquad + O(x^9)$

$\qquad \text{HS}_{\text{NS5}}(\text{CSM}) = 1 + a\chi + a^2(\zeta\chi + \zeta + 1) + a^3(\zeta\chi + \zeta + \chi) + a^4(\zeta\chi + \zeta + \chi + 2)$

$\qquad + a^5(\zeta\chi + \zeta + 2\chi + 1) + a^6(2\zeta\chi + 2\zeta + \chi + 2) + a^7(2\zeta\chi + 2\zeta + 2\chi + 1)$

$\qquad + a^8(2\zeta\chi + 2\zeta + 2\chi + 3) + O(a^9)$

$\qquad \text{HS}_{(1,2\kappa)}(\text{CSM}) = 1 + O(b^9)$

| | | |
|---|---|---|
| $\{1,1\}$ | 4 | $\mathcal{I}(\text{CSM}) = 1$ |

$\qquad + x^1(t^{-2}\chi)$

$\qquad + x^2(t^{-4} - \chi - 1)$

$\qquad + x^3(t^{-6}(\zeta\chi + \zeta + \chi) + t^2)$

$\qquad + x^4(t^{-8}(\zeta\chi + \zeta + 1) + t^{-4}(-\zeta\chi - \zeta) - 2)$

$\qquad + x^5(t^{-10}(\zeta\chi + \zeta + \chi) + t^{-6}(-\zeta\chi - \zeta) + t^{-2}(-2\chi - 1))$

$\qquad + x^6(t^{-12}(\zeta\chi + \zeta + \chi + 2) + t^{-4}(-\zeta\chi - \zeta) + t^4 + 2\chi + 2)$

$\qquad + x^7(t^{-14}(\zeta\chi + \zeta + 2\chi + 1) + t^{-10}(-\chi - 1) + t^{-6}(-2\zeta\chi - 2\zeta)$

$\qquad\qquad + t^{-2}(\zeta\chi + \zeta - 2\chi - 2) + t^2(-2\chi - 2))$

$\qquad + x^8(t^{-16}(\zeta\chi + \zeta + \chi + 2) + t^{-12}(-\chi - 1) + t^{-8}(-\zeta\chi - \zeta)$

$\qquad\qquad + t^{-4}(\zeta\chi + \zeta - 2\chi - 2) + t^4(\chi + 1) + 5\chi + 2)$

$\qquad + O(x^9)$

$\qquad \text{HS}_{\text{NS5}}(\text{CSM}) = 1 + a\chi + a^2 + a^3(\zeta\chi + \zeta + \chi) + a^4(\zeta\chi + \zeta + 1) + a^5(\zeta\chi + \zeta + \chi)$

$\qquad + a^6(\zeta\chi + \zeta + \chi + 2) + a^7(\zeta\chi + \zeta + 2\chi + 1) + a^8(\zeta\chi + \zeta + \chi + 2) + O(a^9)$

$\qquad \text{HS}_{(1,2\kappa)}(\text{CSM}) = 1 + O(b^9)$

| | | |
|---|---|---|
| $\{1,1\}$ | 5 | $\mathcal{I}(\text{CSM}) = 1$ |

$\qquad + x^1(t^{-2}\chi)$

$\qquad + x^2(t^{-4} - \chi - 1)$

$\qquad + x^3(t^{-6}\chi + t^2)$

$\qquad + x^4(t^{-8}(\zeta\chi + \zeta + 1) - 2)$

$\qquad + x^5(t^{-10}(\zeta\chi + \zeta + \chi) + t^{-6}(-\zeta\chi - \zeta) + t^{-2}(-2\chi - 1))$

$\qquad + x^6(t^{-12}(\zeta\chi + \zeta + 1) + t^{-8}(-\zeta\chi - \zeta) + t^4 + 2\chi + 2)$

$\qquad + x^7(t^{-14}(\zeta\chi + \zeta + \chi) + t^{-6}(-\zeta\chi - \zeta) + t^{-2}(-2\chi - 2) + t^2(-2\chi - 2))$

$\qquad + x^8(t^{-16}(\zeta\chi + \zeta + \chi + 2) + t^{-8}(-2\zeta\chi - 2\zeta) + t^{-4}(\zeta\chi + \zeta - 2\chi - 2)$

$\qquad\qquad + t^4(\chi + 1) + 5\chi + 2)$

$\qquad + O(x^9)$

$\qquad \text{HS}_{\text{NS5}}(\text{CSM}) = 1 + a\chi + a^2 + a^3\chi + a^4(1 + \zeta + \zeta\chi) + a^5(\zeta + \chi + \zeta\chi)$

$\qquad + a^6(1 + \zeta + \zeta\chi) + a^7(\zeta + \chi + \zeta\chi) + a^8(2 + \zeta + \chi + \zeta\chi) + O(a^9)$

$\qquad \text{HS}_{(1,2\kappa)}(\text{CSM}) = 1 + O(b^9)$

| | | |
|---|---|---|
| $\{2,2\}$ | 4 | $\mathcal{I}(\text{CSM}) = 1$ |

$$+ x^1(t^{-2}\zeta\chi)$$
$$+ x^2(-\zeta\chi + (\zeta\chi + \zeta + \chi + 2)t^{-4} - 1)$$
$$+ x^3((3\zeta\chi + \zeta + \chi + 1)t^{-6} + (-\zeta\chi - \zeta - \chi - 1)t^{-2} + t^2)$$
$$+ x^4(\zeta\chi + (3\zeta\chi + 3\zeta + 3\chi + 6)t^{-8} + (-3\zeta\chi - 2\zeta - 2\chi - 3)t^{-4} - 2)$$
$$+ O(x^5)$$

$$\mathrm{HS}_{\mathrm{NS5}}(\mathrm{CSM}) = 1 + a\zeta\chi + a^2(\zeta\chi + \zeta + \chi + 2) + a^3(3\zeta\chi + \zeta + \chi + 1)$$
$$+ a^4(3\zeta\chi + 3\zeta + 3\chi + 6) + O(a^5)$$

$$\mathrm{HS}_{(1,2\kappa)}(\mathrm{CSM}) = 1 + O(b^5)$$

| $\{2,2\}$ | 5 | $\mathcal{I}(\mathrm{CSM}) = 1$ |

$$+ x^2((\zeta\chi + \chi + 1)t^{-4} - 1)$$
$$+ x^3((\zeta\chi + \zeta)t^{-6} + (-\zeta\chi - \chi)t^{-2} + t^2)$$
$$+ x^4((2\zeta\chi + \zeta + \chi + 3)t^{-8} + (\zeta(-\chi) - \zeta - 1)t^{-4} - 2)$$
$$+ x^5((2\zeta\chi + 2\zeta + \chi + 1)t^{-10} + (-2\zeta\chi - 2\zeta - \chi - 1)t^{-6} + (\zeta\chi + \chi + 2)t^{-2} - t^2)$$
$$+ x^6(-\zeta\chi + (4\zeta\chi + 2\zeta + 4\chi + 5)t^{-12} + (-2\zeta\chi - 2\zeta - 2\chi - 3)t^{-8}$$
$$+ (\zeta(-\chi) + 2\zeta - 2\chi - 3)t^{-4} + 2t^4 - 2\chi + 1)$$
$$+ O(x^7)$$

$$\mathrm{HS}_{\mathrm{NS5}}(\mathrm{CSM}) = 1 + a^2(\zeta\chi + \chi + 1) + a^3(\zeta\chi + \zeta) + a^4(2\zeta\chi + \zeta + \chi + 3)$$
$$+ a^5(2\zeta\chi + 2\zeta + \chi + 1) + a^6(4\zeta\chi + 2\zeta + 4\chi + 5) + O(a^7)$$

$$\mathrm{HS}_{(1,2\kappa)}(\mathrm{CSM}) = 1 + O(b^7)$$

| $\{2,2\}$ | 6 | $\mathcal{I}(\mathrm{CSM}) = 1$ |

$$+ x^2((\chi + 1)t^{-4} - 1)$$
$$+ x^3(\zeta\chi t^{-6} - \chi t^{-2} + t^2)$$
$$+ x^4((\zeta\chi + \zeta + \chi + 2)t^{-8} + (-\zeta\chi - 1)t^{-4} - 2)$$
$$+ x^5((2\zeta\chi + \zeta)t^{-10} + (-\zeta\chi - \zeta - \chi)t^{-6} + (\chi + 2)t^{-2} - t^2)$$
$$+ x^6((2\zeta\chi + 2\zeta + 2\chi + 3)t^{-12} + (-2\zeta\chi - 2\zeta - 1)t^{-8}$$
$$+ (\zeta\chi - 2\chi - 3)t^{-4} + 2t^4 - 2\chi + 1)$$
$$+ O(x^7)$$

$$\mathrm{HS}_{\mathrm{NS5}}(\mathrm{CSM}) = 1 + a^2(\chi + 1) + a^3\zeta\chi + a^4(\zeta\chi + \zeta + \chi + 2)$$
$$+ a^5(2\zeta\chi + \zeta) + a^6(2\zeta\chi + 2\zeta + 2\chi + 3) + O(a^7)$$

$$\mathrm{HS}_{(1,2\kappa)}(\mathrm{CSM}) = 1 + O(b^7)$$

| $\{2,2\}$ | 7 | $\mathcal{I}(\mathrm{CSM}) = 1$ |

$$+ x^2((\chi + 1)t^{-4} - 1)$$
$$+ x^3(t^2 - \chi t^{-2})$$
$$+ x^4((\zeta\chi + \chi + 2)t^{-8} - t^{-4} - 2)$$
$$+ x^5(\zeta\chi t^{-10} + \zeta t^{-10} - \zeta\chi t^{-6} - \chi t^{-6} + (\chi + 2)t^{-2} - t^2)$$
$$+ x^6((2\zeta\chi + \zeta + 2\chi + 2)t^{-12} + (-\zeta\chi - \zeta - 1)t^{-8} + (-2\chi - 3)t^{-4} + 2t^4 - 2\chi + 1)$$
$$+ O(x^7)$$

$$\mathrm{HS}_{\mathrm{NS5}}(\mathrm{CSM}) = 1 + a^2(\chi + 1) + a^4(\zeta\chi + \chi + 2) + a^5(\zeta\chi + \zeta)$$
$$+ a^6(2\zeta\chi + \zeta + 2\chi + 2) + O(a^7)$$

$$\mathrm{HS}_{(1,2\kappa)}(\mathrm{CSM}) = 1 + O(b^7)$$

| $\{3,3\}$ | 7 | $\mathcal{I}(\mathrm{CSM}) = 1$ |

$$+ x^2((\zeta\chi + 1)t^{-4} - 1)$$
$$+ O(x^3)$$

$$\mathrm{HS}_{\mathrm{NS5}}(\mathrm{CSM}) = 1 + a^2(\zeta\chi + 1) + O(a^3)$$

$$\mathrm{HS}_{(1,2\kappa)}(\mathrm{CSM}) = 1 + O(b^3)$$

| $\{3,3\}$ | 8 | $\mathcal{I}(\mathrm{CSM}) = 1$ |

$$+ x^2(t^{-4} - 1)$$
$$+ O(x^3)$$

$$\text{HS}_{\text{NS5}}(\text{CSM}) = 1 + a^2 + O(a^3)$$

$$\text{HS}_{(1,2\kappa)}(\text{CSM}) = 1 + O(b^3)$$

| | | |
|---|---|---|
| $\{1,2\}$ | 3 | $\mathcal{I}(\text{CSM}) = 1$ $+ x^1((\zeta\chi + \zeta + \chi)t^{-2})$ $+ x^2(-\zeta\chi - \zeta + (\zeta\chi + \zeta + \chi + 2)t^{-4} - \chi - 1)$ $+ x^3((2\zeta\chi + 2\zeta + 2\chi + 1)t^{-6} + (-\zeta\chi - \zeta - \chi - 1)t^{-2} + t^2)$ $+ x^4((2\zeta\chi + 2\zeta + 2\chi + 3)t^{-8} + (-\zeta\chi - \zeta - \chi - 1)t^{-4} - 2)$ $+ O(x^5)$ |

$$\text{HS}_{\text{NS5}}(\text{CSM}) = 1 + a(\zeta\chi + \zeta + \chi) + a^2(\zeta\chi + \zeta + \chi + 2) + a^3(2\zeta\chi + 2\zeta + 2\chi + 1)$$
$$+ a^4(2\zeta\chi + 2\zeta + 2\chi + 3) + O(a^5)$$

$$\text{HS}_{(1,2\kappa)}(\text{CSM}) = 1 + O(b^5)$$

| | | |
|---|---|---|
| $\{1,2\}$ | 4 | $\mathcal{I}(\text{CSM}) = 1$ $+ x^1(\chi t^{-2})$ $+ x^2((\zeta\chi + \zeta + 1)t^{-4} - \chi - 1)$ $+ x^3((\zeta\chi + \zeta + \chi)t^{-6} + (\zeta(-\chi) - \zeta)t^{-2} + t^2)$ $+ x^4((\zeta\chi + \zeta + \chi + 2)t^{-8} + (-\zeta\chi - \zeta)t^{-4} + \chi - 2)$ $+ O(x^5)$ |

$$\text{HS}_{\text{NS5}}(\text{CSM}) = 1 + a\chi + a^2(\zeta\chi + \zeta + 1) + a^3(\zeta\chi + \zeta + \chi)$$
$$+ a^4(\zeta\chi + \zeta + \chi + 2) + O(a^5)$$

$$\text{HS}_{(1,2\kappa)}(\text{CSM}) = 1 + O(b^5)$$

| | | |
|---|---|---|
| $\{1,2\}$ | 5 | $\mathcal{I}(\text{CSM}) = 1$ $+ x^1(\chi t^{-2})$ $+ x^2(t^{-4} - \chi - 1)$ $+ x^3((\zeta\chi + \zeta + \chi)t^{-6} + t^2)$ $+ x^4((\zeta\chi + \zeta + 1)t^{-8} + (-\zeta\chi - \zeta)t^{-4} + \chi - 2)$ $+ O(x^5)$ |

$$\text{HS}_{\text{NS5}}(\text{CSM}) = 1 + a\chi + a^2 + a^3(\zeta\chi + \zeta + \chi) + a^4(\zeta\chi + \zeta + 1) + O(a^5)$$

$$\text{HS}_{(1,2\kappa)}(\text{CSM}) = 1 + O(b^5)$$

| | | |
|---|---|---|
| $\{1,3\}$ | 6 | $\mathcal{I}(\text{CSM}) = 1$ $+ x^1(\chi t^{-2})$ $+ x^2(t^{-4} - \chi - 1)$ $+ x^3((\zeta\chi + \zeta + \chi)t^{-6} + t^2)$ $+ O(x^4)$ |

$$\text{HS}_{\text{NS5}}(\text{CSM}) = 1 + a\chi + a^2 + a^3(\zeta\chi + \zeta + \chi) + O(a^4)$$

$$\text{HS}_{(1,2\kappa)}(\text{CSM}) = 1 + O(b^4)$$

| | | |
|---|---|---|
| $\{1,3\}$ | 7 | $\mathcal{I}(\text{CSM}) = 1$ $+ x^1(\chi t^{-2})$ $+ x^2(t^{-4} - \chi - 1)$ $+ x^3(\chi t^{-6} + t^2)$ $+ O(x^4)$ |

$$\text{HS}_{\text{NS5}}(\text{CSM}) = 1 + a\chi + a^2 + a^3\chi + O(a^4)$$

$$\text{HS}_{(1,2\kappa)}(\text{CSM}) = 1 + O(b^4)$$

| | | |
|---|---|---|
| $\{2,1\}$ | 3 | $\mathcal{I}(\text{CSM}) = 1$ $+ x^1(\zeta\chi t^{-2})$ $+ x^2(-\zeta\chi + (\zeta\chi + 2)t^{-4} - 1)$ $+ x^3((2\zeta\chi + 1)t^{-6} + (-\zeta\chi - 1)t^{-2} + t^2)$ $+ x^4((2\zeta\chi + 3)t^{-8} + (-\zeta\chi - 1)t^{-4} - 2)$ |

$$+ x^5((3\zeta\chi + 2)t^{-10} + (-\zeta\chi - 1)t^{-6} + (-2\zeta\chi - 1)t^{-2})$$
$$+ x^6(2\zeta\chi + (3\zeta\chi + 4)t^{-12} + (-\zeta\chi - 1)t^{-8} + (-3\zeta\chi - 3)t^{-4} + t^4 + 2)$$
$$+ O(x^7)$$

$$\text{HS}_{\text{NS5}}(\text{CSM}) = 1 + a\zeta\chi + a^2(\zeta\chi + 2) + a^3(2\zeta\chi + 1) + a^4(2\zeta\chi + 3)$$
$$+ a^5(3\zeta\chi + 2) + a^6(3\zeta\chi + 4) + O(a^7)$$

$$\text{HS}_{(1,2\kappa)}(\text{CSM}) = 1 + O(b^7)$$

| $\{N,M\}$ | $\kappa$ | |
|---|---|---|
| $\{2,1\}$ | 4 | $\mathcal{I}(\text{CSM}) = 1$ $+ x^2((\zeta\chi + 1)t^{-4} - 1)$ $+ x^3(\zeta\chi t^{-6} - \zeta\chi t^{-2} + t^2)$ $+ x^4((\zeta\chi + 2)t^{-8} - \zeta\chi t^{-4} - 2)$ $+ x^5((\zeta\chi + 1)t^{-10} - t^{-6})$ $+ x^6((2\zeta\chi + 2)t^{-12} - t^{-8} + (-2\zeta\chi - 1)t^{-4} + t^4 + 1)$ $+ O(x^7)$ $\text{HS}_{\text{NS5}}(\text{CSM}) = 1 + a^2(\zeta\chi + 1) + a^3\zeta\chi + a^4(\zeta\chi + 2) + a^5(\zeta\chi + 1)$ $+ a^6(2\zeta\chi + 2) + O(a^7)$ $\text{HS}_{(1,2\kappa)}(\text{CSM}) = 1 + O(b^7)$ |
| $\{2,1\}$ | 5 | $\mathcal{I}(\text{CSM}) = 1$ $+ x^2(t^{-4} - 1)$ $+ x^3(\zeta\chi t^{-6} + t^2)$ $+ x^4((\zeta\chi + 1)t^{-8} - \zeta\chi t^{-4} - 2)$ $+ x^5(\zeta\chi t^{-10} - \zeta\chi t^{-6})$ $+ x^6((\zeta\chi + 2)t^{-12} + t^4 - t^{-4} + 1)$ $+ O(x^7)$ $\text{HS}_{\text{NS5}}(\text{CSM}) = 1 + a^2 + a^3\zeta\chi + a^4(\zeta\chi + 1) + a^5\zeta\chi + a^6(\zeta\chi + 2) + O(a^7)$ $\text{HS}_{(1,2\kappa)}(\text{CSM}) = 1 + O(b^7)$ |
| $\{3,1\}$ | 6 | $\mathcal{I}(\text{CSM}) = 1$ $+ x^2(t^{-4} - 1)$ $+ x^3(\zeta\chi t^{-6} + t^2)$ $+ O(x^4)$ $\text{HS}_{\text{NS5}}(\text{CSM}) = 1 + a^2 + a^3\zeta\chi + O(a^4)$ $\text{HS}_{(1,2\kappa)}(\text{CSM}) = 1 + O(b^4)$ |
| $\{3,1\}$ | 7 | $\mathcal{I}(\text{CSM}) = 1$ $+ x^2(t^{-4} - 1)$ $+ x^3 t^2$ $+ O(x^4)$ $\text{HS}_{\text{NS5}}(\text{CSM}) = 1 + a^2 + O(a^4)$ $\text{HS}_{(1,2\kappa)}(\text{CSM}) = 1 + O(b^4)$ |

**Table 6:** Expanded index and Hilbert series for the linear $\text{SO}(2N)_{+2\kappa} \times \text{Sp}(M)_{-\kappa}$ CSM theory in Figure 9 for some values of $N$, $M$ and $|\kappa| \geqslant N + M$ (for which the theory is good).

| $\{N,M\}$ | $\kappa$ | Index and Hilbert series for CSM $= \text{SO}(2N+1)_{+2\kappa} \times \text{Sp}(M)_{-\kappa}$ |
|---|---|---|
| $\{1,1\}$ | 2 | $\mathcal{I}(\text{CSM}) = 1$ $+ x^1(t^{-2}\zeta\chi)$ $+ x^2(-\zeta\chi + t^{-4}(\zeta\chi + 2) - 1)$ |

$$+ x^3(-\zeta - \chi + t^2 + t^{-6}(2\zeta\chi + 1) + t^{-2}(-\zeta\chi - 1))$$
$$+ x^4(\zeta\chi - 2 + t^2(\zeta + \chi) + t^{-8}(2\zeta\chi + 3) + t^{-4}(-\zeta\chi - 1) + t^{-2}(\zeta + \chi))$$
$$+ x^5(-3\zeta - 3\chi + t^2(-\zeta\chi - 1) + t^{-10}(3\zeta\chi + 2)$$
$$+ t^{-6}(-\zeta\chi - 1) + t^{-2}(-3\zeta\chi - 1))$$
$$+ x^6(3\zeta\chi + 3 + 2t^4 + t^2(2\zeta + 2\chi) + t^{-12}(3\zeta\chi + 4)$$
$$+ t^{-8}(-\zeta\chi - 1) + t^{-4}(-3\zeta\chi - 3))$$
$$+ x^7(t^2(-\zeta\chi - 3) + t^{-14}(4\zeta\chi + 3) + t^{-10}(-\zeta\chi - 1) + t^{-6}(-2\zeta\chi - 2)$$
$$+ t^{-4}(\zeta + \chi) + t^{-2}(\zeta\chi + 2))$$
$$+ x^8(-2 + t^2(-\zeta - \chi) + t^{-16}(4\zeta\chi + 5) + t^{-12}(-\zeta\chi - 1) + t^{-8}(-2\zeta\chi - 2)$$
$$+ t^{-6}(\zeta + \chi) + t^{-4}(-4\zeta\chi - 5) + t^{-2}(-7\zeta - 7\chi))$$
$$+ O(x^9)$$

$$\text{HS}_{\text{NS5}}(\text{CSM}) = 1 + a\zeta\chi + a^2(\zeta\chi + 2) + a^3(2\zeta\chi + 1) + a^4(2\zeta\chi + 3) + a^5(3\zeta\chi + 2)$$
$$+ a^6(3\zeta\chi + 4) + a^7(4\zeta\chi + 3) + a^8(4\zeta\chi + 5) + O(a^9)$$

$$\text{HS}_{(1,2\kappa)}(\text{CSM}) = 1 + O(b^9)$$

---

| {1,1} | 3 | $\mathcal{I}(\text{CSM}) = 1$ |
|---|---|---|

$$+ x^2(t^{-4}(\zeta\chi + 1) - 1)$$
$$+ x^3(-\chi + t^2 + t^{-6}\zeta\chi + t^{-2}(-\zeta\chi))$$
$$+ x^4(-2 + t^2\chi + t^{-8}(\zeta\chi + 2) + t^{-4}(-\zeta\chi) + t^{-2}(\chi - \zeta))$$
$$+ x^5(\zeta - 2\chi - t^2 + t^4\chi + t^{-10}(\zeta\chi + 1) + t^{-6}(-1) + t^{-4}\zeta + t^{-2}\zeta\chi)$$
$$+ x^6(-\zeta\chi + 2 - t^2\chi + 2t^4 + t^6(-\chi) + t^{-12}(2\zeta\chi + 2) + t^{-8}(-1)$$
$$+ t^{-4}(-3\zeta\chi - 1) + t^{-2}(-3\zeta))$$
$$+ x^7(2\zeta + 2\chi - 3t^2 + 3t^4\chi + t^{-14}(2\zeta\chi + 1) + t^{-10}(-\zeta\chi) + t^{-6}(-2\zeta\chi - 1)$$
$$+ t^{-2}(3\zeta\chi))$$
$$+ x^8(-\zeta\chi - 1 - t^4 - 7t^2\chi + t^6(-\chi) + t^{-16}(2\zeta\chi + 3) + t^{-12}(-\zeta\chi) + t^{-8}(-2)$$
$$+ t^{-6}(2\zeta) + t^{-4}(\zeta\chi - 1) + t^{-2}(-4\chi))$$
$$+ O(x^9)$$

$$\text{HS}_{\text{NS5}}(\text{CSM}) = 1 + a^2(\zeta\chi + 1) + a^3\zeta\chi + a^4(\zeta\chi + 2) + a^5(\zeta\chi + 1) + a^6(2\zeta\chi + 2)$$
$$+ a^7(2\zeta\chi + 1) + a^8(2\zeta\chi + 3) + O(a^9)$$

$$\text{HS}_{(1,2\kappa)}(\text{CSM}) = 1 + O(b^9)$$

---

| {1,1} | 4 | $\mathcal{I}(\text{CSM}) = 1$ |
|---|---|---|

$$+ x^2(-1 + t^{-4})$$
$$+ x^3(-\chi + t^2 + t^{-6}\zeta\chi)$$
$$+ x^4(-2 + t^2\chi + t^{-8}(\zeta\chi + 1) + t^{-4}(-\zeta\chi) + t^{-2}\chi)$$
$$+ x^5(-2\chi - t^2 + t^4\chi + t^{-10}\zeta\chi + t^{-6}(-\zeta\chi) + t^{-4}(-\zeta))$$
$$+ x^6(2 - t^2\chi + 2t^4 + t^6(-\chi) + t^{-12}(\zeta\chi + 2) + t^{-6}\zeta + t^{-4}(\zeta\chi - 1) + t^{-2}\zeta)$$
$$+ x^7(2\chi - 3t^2 + 3t^4\chi + t^{-14}(\zeta\chi + 1) + t^{-10}(-1) + t^{-6}(-3\zeta\chi)$$
$$+ t^{-4}(-3\zeta) + t^{-2}(-\zeta\chi))$$
$$+ x^8(-1 - t^4 - 6t^2\chi + t^{-16}(\zeta\chi + 2) + t^{-12}(-1) + t^{-8}(-2\zeta\chi) + t^{-6}\chi$$
$$+ t^{-4}(3\zeta\chi - 2) + t^{-2}(2\zeta - 3\chi))$$
$$+ O(x^9)$$

$$\text{HS}_{\text{NS5}}(\text{CSM}) = 1 + a^2 + a^3\zeta\chi + a^4(\zeta\chi + 1) + a^5\zeta\chi + a^6(\zeta\chi + 2) + a^7(\zeta\chi + 1)$$
$$+ a^8(\zeta\chi + 2) + O(a^9)$$

$$\text{HS}_{(1,2\kappa)}(\text{CSM}) = 1 + O(b^9)$$

---

| {1,1} | 5 | $\mathcal{I}(\text{CSM}) = 1$ |
|---|---|---|

$$+ x^2(-1 + t^{-4})$$
$$+ x^3(-\chi + t^2)$$
$$+ x^4(-2 + t^2\chi + t^{-8}(\zeta\chi + 1) + t^{-2}\chi)$$
$$+ x^5(-2\chi - t^2 + t^4\chi + t^{-10}\zeta\chi + t^{-6}(-\zeta\chi))$$
$$+ x^6(2 - t^2\chi + 2t^4 + t^6(-\chi) + t^{-12}(\zeta\chi + 1) + t^{-8}(-\zeta\chi) + t^{-6}(-\zeta) + t^{-4}(-1))$$
$$+ x^7(2\chi - 3t^2 + 3t^4\chi + t^{-14}\zeta\chi + t^{-8}\zeta + t^{-6}\zeta\chi + t^{-4}\zeta)$$

$$+ x^8(-1 - t^4 - 6t^2\chi + t^{-16}(\zeta\chi + 2) + t^{-8}(-3\zeta\chi) + t^{-6}(\chi - 3\zeta) + t^{-4}(-\zeta\chi - 2)$$
$$+ t^{-2}(-3\chi))$$
$$+ O(x^9)$$

$$\mathrm{HS}_{\mathrm{NS5}}(\mathrm{CSM}) = 1 + a^2 + a^4(\zeta\chi + 1) + a^5\zeta\chi + a^6(\zeta\chi + 1) + a^7\zeta\chi$$
$$+ a^8(\zeta\chi + 2) + O(a^9)$$

$$\mathrm{HS}_{(1,2\kappa)}(\mathrm{CSM}) = 1 + O(b^9)$$

| $\{2,2\}$ | 4 | $\mathcal{I}(\mathrm{CSM}) = 1$ |
|---|---|---|

$$+ x^1(t^{-2}\zeta\chi)$$
$$+ x^2\left(-\zeta\chi + (\zeta\chi + 2)t^{-4} - 1\right)$$
$$+ x^3\left((3\zeta\chi + 1)t^{-6} + (-\zeta\chi - 1)t^{-2} + t^2\right)$$
$$+ O(x^4)$$

$$\mathrm{HS}_{\mathrm{NS5}}(\mathrm{CSM}) = 1 + a\zeta\chi + a^2(\zeta\chi + 2) + a^3(3\zeta\chi + 1) + O(a^4)$$

$$\mathrm{HS}_{(1,2\kappa)}(\mathrm{CSM}) = 1 + O(b^4)$$

| $\{2,2\}$ | 5 | $\mathcal{I}(\mathrm{CSM}) = 1$ |
|---|---|---|

$$+ x^2(t^{-4}(\zeta\chi + 1) - 1)$$
$$+ x^3(t^{-6}\zeta\chi - t^{-2}\zeta\chi + t^2)$$
$$+ x^4(t^{-8}(2\zeta\chi + 3) + t^{-4}(-\zeta\chi - 1) - t^{-2}\chi - 2)$$
$$+ O(x^5)$$

$$\mathrm{HS}_{\mathrm{NS5}}(\mathrm{CSM}) = 1 + a^2(\zeta\chi + 1) + a^3\zeta\chi + a^4(2\zeta\chi + 3) + O(a^5)$$

$$\mathrm{HS}_{(1,2\kappa)}(\mathrm{CSM}) = 1 + O(b^5)$$

| $\{2,2\}$ | 6 | $\mathcal{I}(\mathrm{CSM}) = 1$ |
|---|---|---|

$$+ x^2(t^{-4} - 1)$$
$$+ x^3(t^{-6}\zeta\chi + t^2)$$
$$+ x^4(t^{-8}(\zeta\chi + 2) + t^{-4}(-\zeta\chi - 1) - t^{-2}\chi - 2)$$
$$+ O(x^5)$$

$$\mathrm{HS}_{\mathrm{NS5}}(\mathrm{CSM}) = 1 + a^2 + a^3\zeta\chi + a^4(\zeta\chi + 2) + O(a^5)$$

$$\mathrm{HS}_{(1,2\kappa)}(\mathrm{CSM}) = 1 + O(b^5)$$

| $\{2,2\}$ | 7 | $\mathcal{I}(\mathrm{CSM}) = 1$ |
|---|---|---|

$$+ x^2(t^{-4} - 1)$$
$$+ x^3 t^2$$
$$+ x^4(t^{-8}(\zeta\chi + 2) - t^{-4} - t^{-2}\chi - 2)$$
$$+ O(x^5)$$

$$\mathrm{HS}_{\mathrm{NS5}}(\mathrm{CSM}) = 1 + a^2 + a^4(\zeta\chi + 2) + O(a^5)$$

$$\mathrm{HS}_{(1,2\kappa)}(\mathrm{CSM}) = 1 + O(b^5)$$

| $\{1,2\}$ | 3 | $\mathcal{I}(\mathrm{CSM}) = 1$ |
|---|---|---|

$$+ x^1(\zeta\chi t^{-2})$$
$$+ x^2(-\zeta\chi + (\zeta\chi + 2)t^{-4} + (\zeta + \chi)t^{-2} - 1)$$
$$+ x^3(-2\zeta + (2\zeta\chi + 1)t^{-6} + (-\zeta\chi - 1)t^{-2} + t^2 - 2\chi)$$
$$+ O(x^4)$$

$$\mathrm{HS}_{\mathrm{NS5}}(\mathrm{CSM}) = 1 + a^1\zeta\chi + a^2(\zeta\chi + 2) + a^3(2\zeta\chi + 1) + O(a^4)$$

$$\mathrm{HS}_{(1,2\kappa)}(\mathrm{CSM}) = 1 + O(b^4)$$

| $\{1,2\}$ | 4 | $\mathcal{I}(\mathrm{CSM}) = 1$ |
|---|---|---|

$$+ x^2((\zeta\chi + 1)t^{-4} + \chi t^{-2} - 1)$$
$$+ x^3(\zeta\chi t^{-6} + (\zeta - \chi)t^{-4} - \zeta\chi t^{-2} + t^2 - 2\chi)$$
$$+ O(x^4)$$

$$\mathrm{HS}_{\mathrm{NS5}}(\mathrm{CSM}) = 1 + a^2(\zeta\chi + 1) + a^3\zeta\chi + O(a^4)$$

$$\mathrm{HS}_{(1,2\kappa)}(\mathrm{CSM}) = 1 + O(b^4)$$

| | | |
|---|---|---|
| $\{1,2\}$ | 5 | $\mathcal{I}(\mathrm{CSM}) = 1$ |

$$+ x^2(t^{-4} + \chi t^{-2} - 1)$$
$$+ x^3(\zeta\chi t^{-6} - \chi t^{-4} + t^2 - 2\chi)$$
$$+ O(x^4)$$

$$\mathrm{HS}_{\mathrm{NS5}}(\mathrm{CSM}) = 1 + a^2 + a^3\zeta\chi + O(a^4)$$

$$\mathrm{HS}_{(1,2\kappa)}(\mathrm{CSM}) = 1 + O(b^4)$$

| | | |
|---|---|---|
| $\{1,3\}$ | 4 | $\mathcal{I}(\mathrm{CSM}) = 1$ |

$$+ x^1(\zeta\chi t^{-2})$$
$$+ x^2(-\zeta\chi + (\zeta\chi + 2)t^{-4} + (\zeta + \chi)t^{-2} - 1)$$
$$+ O(x^3)$$

$$\mathrm{HS}_{\mathrm{NS5}}(\mathrm{CSM}) = 1 + a\zeta\chi + a^2(\zeta\chi + 2) + O(a^3)$$

$$\mathrm{HS}_{(1,2\kappa)}(\mathrm{CSM}) = 1 + O(b^3)$$

| | | |
|---|---|---|
| $\{1,3\}$ | 5 | $\mathcal{I}(\mathrm{CSM}) = 1$ |

$$+ x^2((\zeta\chi + 1)t^{-4} + \chi t^{-2} - 1)$$
$$+ O(x^3)$$

$$\mathrm{HS}_{\mathrm{NS5}}(\mathrm{CSM}) = 1 + a^2(\zeta\chi + 1) + O(a^3)$$

$$\mathrm{HS}_{(1,2\kappa)}(\mathrm{CSM}) = 1 + O(b^3)$$

| | | |
|---|---|---|
| $\{1,3\}$ | 6 | $\mathcal{I}(\mathrm{CSM}) = 1$ |

$$+ x^2(t^{-4} + \chi t^{-2} - 1)$$
$$+ O(x^3)$$

$$\mathrm{HS}_{\mathrm{NS5}}(\mathrm{CSM}) = 1 + a^2 + O(a^3)$$

$$\mathrm{HS}_{(1,2\kappa)}(\mathrm{CSM}) = 1 + O(b^3)$$

| | | |
|---|---|---|
| $\{2,1\}$ | 3 | $\mathcal{I}(\mathrm{CSM}) = 1$ |

$$+ x^1(t^{-2}\zeta\chi)$$
$$+ x^2(-\zeta\chi + t^{-4}(\zeta\chi + 2) - 1)$$
$$+ x^3((2\zeta\chi + 1)t^{-6} + (-\zeta\chi - 1)t^{-2} + t^2)$$
$$+ x^4(\zeta\chi + (2\zeta\chi + 3)t^{-8} + (-\zeta\chi - 1)t^{-4} - 2)$$
$$+ x^5((3\zeta\chi + 2)t^{-10} + (-\zeta\chi - 1)t^{-6} + t^2(-\zeta\chi - 1) + (-3\zeta\chi - 1)t^{-2})$$
$$+ O(x^6)$$

$$\mathrm{HS}_{\mathrm{NS5}}(\mathrm{CSM}) = 1 + a\zeta\chi + a^2(\zeta\chi + 2) + a^3(2\zeta\chi + 1) + a^4(2\zeta\chi + 3)$$
$$+ a^5(3\zeta\chi + 2) + O(a^6)$$

$$\mathrm{HS}_{(1,2\kappa)}(\mathrm{CSM}) = 1 + O(b^6)$$

| | | |
|---|---|---|
| $\{2,1\}$ | 4 | $\mathcal{I}(\mathrm{CSM}) = 1$ |

$$+ x^2(t^{-4}(\zeta\chi + 1) - 1)$$
$$+ x^3(t^{-6}\zeta\chi - t^{-2}\zeta\chi + t^2)$$
$$+ x^4(t^{-8}(\zeta\chi + 2) - t^{-4}\zeta\chi - 2)$$
$$+ x^5(t^{-10}(\zeta\chi + 1) - t^{-6} + t^{-2}\zeta\chi - t^2)$$
$$+ O(x^6)$$

$$\mathrm{HS}_{\mathrm{NS5}}(\mathrm{CSM}) = 1 + a^2(\zeta\chi + 1) + a^3\zeta\chi + a^4(\zeta\chi + 2) + a^5(\zeta\chi + 1) + O(a^6)$$

$$\mathrm{HS}_{(1,2\kappa)}(\mathrm{CSM}) = 1 + O(b^6)$$

| | | |
|---|---|---|
| $\{2,1\}$ | 5 | $\mathcal{I}(\mathrm{CSM}) = 1$ |

$$+ x^2(t^{-4} - 1)$$
$$+ x^3(\zeta\chi t^{-6} + t^2)$$
$$+ x^4((\zeta\chi + 1)t^{-8} - \zeta\chi t^{-4} - 2)$$
$$+ x^5(\zeta\chi t^{-10} - \zeta\chi t^{-6} - t^2)$$

$$+ O(x^6)$$

$$\text{HS}_{\text{NS5}}(\text{CSM}) = 1 + a^2 + a^3\zeta\chi + a^4(\zeta\chi + 1) + a^5\zeta\chi + O(a^6)$$

$$\text{HS}_{(1,2\kappa)}(\text{CSM}) = 1 + O(b^6)$$

| $\{3,1\}$ | 4 | $\mathcal{I}(\text{CSM}) = 1$ |
|---|---|---|
| | | $+ x^1(\zeta\chi t^{-2})$ |
| | | $+ x^2(-\zeta\chi + (\zeta\chi + 2)t^{-4} - 1)$ |
| | | $+ x^3((2\zeta\chi + 1)t^{-6} + (-\zeta\chi - 1)t^{-2} + t^2)$ |
| | | $+ x^4(\zeta\chi + (2\zeta\chi + 3)t^{-8} + (-\zeta\chi - 1)t^{-4} - 2)$ |
| | | $+ O(x^5)$ |

$$\text{HS}_{\text{NS5}}(\text{CSM}) = 1 + a(\zeta\chi) + a^2(\zeta\chi + 2) + a^3(2\zeta\chi + 1) + a^4(2\zeta\chi + 3) + O(a^5)$$

$$\text{HS}_{(1,2\kappa)}(\text{CSM}) = 1 + O(b^5)$$

| $\{3,1\}$ | 5 | $\mathcal{I}(\text{CSM}) = 1$ |
|---|---|---|
| | | $+ x^2((\zeta\chi + 1)t^{-4} - 1)$ |
| | | $+ x^3(\zeta\chi t^{-6} - \zeta\chi t^{-2} + t^2)$ |
| | | $+ x^4((\zeta\chi + 2)t^{-8} - \zeta\chi t^{-4} - 2)$ |
| | | $+ O(x^5)$ |

$$\text{HS}_{\text{NS5}}(\text{CSM}) = 1 + a^2(\zeta\chi + 1) + a^3\zeta\chi + a^4(\zeta\chi + 2) + O(a^5)$$

$$\text{HS}_{(1,2\kappa)}(\text{CSM}) = 1 + O(b^5)$$

| $\{3,1\}$ | 6 | $\mathcal{I}(\text{CSM}) = 1$ |
|---|---|---|
| | | $+ x^2(t^{-4} - 1)$ |
| | | $+ x^3(\zeta\chi t^{-6} + t^2)$ |
| | | $+ x^4((\zeta\chi + 1)t^{-8} - \zeta\chi t^{-4} - 2)$ |
| | | $+ O(x^5)$ |

$$\text{HS}_{\text{NS5}}(\text{CSM}) = 1 + a^2 + a^3\zeta\chi + a^4(\zeta\chi + 1) + O(a^5)$$

$$\text{HS}_{(1,2\kappa)}(\text{CSM}) = 1 + O(b^5)$$

**Table 7:** Expanded index and Hilbert series for the linear $\text{SO}(2N+1)_{+2\kappa} \times \text{Sp}(M)_{-\kappa}$ CSM theory in Figure 11 for some values of $N$, $M$ and $|\kappa| \geqslant N + M$ (for which the theory is good).

## B.3   Linear examples with 3 nodes

In the following tables the index expansions for some instances of the 3 nodes examples of Section 4.2.2 have been reported. In particular, in Table 8 one can find the expanded index and the Hilbert series for the linear $\text{SO}(2N)_{+2\kappa} \times \text{Sp}(M)_{-\kappa} \times \text{SO}(2N)_{+2\kappa}$ CSM theory for some values of $N$, $M$ and $\kappa$, while in Table 9 the same is shown for the $\text{SO}(2N+1)_{+2\kappa} \times \text{Sp}(M)_{-\kappa} \times \text{SO}(2N+1)_{+2\kappa}$ CSM theory. Moreover, in Table 10 the aforementioned information is reported for the $\text{Sp}(N)_{+\kappa} \times \text{SO}(2M+1) \times \text{Sp}(N)_{-\kappa}$ CSM theory.

| $\{N, M\}$ | $\kappa$ | Index and Hilbert series for CSM $= \text{SO}(2N)_{+2\kappa} \times \text{Sp}(M)_{-\kappa} \times \text{SO}(2N)_{+2\kappa}$ |
|---|---|---|
| $\{1,1\}$ | 1 | $\mathcal{I}(\text{CSM}) = 1$ |
| | | $+ x(t^{-2}(\zeta_1\chi_1\chi_2 + \zeta_1\chi_2 + \chi_1) + t^2(\zeta_2\chi_1\chi_2 + \zeta_2\chi_1 + \chi_2))$ |
| | | $+ x^2(t^{-4}(\zeta_1\chi_1\chi_2 + \zeta_1\chi_2 + \chi_1 + 2) + t^4(\zeta_2\chi_1\chi_2 + \zeta_2\chi_1 + \chi_2 + 2) - \zeta_1\chi_1\chi_2 - \zeta_1\chi_2$ |
| | | $\quad -\zeta_2\chi_1\chi_2 - \zeta_2\chi_1 - \chi_1 - \chi_2 - 1)$ |
| | | $+ x^3(t^{-6}(2\zeta_1\chi_1\chi_2 + 2\zeta_1\chi_2 + 2\chi_1 + 1) + t^{-2}(-\zeta_1\chi_1\chi_2 - \zeta_1\chi_2 - \chi_1 - 1)$ |
| | | $\quad + t^2(-\zeta_2\chi_1\chi_2 - \zeta_2\chi_1 - \chi_2 - 1) + t^6(2\zeta_2\chi_1\chi_2 + 2\zeta_2\chi_1 + 2\chi_2 + 1))$ |
| | | $+ x^4(t^{-8}(2\zeta_1\chi_1\chi_2 + 2\zeta_1\chi_2 + 2\chi_1 + 3) + t^{-4}(-\zeta_1\chi_1\chi_2 - \zeta_1\chi_2 - \chi_1 - 1)$ |

$$+ t^4(-\zeta_2\chi_1\chi_2 - \zeta_2\chi_1 - \chi_2 - 1) + t^8(2\zeta_2\chi_1\chi_2 + 2\zeta_2\chi_1 + 2\chi_2 + 3) + \zeta_1\zeta_2\chi_1\chi_2$$
$$+ \zeta_1\zeta_2\chi_1 + \zeta_1\zeta_2\chi_2 + \zeta_1\zeta_2 + \zeta_1\chi_1 + \zeta_1 + \zeta_2\chi_2 + \zeta_2 + \chi_1\chi_2 - 2)$$
$$+ x^5(t^{-10}(3\zeta_1\chi_1\chi_2 + 3\zeta_1\chi_2 + 3\chi_1 + 2) + t^{-6}(-\zeta_1\chi_1\chi_2 - \zeta_1\chi_2 - \chi_1 - 1)$$
$$+ t^{-2}(-\zeta_1\zeta_2\chi_1\chi_2 - \zeta_1\zeta_2\chi_1 - \zeta_1\zeta_2\chi_2 - \zeta_1\zeta_2 - 3\zeta_1\chi_1\chi_2 - \zeta_1\chi_1 - 3\zeta_1\chi_2 - \zeta_1$$
$$- \zeta_2\chi_2 - \zeta_2 - \chi_1\chi_2 - 3\chi_1 - 2) + t^2(-\zeta_1\zeta_2\chi_1\chi_2 - \zeta_1\zeta_2\chi_1 - \zeta_1\zeta_2\chi_2 - \zeta_1\zeta_2 - \zeta_1\chi_1$$
$$- \zeta_1 - 3\zeta_2\chi_1\chi_2 - 3\zeta_2\chi_1 - \zeta_2\chi_2 - \zeta_2 - \chi_1\chi_2 - 3\chi_2 - 2)$$
$$+ t^6(-\zeta_2\chi_1\chi_2 - \zeta_2\chi_1 - \chi_2 - 1) + t^{10}(3\zeta_2\chi_1\chi_2 + 3\zeta_2\chi_1 + 3\chi_2 + 2))$$
$$+ O(x^6)$$

$$\mathrm{HS}_{\mathrm{NS5}}(\mathrm{CSM}) = 1 + a(\zeta_1\chi_1\chi_2 + \zeta_1\chi_2 + \chi_1) + a^2(\zeta_1\chi_1\chi_2 + \zeta_1\chi_2 + \chi_1 + 2)$$
$$+ a^3(2\zeta_1\chi_1\chi_2 + 2\zeta_1\chi_2 + 2\chi_1 + 1) + a^4(2\zeta_1\chi_1\chi_2 + 2\zeta_1\chi_2 + 2\chi_1 + 3)$$
$$+ a^5(3\zeta_1\chi_1\chi_2 + 3\zeta_1\chi_2 + 3\chi_1 + 2) + O(a^6)$$

$$\mathrm{HS}_{(1,2\kappa)}(\mathrm{CSM}) = \mathrm{HS}_{\mathrm{NS5}}(\mathrm{CSM})$$

---

| $\{1,1\}$ | 2 | $\mathcal{I}(\mathrm{CSM}) = 1$ |

$$+ x(t^{-2}\chi_1 + t^2\chi_2)$$
$$+ x^2(t^{-4}(\zeta_1\chi_1\chi_2 + \zeta_1\chi_2 + 1) + t^4(\zeta_2\chi_1\chi_2 + \zeta_2\chi_1 + 1) + \chi_1\chi_2 - \chi_1 - \chi_2 - 1)$$
$$+ x^3(t^{-6}(\zeta_1\chi_1\chi_2 + \zeta_1\chi_2 + \chi_1) + t^{-2}(-\zeta_1\chi_1\chi_2 - \zeta_1\chi_2 - \chi_1\chi_2)$$
$$+ t^2(-\zeta_2\chi_1\chi_2 - \zeta_2\chi_1 - \chi_1\chi_2) + t^6(\zeta_2\chi_1\chi_2 + \zeta_2\chi_1 + \chi_2))$$
$$+ x^4(t^{-8}(\zeta_1\chi_1\chi_2 + \zeta_1\chi_2 + \chi_1 + 2) + t^{-4}(-\zeta_1\chi_1\chi_2 - \zeta_1\chi_2) + t^4(-\zeta_2\chi_1\chi_2 - \zeta_2\chi_1)$$
$$+ t^8(\zeta_2\chi_1\chi_2 + \zeta_2\chi_1 + \chi_2 + 2) + 3\chi_1\chi_2 + \chi_1 + \chi_2 - 2)$$
$$+ x^5(t^{-10}(\zeta_1\chi_1\chi_2 + \zeta_1\chi_2 + 2\chi_1 + 1) + t^{-6}(-\chi_1 - 1)$$
$$+ t^{-2}(\zeta_1\chi_1 + \zeta_1 - 2\chi_1\chi_2 - 3\chi_1 - 1) + t^2(\zeta_2\chi_2 + \zeta_2 - 2\chi_1\chi_2 - 3\chi_2 - 1)$$
$$+ t^6(-\chi_2 - 1) + t^{10}(\zeta_2\chi_1\chi_2 + \zeta_2\chi_1 + 2\chi_2 + 1))$$
$$+ O(x^6)$$

$$\mathrm{HS}_{\mathrm{NS5}}(\mathrm{CSM}) = 1 + a\chi_1 + a^2(\zeta_1\chi_1\chi_2 + \zeta_1\chi_2 + 1)$$
$$+ a^3(\zeta_1\chi_1\chi_2 + \zeta_1\chi_2 + \chi_1) + a^4(\zeta_1\chi_1\chi_2 + \zeta_1\chi_2 + \chi_1 + 2)$$
$$+ a^5(\zeta_1\chi_1\chi_2 + \zeta_1\chi_2 + 2\chi_1 + 1) + O(a^6)$$

$$\mathrm{HS}_{(1,2\kappa)}(\mathrm{CSM}) = \mathrm{HS}_{\mathrm{NS5}}(\mathrm{CSM})$$

---

| $\{1,1\}$ | 3 | $\mathcal{I}(\mathrm{CSM}) = 1$ |

$$+ x(t^{-2}\chi_1 + t^2\chi_2)$$
$$+ x^2(t^{-4} + t^4 + \chi_1\chi_2 - \chi_1 - \chi_2 - 1)$$
$$+ x^3(t^{-6}(\zeta_1\chi_1\chi_2 + \zeta_1\chi_2 + \chi_1) + t^{-2}(-\chi_1\chi_2) + t^6(\zeta_2\chi_1\chi_2 + \zeta_2\chi_1 + \chi_2) - t^2\chi_1\chi_2)$$
$$+ x^4(t^{-8}(\zeta_1\chi_1\chi_2 + \zeta_1\chi_2 + 1) + t^{-4}(-\zeta_1\chi_1\chi_2 - \zeta_1\chi_2) + t^4(-\zeta_2\chi_1\chi_2 - \zeta_2\chi_1)$$
$$+ t^8(\zeta_2\chi_1\chi_2 + \zeta_2\chi_1 + 1) + 3\chi_1\chi_2 + \chi_1 + \chi_2 - 2)$$
$$+ x^5(t^{-10}(\zeta_1\chi_1\chi_2 + \zeta_1\chi_2 + \chi_1) + t^{-6}(-\zeta_1\chi_1\chi_2 - \zeta_1\chi_2) + t^{-2}(-\chi_1\chi_2 - 3\chi_1 - 1)$$
$$+ t^2(-\chi_1\chi_2 - 3\chi_2 - 1) + t^6(-\zeta_2\chi_1\chi_2 - \zeta_2\chi_1) + t^{10}(\zeta_2\chi_1\chi_2 + \zeta_2\chi_1 + \chi_2))$$
$$+ O(x^6)$$

$$\mathrm{HS}_{\mathrm{NS5}}(\mathrm{CSM}) = 1 + a\chi_1 + a^2 + a^3(\zeta_1\chi_1\chi_2 + \zeta_1\chi_2 + \chi_1)$$
$$+ a^4(\zeta_1\chi_1\chi_2 + \zeta_1\chi_2 + 1) + a^5(\zeta_1\chi_1\chi_2 + \zeta_1\chi_2 + \chi_1) + O(a^6)$$

$$\mathrm{HS}_{(1,2\kappa)}(\mathrm{CSM}) = \mathrm{HS}_{\mathrm{NS5}}(\mathrm{CSM})$$

---

| $\{1,1\}$ | 4 | $\mathcal{I}(\mathrm{CSM}) = 1$ |

$$+ x(t^{-2}\chi_1 + t^2\chi_2)$$
$$+ x^2(t^{-4} + t^4 + \chi_1\chi_2 - \chi_1 - \chi_2 - 1)$$
$$+ x^3(t^{-6}\chi_1 + t^{-2}(-\chi_1\chi_2) + t^6\chi_2 - t^2\chi_1\chi_2)$$
$$+ x^4(t^{-8}(\zeta_1\chi_1\chi_2 + \zeta_1\chi_2 + 1) + t^8(\zeta_2\chi_1\chi_2 + \zeta_2\chi_1 + 1) + 3\chi_1\chi_2 + \chi_1 + \chi_2 - 2)$$
$$+ x^5(t^{-10}(\zeta_1\chi_1\chi_2 + \zeta_1\chi_2 + \chi_1) + t^{-6}(-\chi_1\chi_2 - \zeta_1\chi_2) + t^{-2}(-\chi_1\chi_2 - 3\chi_1 - 1)$$
$$+ t^2(-\chi_1\chi_2 - 3\chi_2 - 1) + t^6(-\zeta_2\chi_1\chi_2 - \zeta_2\chi_1) + t^{10}(\zeta_2\chi_1\chi_2 + \zeta_2\chi_1 + \chi_2))$$
$$+ O(x^6)$$

$$\mathrm{HS}_{\mathrm{NS5}}(\mathrm{CSM}) = 1 + a\chi_1 + a^2 + a^3\chi_1 + a^4(\zeta_1\chi_1\chi_2 + \zeta_1\chi_2 + 1)$$
$$+ a^5(\zeta_1\chi_1\chi_2 + \zeta_1\chi_2 + \chi_1) + O(a^6)$$

$$\text{HS}_{(1,2\kappa)}(\text{CSM}) = \text{HS}_{\text{NS5}}(\text{CSM})$$

| $\{N,M\}$ | $\kappa$ | |
|---|---|---|
| $\{1,1\}$ | 5 | $\mathcal{I}(\text{CSM}) = 1$ |

$\{1,1\}$  5  $\mathcal{I}(\text{CSM}) = 1$
$$+ x(t^{-2}\chi_1 + t^2\chi_2)$$
$$+ x^2(t^{-4} + t^4 + \chi_1\chi_2 - \chi_1 - \chi_2 - 1)$$
$$+ x^3(t^{-6}\chi_1 + t^{-2}(-\chi_1\chi_2) + t^6\chi_2 - t^2\chi_1\chi_2)$$
$$+ x^4(t^{-8} + t^8 + 3\chi_1\chi_2 + \chi_1 + \chi_2 - 2)$$
$$+ x^5(t^{-10}(\zeta_1\chi_1\chi_2 + \zeta_1\chi_2 + \chi_1) + t^{-2}(-\chi_1\chi_2 - 3\chi_1 - 1) + t^2(-\chi_1\chi_2 - 3\chi_2 - 1)$$
$$\quad + t^{10}(\zeta_2\chi_1\chi_2 + \zeta_2\chi_1 + \chi_2))$$
$$+ O(x^6)$$

$$\text{HS}_{\text{NS5}}(\text{CSM}) = 1 + a\chi_1 + a^2 + a^3\chi_1 + a^4 + a^5(\zeta_1\chi_1\chi_2 + \zeta_1\chi_2 + \chi_1) + O(a^6)$$

$$\text{HS}_{(1,2\kappa)}(\text{CSM}) = \text{HS}_{\text{NS5}}(\text{CSM})$$

$\{1,2\}$  4  $\mathcal{I}(\text{CSM}) = 1$
$$+ x(t^{-2}\chi_1 + t^2\chi_2)$$
$$+ x^2(t^{-4} + t^4 + 2\chi_1\chi_2 - \chi_1 - \chi_2 - 1)$$
$$+ x^3(t^{-6}(\zeta_1\chi_1\chi_2 + \zeta_1\chi_2 + \chi_1) + t^{-2}(-2\chi_1\chi_2 - \chi_1 + \chi_2) + t^2(-2\chi_1\chi_2 + \chi_1 - \chi_2)$$
$$\quad + t^6(\zeta_2\chi_1\chi_2 + \zeta_2\chi_1 + \chi_2))$$
$$+ O(x^4)$$

$$\text{HS}_{\text{NS5}}(\text{CSM}) = 1 + a\chi_1 + a^2 + a^3(\zeta_1\chi_1\chi_2 + \zeta_1\chi_2 + \chi_1) + O(a^4)$$

$$\text{HS}_{(1,2\kappa)}(\text{CSM}) = \text{HS}_{\text{NS5}}(\text{CSM})$$

**Table 8:** Expanded index and Hilbert series for the linear $\text{SO}(2N)_{+2\kappa} \times \text{Sp}(M)_{-\kappa} \times \text{SO}(2N)_{+2\kappa}$ CSM theory in Figure 12 for some values of $N$, $M$ and $|\kappa| \geqslant M$ (for which the theory is good).

| $\{N,M\}$ | $\kappa$ | Index and Hilbert series for CSM $= \text{SO}(2N+1)_{+2\kappa} \times \text{Sp}(M)_{-\kappa} \times \text{SO}(2N+1)_{+2\kappa}$ |
|---|---|---|
| $\{1,1\}$ | 1 | $\mathcal{I}(\text{CSM}) = 1$ |

$\{1,1\}$  1  $\mathcal{I}(\text{CSM}) = 1$
$$+ x(t^{-2}\zeta_1\chi_1\chi_2 + t^2\zeta_2\chi_1\chi_2)$$
$$+ x^2(t^{-4}(\zeta_1\chi_1\chi_2 + 2) + t^4(\zeta_2\chi_1\chi_2 + 2) + \zeta_1\zeta_2 - \zeta_1\chi_1\chi_2 - \zeta_2\chi_1\chi_2 - 1)$$
$$+ x^3(t^{-6}(2\zeta_1\chi_1\chi_2 + 1) + t^{-2}(-\zeta_1\zeta_2 - \zeta_1\chi_1\chi_2 - 1) + t^2(-\zeta_1\zeta_2 - \zeta_2\chi_1\chi_2 - 1)$$
$$\quad + t^6(2\zeta_2\chi_1\chi_2 + 1) - \zeta_1\zeta_2\chi_1\chi_2 - \zeta_1 - \zeta_2 - \chi_1\chi_2)$$
$$+ x^4(t^{-8}(2\zeta_1\chi_1\chi_2 + 3) + t^{-4}(-\zeta_1\chi_1\chi_2 - 1) + t^{-2}(\zeta_1\zeta_2\chi_1\chi_2 + \zeta_1 + \zeta_2 + \chi_1\chi_2)$$
$$\quad + t^2(\zeta_1\zeta_2\chi_1\chi_2 + \zeta_1 + \zeta_2 + \chi_1\chi_2) + t^4(-\zeta_2\chi_1\chi_2 - 1) + t^8(2\zeta_2\chi_1\chi_2 + 3)$$
$$\quad + 3\zeta_1\zeta_2 + \zeta_1\chi_1\chi_2 + \zeta_2\chi_1\chi_2 - 2)$$
$$+ x^5(t^{-10}(3\zeta_1\chi_1\chi_2 + 2) + t^{-6}(-\zeta_1\chi_1\chi_2 - 1) + t^{-2}(-\zeta_1\zeta_2 - 3\zeta_1\chi_1\chi_2 + \zeta_2\chi_1\chi_2 - 1)$$
$$\quad + t^2(-\zeta_1\zeta_2 + \zeta_1\chi_1\chi_2 - 3\zeta_2\chi_1\chi_2 - 1) + t^6(-\zeta_2\chi_1\chi_2 - 1)$$
$$\quad + t^{10}(3\zeta_2\chi_1\chi_2 + 2) - \zeta_1\zeta_2\chi_1\chi_2 - \zeta_1 - \zeta_2 - \chi_1\chi_2)$$
$$+ O(x^6)$$

$$\text{HS}_{\text{NS5}}(\text{CSM}) = 1 + a(\zeta_1\chi_1\chi_2) + a^2(\zeta_1\chi_1\chi_2 + 2) + a^3(2\zeta_1\chi_1\chi_2 + 1) + a^4(2\zeta_1\chi_1\chi_2 + 3)$$
$$+ a^5(3\zeta_1\chi_1\chi_2 + 2) + O(a^6)$$

$$\text{HS}_{(1,2\kappa)}(\text{CSM}) = \text{HS}_{\text{NS5}}(\text{CSM})$$

$\{1,1\}$  2  $\mathcal{I}(\text{CSM}) = 1$
$$+ x^2((\zeta_1\chi_1\chi_2 + 1)t^{-4} + t^4(\zeta_2\chi_1\chi_2 + 1) - 1)$$
$$+ x^3(\zeta_1\chi_1\chi_2 t^{-6} + \zeta_2\chi_1\chi_2 t^6 - \zeta_1\chi_1\chi_2 t^{-2} - \zeta_2\chi_1\chi_2 t^2)$$
$$+ x^4((\zeta_1\chi_1\chi_2 + 2)t^{-8} + t^8(\zeta_2\chi_1\chi_2 + 2) - \zeta_1\chi_1\chi_2 t^{-4} - \zeta_2\chi_1\chi_2 t^4 - 2)$$
$$+ x^5((\zeta_1\chi_1\chi_2 + 1)t^{-10} + t^{10}(\zeta_2\chi_1\chi_2 + 1) - t^6 - t^{-6} + \zeta_1\chi_1\chi_2 t^{-2}$$
$$\quad + \zeta_2\chi_1\chi_2 t^2 + \chi_1\chi_2)$$
$$+ O(x^6)$$

$$\text{HS}_{\text{NS5}}(\text{CSM}) = 1 + a^2(\zeta_1\chi_1\chi_2 + 1) + a^3\zeta_1\chi_1\chi_2 + a^4(\zeta_1\chi_1\chi_2 + 2)$$

$$+ a^5(\zeta_1\chi_1\chi_2 + 1) + O(a^6)$$

$$\text{HS}_{(1,2\kappa)}(\text{CSM}) = \text{HS}_{\text{NS5}}(\text{CSM})$$

| $\{1,1\}$ | 3 | $\mathcal{I}(\text{CSM}) = 1$ |
|---|---|---|
| | | $+ x^2(t^4 + t^{-4} - 1)$ |
| | | $+ x^3(\zeta_1\chi_1\chi_2 t^{-6} + \zeta_2\chi_1\chi_2 t^6)$ |
| | | $+ x^4((\zeta_1\chi_1\chi_2 + 1)t^{-8} + t^8(\zeta_2\chi_1\chi_2 + 1) - \zeta_1\chi_1\chi_2 t^{-4} - \zeta_2\chi_1\chi_2 t^4 - 2)$ |
| | | $+ x^5(\zeta_1\chi_1\chi_2 t^{-10} + \zeta_2\chi_1\chi_2 t^{10} - \zeta_1\chi_1\chi_2 t^{-6} - \zeta_2\chi_1\chi_2 t^6 + \chi_1\chi_2)$ |
| | | $+ O(x^6)$ |
| | | $\text{HS}_{\text{NS5}}(\text{CSM}) = 1 + a^2 + a^3\zeta_1\chi_1\chi_2 + a^4(\zeta_1\chi_1\chi_2 + 1) + a^5\zeta_1\chi_1\chi_2 + O(a^6)$ |
| | | $\text{HS}_{(1,2\kappa)}(\text{CSM}) = \text{HS}_{\text{NS5}}(\text{CSM})$ |
| $\{1,1\}$ | 4 | $\mathcal{I}(\text{CSM}) = 1$ |
| | | $+ x^2(t^4 + t^{-4} - 1)$ |
| | | $+ x^4((\zeta_1\chi_1\chi_2 + 1)t^{-8} + t^8(\zeta_2\chi_1\chi_2 + 1) - 2)$ |
| | | $+ x^5(\zeta_1\chi_1\chi_2 t^{-10} + \zeta_2\chi_1\chi_2 t^{10} - \zeta_1\chi_1\chi_2 t^{-6} - \zeta_2\chi_1\chi_2 t^6 + \chi_1\chi_2)$ |
| | | $+ O(x^6)$ |
| | | $\text{HS}_{\text{NS5}}(\text{CSM}) = 1 + a^2 + a^4(\zeta_1\chi_1\chi_2 + 1) + a^5\zeta_1\chi_1\chi_2 + O(a^6)$ |
| | | $\text{HS}_{(1,2\kappa)}(\text{CSM}) = \text{HS}_{\text{NS5}}(\text{CSM})$ |
| $\{1,1\}$ | 5 | $\mathcal{I}(\text{CSM}) = 1$ |
| | | $+ x^2(t^4 + t^{-4} - 1)$ |
| | | $+ x^4(t^8 + t^{-8} - 2)$ |
| | | $+ x^5(\zeta_1\chi_1\chi_2 t^{-10} + \zeta_2\chi_1\chi_2 t^{10} + \chi_1\chi_2)$ |
| | | $+ O(x^6)$ |
| | | $\text{HS}_{\text{NS5}}(\text{CSM}) = 1 + a^2 + a^4 + a^5\zeta_1\chi_1\chi_2 + O(a^6)$ |
| | | $\text{HS}_{(1,2\kappa)}(\text{CSM}) = \text{HS}_{\text{NS5}}(\text{CSM})$ |

**Table 9:** Index and Hilbert series for the linear $\text{SO}(2N + 1)_{+2\kappa} \times \text{Sp}(M)_{-\kappa} \times \text{SO}(2N + 1)_{+2\kappa}$ CSM theory for some values of $N$, $M$ and $|\kappa| \geqslant M$ (for which the theory is good).

| $\{N, M\}$ | $\kappa$ | Index and Hilbert series for CSM $= \text{Sp}(N)_{+\kappa} \times \text{SO}(2M + 1) \times \text{Sp}(N)_{-\kappa}$ |
|---|---|---|
| $\{1,1\}$ | 1 | $\mathcal{I}(\text{CSM}) = 1$ |
| | | $+ x(t^{-2}\zeta + t^2\zeta)$ |
| | | $+ x^2(t^{-4}(\zeta + 2) + t^4(\zeta + 2) - 2\zeta)$ |
| | | $+ x^3(t^{-6}(2\zeta + 1) + t^{-2}(-\zeta - 2) + t^2(-\zeta - 2) + t^6(2\zeta + 1) - 2\zeta\chi - 2\chi)$ |
| | | $+ x^4(t^{-8}(2\zeta + 3) + t^{-4}(-\zeta - 1) + t^{-2}(2\zeta\chi + 2\chi) + t^2(2\zeta\chi + 2\chi)$ |
| | | $\qquad + t^4(-\zeta - 1) + t^8(2\zeta + 3) + 2\zeta + 1)$ |
| | | $+ x^5(t^{-10}(3\zeta + 2) + t^{-6}(-\zeta - 1) + t^{-2}(-2\zeta - 2) + t^2(-2\zeta - 2) + t^6(-\zeta - 1)$ |
| | | $\qquad + t^{10}(3\zeta + 2) - 2\zeta\chi - 2\chi)$ |
| | | $+ O(x^6)$ |
| | | $\text{HS}_{\text{NS5}}(\text{CSM}) = 1 + a\zeta + a^2(\zeta + 2) + a^3(2\zeta + 1) + a^4(2\zeta + 3) + a^5(3\zeta + 2) + O(a^6)$ |
| | | $\text{HS}_{(1,2\kappa)}(\text{CSM}) = \text{HS}_{\text{NS5}}(\text{CSM})$ |
| $\{1,1\}$ | 2 | $\mathcal{I}(\text{CSM}) = 1$ |
| | | $+ x^1(t^2\zeta)$ |
| | | $+ x^2(t^{-4} + t^2(2\zeta\chi + 2\chi) + t^4(\zeta + 2) - \zeta)$ |
| | | $+ x^3(t^{-6}\zeta + t^{-2}(-1) + t^2(-\zeta - 2) + t^6(2\zeta + 1) - 4\zeta\chi - 4\chi)$ |
| | | $+ x^4(t^{-8}(\zeta + 1) + t^{-4}(-\zeta) + t^{-2}(2\zeta\chi + 2\chi) + t^2(-2\zeta\chi - 2\chi) + t^4(-3\zeta - 3)$ |
| | | $\qquad + t^6(2\zeta\chi + 2\chi) + t^8(2\zeta + 3) - \zeta - 1)$ |

$$+ x^5(t^{-10}\zeta + t^{-6}(-\zeta) + t^{-2}(5\zeta + 3) + t^2(6\zeta + 3) + t^6(-2\zeta - 2)$$
$$+ t^{10}(3\zeta + 2) + 6\zeta\chi + 6\chi)$$
$$+ O(x^6)$$

$$\text{HS}_{\text{NS5}}(\text{CSM}) = 1 + a^2 + a^3\zeta + a^4(\zeta + 1) + a^5\zeta + O(a^6)$$

$$\text{HS}_{(1,2\kappa)}(\text{CSM}) = \text{HS}_{\text{NS5}}(\text{CSM})$$

| $\{1,1\}$ | 3 | $\mathcal{I}(\text{CSM}) = 1$ |
|---|---|---|

$$+ x^1(t^2\zeta)$$
$$+ x^2(t^{-4} + t^2(2\zeta\chi + 2\chi) + t^4(\zeta + 2) - \zeta)$$
$$+ x^3(t^{-2}(-1) + t^2(-\zeta - 2) + t^6(2\zeta + 1) - 4\zeta\chi - 4\chi)$$
$$+ x^4(t^{-8} + t^{-2}(2\zeta\chi + 2\chi) + t^2(-2\zeta\chi - 2\chi) + t^4(-3\zeta - 3) + t^6(2\zeta\chi + 2\chi)$$
$$+ t^8(2\zeta + 3) - \zeta - 1)$$
$$+ x^5(t^{-10}\zeta + t^{-2}(3\zeta + 3) + t^2(6\zeta + 3) + t^4(2\zeta\chi + 2\chi) + t^6(-2\zeta - 2)$$
$$+ t^{10}(3\zeta + 2) + 8\zeta\chi + 8\chi)$$
$$+ O(x^6)$$

$$\text{HS}_{\text{NS5}}(\text{CSM}) = 1 + a^2 + a^4 + a^5\zeta + O(a^6)$$

$$\text{HS}_{(1,2\kappa)}(\text{CSM}) = \text{HS}_{\text{NS5}}(\text{CSM})$$

| $\{1,1\}$ | 4 | $\mathcal{I}(\text{CSM}) = 1$ |
|---|---|---|

$$+ x^1(t^2\zeta)$$
$$+ x^2(t^{-4} + t^2(2\zeta\chi + 2\chi) + t^4(\zeta + 2) - \zeta)$$
$$+ x^3(t^{-2}(-1) + t^2(-\zeta - 2) + t^6(2\zeta + 1) - 4\zeta\chi - 4\chi)$$
$$+ x^4(t^{-8} + t^{-2}(2\zeta\chi + 2\chi) + t^2(-2\zeta\chi - 2\chi) + t^4(-3\zeta - 3)$$
$$+ t^6(2\zeta\chi + 2\chi) + t^8(2\zeta + 3) - \zeta - 1)$$
$$+ x^5(t^{-2}(3\zeta + 3) + t^2(6\zeta + 3) + t^4(2\zeta\chi + 2\chi) + t^6(-2\zeta - 2)$$
$$+ t^{10}(3\zeta + 2) + 8\zeta\chi + 8\chi)$$
$$+ O(x^6)$$

$$\text{HS}_{\text{NS5}}(\text{CSM}) = 1 + a^2 + a^4 + O(a^6)$$

$$\text{HS}_{(1,2\kappa)}(\text{CSM}) = \text{HS}_{\text{NS5}}(\text{CSM})$$

**Table 10:** Expanded index and Hilbert series for the linear $\text{Sp}(N)_{+\kappa} \times \text{SO}(2M + 1) \times \text{Sp}(N)_{-\kappa}$ CSM theory in Figure 13 for some values of $N \geqslant M$ and $|\kappa| \geqslant N$ (for which the theory is good).

## B.4   A puzzle

In the following tables one can find the index expansion and the Hilbert series for the puzzle theories of Section 5, together with the Hilbert series for the quiver-like object "MQ".

| $\{N, L, M\}$ | $\kappa$ | Index and Hilbert series |
|---|---|---|
| $\{1,1,1\}$ | 4 | $\mathcal{I}(\text{CSM}) = 1$ |

$$+ x(t^{-2}(\chi_1 + \chi_3))$$
$$+ x^2(t^{-4}(\zeta_3\chi_3 + \zeta_3 + \chi_1\chi_3 + 2) + \chi_1\chi_3 - \chi_1 - \chi_3 - 1)$$
$$+ x^3(t^{-6}(\zeta_1\chi_1\chi_3 + \zeta_1\chi_3 + \zeta_3\chi_1\chi_3 + \zeta_3\chi_1 + \zeta_3\chi_3 + \zeta_3 + 2\chi_1 + 2\chi_3)$$
$$+ t^{-2}(-\zeta_3\chi_3 - \zeta_3 - 3\chi_1\chi_3 - 1) + t^2(-\chi_1\chi_3 - \chi_1 - \chi_3 + 1))$$
$$+ x^4(t^{-8}(\zeta_1\zeta_3\chi_1\chi_3 + \zeta_1\zeta_3\chi_1 + \zeta_1\zeta_3\chi_3 + \zeta_1\zeta_3$$
$$+ \zeta_1\chi_1\chi_3 + \zeta_1\chi_1 + \zeta_1\chi_3 + \zeta_1 + \zeta_3\chi_1\chi_3 + \zeta_3\chi_1$$
$$+ 2\zeta_3\chi_3 + 2\zeta_3 + 2\chi_1\chi_3 + \chi_3 + 4) + t^{-4}(-\zeta_1\chi_1\chi_3 - \zeta_1\chi_3$$
$$- 2\zeta_3\chi_1\chi_3 - 2\zeta_3\chi_1 - 2\zeta_3\chi_3 - 2\zeta_3 + \chi_1\chi_3 - 2\chi_1 - 2\chi_3 - 1)$$
$$+ t^4(\chi_1 + \chi_3) + 2\chi_1\chi_3 + 2\chi_1 + 2\chi_3 - 3)$$
$$+ O(x^5)$$

$$\mathrm{HS}_{\mathrm{NS5}}(\mathrm{CSM}) = 1 + a(\chi_1 + \chi_3) + a^2(\zeta_3\chi_3 + \zeta_3 + \chi_1\chi_3 + 2)$$
$$+ a^3(\zeta_1\chi_1\chi_3 + \zeta_1\chi_3 + \zeta_3\chi_1\chi_3 + \zeta_3\chi_1 + \zeta_3\chi_3 + \zeta_3 + 2\chi_1 + 2\chi_3)$$
$$+ a^4(\zeta_1\zeta_3\chi_1\chi_3 + \zeta_1\zeta_3\chi_1 + \zeta_1\zeta_3\chi_3 + \zeta_1\zeta_3 + \zeta_1\chi_1\chi_3$$
$$+ \zeta_1\chi_1 + \zeta_1\chi_3 + \zeta_1 + \zeta_3\chi_1\chi_3 + \zeta_3\chi_1 + 2\zeta_3\chi_3 + 2\zeta_3 + 2\chi_1\chi_3 + \chi_3 + 4) + O(a^5)$$

$$\mathrm{HS}_{\mathcal{C}}(``\mathsf{MQ}_{\mathrm{NS5}}") = 1 + a(\chi_1 + \chi_3) + a^2(\zeta_3\chi_3 + \zeta_3 + \chi_1\chi_3 + 2)$$
$$+ a^3(\zeta_1\chi_1 + \zeta_1 + \zeta_3\chi_1\chi_3 + \zeta_3\chi_1 + \zeta_3\chi_3 + \zeta_3 + 2\chi_1 + 2\chi_3)$$
$$+ a^4(\zeta_1\zeta_3\chi_1\chi_3 + \zeta_1\zeta_3\chi_1 + \zeta_1\zeta_3\chi_3 + \zeta_1\zeta_3 + \zeta_1\chi_1\chi_3 + \zeta_1\chi_1$$
$$+ \zeta_1\chi_3 + \zeta_1 + \zeta_3\chi_1\chi_3 + \zeta_3\chi_1 + 2\zeta_3\chi_3 + 2\zeta_3 + 2\chi_1\chi_3 + \chi_3 + 4)$$
$$+ O(a^5)$$

| $\{N, L, M\}$ | $\kappa$ | Index and Hilbert series |
|---|---|---|

| $\{1,1,1\}$ | 5 | $\mathcal{I}(\mathrm{CSM}) = 1$ |

$$+ x(\chi_1 + \chi_3)t^{-2}$$
$$+ x^2((\zeta_3\chi_3 + \zeta_3 + \chi_1\chi_3 + 2)t^{-4} + \chi_1\chi_3 - \chi_1 - \chi_3 - 1)$$
$$+ x^3((\zeta_3\chi_1\chi_3 + \zeta_3\chi_1 + \zeta_3\chi_3 + \zeta_3 + 2\chi_1 + 2\chi_3)t^{-6}$$
$$+ (-\zeta_3\chi_3 - \zeta_3 - 3\chi_1\chi_3 - 1)t^{-2} + t^2(-\chi_1\chi_3 - \chi_1 - \chi_3 + 1))$$
$$+ x^4((\zeta_1\chi_1\chi_3 + \zeta_1\chi_3 + \zeta_3\chi_1\chi_3 + \zeta_3\chi_1 + 2\zeta_3\chi_3$$
$$+ 2\zeta_3 + 2\chi_1\chi_3 + \chi_3 + 4)t^{-8} + (-2\zeta_3\chi_1\chi_3 - 2\zeta_3\chi_1 - 2\zeta_3\chi_3$$
$$- 2\zeta_3 + \chi_1\chi_3 - 2\chi_1 - 2\chi_3 - 1)t^{-4} + t^4(\chi_1 + \chi_3) + 2\chi_1\chi_3 + 2\chi_1 + 2\chi_3 - 3)$$
$$+ O(x^5)$$

$$\mathrm{HS}_{\mathrm{NS5}}(\mathrm{CSM}) = 1 + a(\chi_1 + \chi_3) + a^2(\zeta_3\chi_3 + \zeta_3 + \chi_1\chi_3 + 2)$$
$$+ a^3(\chi_3\zeta_3\chi_1 + \zeta_3\chi_1 + \zeta_3\chi_3 + \zeta_3 + 2\chi_1 + 2\chi_3)$$
$$+ a^4(\chi_3\zeta_1\chi_1 + \zeta_1\chi_3 + \chi_3\zeta_3\chi_1 + \zeta_3\chi_1 + 2\zeta_3\chi_3 + 2\zeta_3 + 2\chi_1\chi_3 + \chi_3 + 4)$$
$$+ O(a^5)$$

$$\mathrm{HS}_{\mathcal{C}}(``\mathsf{MQ}_{\mathrm{NS5}}") = 1 + a(\chi_1 + \chi_3) + a^2(\zeta_3\chi_3 + \zeta_3 + \chi_1\chi_3 + 2)$$
$$+ a^3(\zeta_3\chi_1\chi_3 + \zeta_3\chi_1 + \zeta_3\chi_3 + \zeta_3 + 2\chi_1 + 2\chi_3)$$
$$+ a^4(\zeta_1\chi_1 + \zeta_1 + \zeta_3\chi_1\chi_3 + \zeta_3\chi_1 + 2\zeta_3\chi_3 + 2\zeta_3 + 2\chi_1\chi_3 + \chi_3 + 4)$$

| $\{1,1,1\}$ | 6 | $\mathcal{I}(\mathrm{CSM}) = 1$ |

$$+ x(\chi_1 + \chi_3)t^{-2}$$
$$+ x^2((\zeta_3\chi_3 + \zeta_3 + \chi_1\chi_3 + 2)t^{-4} + \chi_1\chi_3 - \chi_1 - \chi_3 - 1)$$
$$+ x^3((\zeta_3\chi_1\chi_3 + \zeta_3\chi_1 + \zeta_3\chi_3 + \zeta_3 + 2\chi_1 + 2\chi_3)t^{-6}$$
$$+ (-\zeta_3\chi_3 - \zeta_3 - 3\chi_1\chi_3 - 1)t^{-2} + t^2(-\chi_1\chi_3 - \chi_1 - \chi_3 + 1))$$
$$+ O(x^4)$$

$$\mathrm{HS}_{\mathrm{NS5}}(\mathrm{CSM}) = 1 + a(\chi_1 + \chi_3) + a^2(\zeta_3\chi_3 + \zeta_3 + \chi_1\chi_3 + 2)$$
$$+ a^3(\zeta_3\chi_1\chi_3 + \zeta_3\chi_1 + \zeta_3\chi_3 + \zeta_3 + 2\chi_1 + 2\chi_3) + O(a^4)$$

$$\mathrm{HS}_{\mathcal{C}}(``\mathsf{MQ}_{\mathrm{NS5}}") = 1 + a(\chi_1 + \chi_3) + a^2(\zeta_3\chi_3 + \zeta_3 + \chi_1\chi_3 + 2)$$
$$+ a^3(\zeta_3\chi_1\chi_3 + \zeta_3\chi_1 + \zeta_3\chi_3 + \zeta_3 + 2\chi_1 + 2\chi_3) + O(a^4)$$

| $\{1,2,2\}$ | 5 | $\mathcal{I}(\mathrm{CSM}) = 1$ |

$$+ x(\zeta_3 + \chi_1)t^{-2}$$
$$+ O(x^3)$$

$$\mathrm{HS}_{\mathrm{NS5}}(\mathrm{CSM}) = 1 + a(\zeta_3 + \chi_1) + O(a^2)$$

$$\mathrm{HS}_{\mathcal{C}}(``\mathsf{MQ}_{\mathrm{NS5}}") = 1 + a(\zeta_3 + \chi_1) + O(a^2)$$

**Table 11:** Expanded index of CSM theory $\mathrm{SO}(2N)_{+2\kappa} \times \mathrm{Sp}(N+L)_{-\kappa} \times \mathrm{SO}(2M)$ (see Figure 5.1) and monopole formula of the quiver-like object "$\mathsf{MQ}_{\mathrm{NS5}}$". The results match if the following fugacity map is used $\zeta_1 \to \zeta_1\chi_1\chi_3$.

| $\{N, L, M\}$ | $\kappa$ | Index and Hilbert series |
|---|---|---|
| $\{1,1,1\}$ | 3 | $\mathcal{I}(\mathrm{CSM}) = 1$ |

$$+ x \chi_1 t^{-2}$$
$$+ x^2((\zeta_1\chi_1 + \zeta_1 + \chi_1 + 3)t^{-4} + t^2 - \chi_1 - 1)$$
$$+ x^3((2\zeta_1\chi_1 + 2\zeta_1 + 3\chi_1 + 1)t^{-6} + (-\zeta_1\chi_1 - \zeta_1 - 2\chi_1 - 2)t^{-2} - t^4 + t^2 - 1)$$
$$+ O(x^4)$$

$$\mathrm{HS}_{\mathrm{NS5}}(\mathrm{CSM}) = 1 + a\chi_1 + a^2(\zeta_1\chi_1 + \zeta_1 + \chi_1 + 3)$$
$$+ a^3(2\zeta_1\chi_1 + 2\zeta_1 + 3\chi_1 + 1) + O(a^4)$$

$$\mathrm{HS}_{\mathcal{C}}(\text{``}\mathsf{MQ}_{\mathrm{NS5}}\text{''}) = 1 + a + 6a^2 + 8a^3 + O(a^4)$$

| $\{1,1,1\}$ | 4 | $\mathcal{I}(\mathrm{CSM}) = 1$ |

$$+ \chi_1 x t^{-2}$$
$$+ x^2((\chi_1 + 3)t^{-4} + t^2 - \chi_1 - 1)$$
$$+ x^3((\zeta_1\chi_1 + \zeta_1 + 3\chi_1 + 1)t^{-6} - t^4 + (-2\chi_1 - 2)t^{-2} + t^2 - 1)$$
$$+ O(x^4)$$

$$\mathrm{HS}_{\mathrm{NS5}}(\mathrm{CSM}) = 1 + a\chi_1 + a^2(\chi_1 + 3) + a^3(\zeta_1\chi_1 + \zeta_1 + 3\chi_1 + 1) + O(a^4)$$

$$\mathrm{HS}_{\mathcal{C}}(\text{``}\mathsf{MQ}_{\mathrm{NS5}}\text{''}) = 1 + a + 4a^2 + 6a^3 + O(a^4)$$

| $\{1,1,1\}$ | 5 | $\mathcal{I}(\mathrm{CSM}) = 1$ |

$$+ x \chi_1 t^{-2}$$
$$+ x^2((\chi_1 + 3)t^{-4} + t^2 - \chi_1 - 1)$$
$$+ x^3((3\chi_1 + 1)t^{-6} - t^4 + (-2\chi_1 - 2)t^{-2} + t^2 - 1)$$
$$+ O(x^4)$$

$$\mathrm{HS}_{\mathrm{NS5}}(\mathrm{CSM}) = 1 + a\chi_1 + a^2(\chi_1 + 3) + a^3(3\chi_1 + 1) + O(a^4)$$

$$\mathrm{HS}_{\mathcal{C}}(\text{``}\mathsf{MQ}_{\mathrm{NS5}}\text{''}) = 1 + a + 4a^2 + 4a^3 + O(a^4)$$

**Table 12:** Expanded index of CSM theory $\mathrm{Sp}(2N)_{+\kappa} \times \mathrm{SO}(2N+2L+1)_{-2\kappa} \times \mathrm{Sp}(2M)$ (Figure 5.3) and monopole formula of the quiver-like object "$\mathsf{MQ}_{\mathrm{NS5}}$". The results match if $\zeta_1 = \chi_1 = 1$.

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
