# Peer review of "Orthosymplectic Chern-Simons Matter Theories: Global Forms, Dualities, and Vacua"

_SciPost Physics_

## Round 1 · Referee Report · Anonymous (Referee 1) · 2025-11-28

Report

This article generalizes the previous results of [arXiv:2503.02744] to 3d $\mathcal{N}=3, 4$ Chern-Simons-matter quiver gauge theories with orthosymplectic gauge groups. The authors derive magnetic quivers such that the Coulomb branch of the magnetic theory describes each maximal branch of the aforementioned electric theories. The primary technique relies on brane configurations consisting of D3-branes, an O3-orientifold plane, and various types of fivebranes. For the $\mathcal{N}=4$ case, the authors verify their results by comparing the Hilbert series of the magnetic quiver against the corresponding limit of the supersymmetric index. This paper presents a number of novel and interesting results. It definitely deserves publication in SciPost, subject to the following minor corrections.

Requested changes

  1. In Example 1 of Section 5, the authors state: If one tries to move ... all the way to the left (or all the way to the right), the putative magnetic quivers fail to match the NS5-branch limit of the CSM index.'' However, the problematic magnetic quivers arising from such moves are not explicitly shown. While the authors provide the correct magnetic quiver in (5.2), I recommend that they also explicitly present the specific quivers that fail. This would be highly instructive for future work, helping readers understand why such brane moves do not yield thecorrect'' result. The same recommendation applies to Example 2.

  2. In (5.2), the magnetic quiver contains a half-hypermultiplet in the representation $(2N, 2M)$ of $SO(2N) \times SO(2M)$. To clarify the contribution of such matter, it would be useful for the authors to explicitly write down the conformal dimension of the monopole operators in the monopole formula used to compute the Coulomb branch Hilbert series of (5.2). This request also applies to the half-hypermultiplet in the representation $(2N, 2M)$ of $Sp(N) \times Sp(M)$ in (5.4).

Recommendation

Ask for minor revision

---

## Round 1 · Referee Report · Anonymous (Referee 2) · 2025-12-11

Report

The paper is devoted to the study of maximal branches in the 3d orthosymplectic Chern-Simons matter theories with $\mathcal{N}\geq 3$ supersymmetry that can be engineered with brane set-ups in Type IIB that include O3 planes. The main goal is to find magnetic quivers for such maximal branches, i.e. 3d $\mathcal{N}=4$ non-CS quivers whose Coulomb branches coincide with the maximal branches of the orthosymplectic CSM theory of interest. The strategy to determine such magnetic quivers is an extension of the some used by a subset of the authors in a previous paper to study maximal branches of unitary CSM theories. This involves various brane moves that are implemented in the quivers by Giveon-Kutasov dualities and generalizations thereof, as well as a proposal for how to read off the magnetic quiver from a given brane configuration. Such a proposal is extensively tested in the subclass of orthosymplectic CSM theories that have $\mathcal{N}=4$ supersymmetry, for which one can independently extract the Hilbert series of the maximal branches from a limit of the superconformal index and compare it against the Coulomb branch Hilbert series of the magnetic quivers.

The results of this paper are new and provide an important setup forward in our understanding of the moduli spaces of three-dimensional supersymmetric gauge theories. The strategy is clearly explained, with several examples provided as well as explicit computations in the appendices. For these reasons, I suggest this paper for publication on Scipost, provided that the following points are addressed/clarified.

1) In the unitary case, when we apply a GK duality on a node of the quiver, the non-trivial mapping of the topological symmetry fugacities of the adjacent nodes can be deduced from the background CS couplings in the GK duality that the authors highlight in figure 1. Presumably, similar background couplings for the $\mathbb{Z}_2^{\mathcal{M}}$ and $\mathbb{Z}_2^{\mathcal{C}}$ are present also in the orthosymplectic versions of the GK duality. This might shed light on how the fugacity map for these symmetries works when applying a GK duality inside a orthosymplectic quiver, which is a problem that the authors address for quivers with 2 and 3 nodes, but not higher number of nodes. In particular, if what I suggest is possible, it means that applying a GK duality on a node can only affect the fugacities of its adjacent nodes, so the analysis that the authors do in the cases with 3 gauge nodes should be enough to extrapolate the result for a generic quiver.

2) In the unitary example with two gauge nodes of page 15, the authors explain that there are two distinct NS5 phases which however have the same magnetic quiver. Does this mean that there are two distinct but identical branches in the moduli space of this theory? My understanding is that this is not the case, since this theory has $\mathcal{N}=4$ supersymmetry and so it only has two maximal branches (Higgs and Coulomb), corresponding to the trivial $(1,\kappa)$ branch and the NS5 branch for just one of the two distinct phases mentioned above. I think it would be good to clarify this point to avoid confusion.

3) Similarly to the previous comment, it seems that in the example on page 18 for $N=M$ the two phases (1) and (2) are both possible, however there is probably only one NS5 branch.

4) I think it would be good to specify the range of validity of the equations (4.8)-(4.9) in terms of the parameters N,M. Moreover, the discussion around these equations is not completely clear. For example, from eq. (4.8b) it seems that for $N<M$ there is no NS5 branch for the $SO$ global form of the CSM theory, since the magnetic quiver that the authors find only captures the $O^-$ variant of the theory. Is this what the authors claim? I also wonder if perhaps the NS5 branch of the $SO$ CSM theory for $N<M$ corresponds to instead taking the gauge group of the magnetic quiver to be $Sp/\mathbb{Z}_2$ rather than just $Sp$. This modification would indeed affect the computation of the Coulomb branch Hilbert series of the magnetic quiver. I have a similar confusion for the other global forms of the CSM theory for $N<M$: since I believe equations (4.9) only hold for $N\geq M$, how can we get the NS5 branch for $N<M$?

5) The generalization of the procedure that the authors propose in section 4.4 for the case of $(p,q)$ branes is a bit surprising. For $(1,q)$ the authors previously proposed that each such brane should contribute with $q$ flavor to the magnetic quiver. This proposal was cross-checked in the $\mathcal{N}=4$ cases with index computations. The proposed generalization to $(p,q)$ is that it still contributes $q$ flavors, with no dependence on $p$. Since for these cases it is not possible to have an independent index check, it would be good if the authors could provide some argument for this proposal, or at least give some intuition for the independence of $p$ of their prescription.

6) The authors say that the quiver in (5.2) is unconventional, since the field associated to the link between the $SO(2N)$ and the $SO(2M)$ nodes should contribute to the monopole formula as half of a bi-vector hypermultiplet. Isn’t this just the same as having a half-hypermultiplet in such representation? Similarly for the quiver in (5.4). Moreover, in this second example I would call the representation of $Sp(N)\times Sp(M)$ bifundamental rather than bivector.

7) As a general comment, it feels a bit weird the choice of notation for the $O3^-$ plane of not drawing any line at all. This might be confusing in some drawings, so coming up with a better notation is advisable.

I have also found a few typos: - Page 3, first bulletpoint of the notation: “3 $\mathcal{N}=4$ vector multiplets” -> “3d $\mathcal{N}=4$ vector multiplets”. - Caption on figure 1 on page 6: “These fugacity assignment” -> “This fugacity assignment”. - Page 23, last paragraph: “The later is reflected” -> “The latter is reflected”. - On pages 27/28, sometimes $(1,\kappa)$ is written as $(1,k)$ instead. - The first couple of sentences of section 4.4 seem incomplete. - Page 32 below eq. (4.32): “$N$ full D3s ending in it” -> “$N$ full D3s ending on it”

Recommendation

Ask for minor revision

---

## Editorial Decision

awaiting_resubmission